



# Estimating seasonal bulk density of level sea ice using the data derived from in situ and ICESat-2 synergistic observations during MOSAiC

Yi Zhou[1,2], Xianwei Wang[1,2], Ruibo Lei[3], Arttu Jutila[4], Donald K. Perovich[5], Luisa von Albedyll[6], Dmitry V. Divine[7], Yu Zhang[8,9], Christian Haas[6,10]

[1]School of Oceanography, Shanghai Jiao Tong University, Shanghai, China
[2]Key Laboratory of Polar Ecosystem and Climate Change (Shanghai Jiao Tong University), Ministry of Education, Shanghai, China
[3]Key Laboratory for Polar Science, Ministry of Natural Resources, Polar Research Institute of China, Shanghai, China
[4]Finnish Meteorological Institute, Helsinki, Finland
[5]Thayer School of Engineering, Dartmouth College, Hanover, NH, USA
[6]Alfred Wegener Institute, Helmholtz Centre for Polar and Marine Research, Bremerhaven, Germany
[7]Norwegian Polar Institute, Tromsø, Norway
[8]College of Oceanography and Ecological Science, Shanghai Ocean University, Shanghai, China
[9]Southern Marine Science and Engineering Guangdong Laboratory, Zhuhai, China
[10]Institute of Environmental Physics, University of Bremen, Bremen, Germany

*Correspondence:* Xianwei Wang (xianwei.wang@sjtu.edu.cn) and Ruibo Lei (leiruibo@pric.org.cn)

**Abstract.** Satellite retrievals of Arctic sea ice thickness typically assume fixed values of sea ice bulk density (IBD), overlooking its seasonal evolution and spatial heterogeneity, which are influenced by factors such as the age, deformation, brine, and air inclusions of the sea ice. This study investigates the seasonal variability of IBD during the Arctic freezing season from October to April, across the Distributed Network (DN) scale of the Multidisciplinary drifting Observatory for the Study of Arctic Climate (MOSAiC) expedition. To estimate IBD, we combined sea ice and snow observations from ice mass balance buoys, snow pits, repeated transects, and ice cores, together with high-resolution along-track freeboard data obtained from airborne laser scanning (ALS) and the Ice, Cloud, and land Elevation Satellite-2 (ICESat-2). Assuming hydrostatic equilibrium, IBDs were determined for the level ice components of the MOSAiC ice floes, which consisted predominantly of second-year ice (SYI). Our results revealed significant seasonal variability of the IBD with two main phases during the MOSAiC freezing season at scales of DN (~50 km), L-sites (~25 km), and Main Coring Site (MCS, ~50 m). Throughout the freezing season, the mean IBD estimated at the DN scale ($910 \pm 7$ kg m$^{-3}$) was close to that of the SYI cores at the MCS ($912 \pm 2$ kg m$^{-3}$), highlighting the SYI-dominated regional ice properties. We also identified that sea ice freeboard, along with the ratios of ice freeboard to total freeboard or ice freeboard to thickness, are critical indicators to determine IBD at the scale of tens of kilometers. We have therefore developed parameterizations for IBD that are expected to be applicable throughout the freezing season for the SYI region, which is also the ice type that currently dominates the central Arctic Ocean. The proposed parameterizations have the potential to optimize basin-scale IBD estimation and improve satellite-derived sea ice thickness.



## 1 Introduction

Thickness has been long recognized as a critical variable for Arctic sea ice, reflecting its overall physical state within the context of a warming climate (Sumata et al., 2023). Various techniques have been employed to measure ice thickness and its

spatio-temporal variability, including sea ice mass balance buoys (IMBs, Perovich et al., 2014), airborne or ship-based electromagnetic induction sounding (Haas, 1998; Haas et al., 2009), upward-looking sonars (Rothrock et al., 2008; Hansen et al., 2013), manual in situ measurements (Jezek et al., 1998; Rösel et al., 2018), and satellite altimeters (Landy et al., 2022; Petty et al., 2023). In particular, satellite altimeters have become the primary means of determining ice thickness over the Arctic Ocean thanks to their ability to provide basin-scale and long-term freeboard observations. Assuming hydrostatic

equilibrium, ice thickness can be converted from satellite-derived freeboard measurements, which necessitates prior knowledge of sea ice bulk density (IBD) and snow load (Laxon et al., 2003). However, satellite-derived ice thicknesses contain notable uncertainties due to the limited understanding of the spatial and temporal variability of IBD and snow mass (Zygmuntowska et al., 2014; Kern and Spreen, 2015). Combining the potential biases of these conversion parameters and different interpretations of the radar waveforms, sea ice thickness estimates derived from CryoSat-2 altimetry data have

systematic uncertainties of up to 0.6 m for first-year ice (FYI) and up to 1.2 m for multi-year ice (MYI) (Ricker et al., 2014).

Compared to snow mass, uncertainty in IBD may contribute more substantially to the total ice thickness error, while the actual contribution varies seasonally and geographically (Zygmuntowska et al., 2014; Kern et al., 2015; Kwok and Cunningham, 2015; Landy et al., 2020). Specifically, IBD accounts for 30–35 % of the average absolute uncertainty in the Ice, Cloud, and land Elevation Satellite (ICESat)-based ice thickness, experiencing higher uncertainty in autumn than in

spring, due to lower snow accumulation (Zygmuntowska et al., 2014). For radar altimeter-based ice thickness estimates, variations in IBD settings can result in thickness deviations exceeding 0.5 m (Kern et al., 2015). In particular, a systematic assessment of the uncertainties in the CryoSat-2 derived sea ice thicknesses showed that IBD contributes about 16 % and 12 % to the total uncertainty in the FYI and MYI estimates, respectively (Landy et al., 2020). The considerable heterogeneity in the internal structure of sea ice and the significant seasonal variations in its porosity (Timco and Weeks, 2010), linked to

ice growth and decay processes, underscore the importance of adequately representing the spatial and seasonal variability of IBD for ice thickness retrieval.

Due to its close correlation with ice porosity, IBD is regarded as a pivotal parameter for thermodynamic and mechanical modelling of sea ice (Ono, 1967; Wang et al., 2021), influencing key processes such as sea ice permeability, summer meltwater infiltration, and biogeochemical cycles in the Arctic Ocean (Perovich et al., 2021; Angelopoulos et al., 2022).

Conventional techniques for measuring IBD typically involve mass/volume, submersion, and specific gravity methods, each of which requires sampling, ice block preparation, and measurement (Timco and Frederking, 1996). These methods are valued for their ability to provide direct measurements of the physical properties of sea ice. However, the scarcity of ice core sampling, particularly in winter, limits the spatial representativeness of core-based density measurements (Timco and Frederking, 1996; Hutchings et al., 2015). Recently, Pustogvar and Kulyakhtin (2016) systematically evaluated the



uncertainties of various sea ice density measurement methods, and demonstrated that the hydrostatic weighing method is superior in capturing the natural variations of ice density and in mitigating sampling constraints. Meanwhile, several studies have integrated sea ice and snow pack data from ground-based (Alexandrov et al., 2010) or airborne (Jutila et al., 2022) measurements to estimate IBD using the hydrostatic equilibrium method. However, due to logistical challenges, most of the observations are mainly limited to the late freezing season, which still limits the insight into the seasonal variation of IBD.

Therefore, an accurate representation of the seasonal evolution of IBD still requires the development of robust parameterizations, achievable through the continuous acquisition of observational data throughout the freezing season.

Satellite retrievals of sea ice thickness extensively rely on typical IBD values obtained from limited ground-based measurements (Alexandrov et al., 2010; Quartly et al., 2019; Ji et al., 2021). For the majority of CryoSat-2 ice thickness products, the IBD climatology developed by Alexandrov et al. (2010), referred to as A10, serves as the primary input for the

ice freeboard-to-thickness conversion (Sallila et al., 2019). The FYI density in A10, calculated using the hydrostatic equilibrium method, was determined to be $917 \pm 36$ kg m$^{-3}$. In contrast, given the significant differences in ice density and porosity between exposed and submerged MYI, the MYI density in A10 ($882 \pm 23$ kg m$^{-3}$) was calculated by weighting the ice portions above and below the sea surface. While the reference IBD for the submerged portion was taken from Timco and Frederking (1996), Jutila et al. (2022a) and Shi et al. (2023) have recently argued that the reference IBD of MYI above the

sea surface defined in A10 is too low (only 550 kg m$^{-3}$). Upon adjusting the reference density, both studies found that the recalculated average MYI density was significantly higher than that proposed in A10. Furthermore, several studies have explored the relationship between IBD and other sea ice parameters, aiming to develop IBD parameterizations. For instance, Kovacs (1997) discovered a strong negative relationship between IBD and the square root of sea ice thickness from ice cores collected in the Beaufort Sea; Jutila et al. (2022a) found a negative exponential relationship between IBD and sea ice

freeboard from airborne measurements. However, the applicability and robustness of these parameterizations for estimating IBD remain uncertain, as they are derived from limited field data collected primarily during the late freezing season. Therefore, it is imperative to investigate the seasonal variation of IBD throughout the freezing season to fill this gap.

Recent synergistic observations from the Multidisciplinary drifting Observatory for the Study of Arctic Climate (MOSAiC) expedition and ICESat-2 provided a unique opportunity to estimate IBD throughout the Arctic freezing season. During the

MOSAiC expedition, a variety of IMBs were deployed to measure vertical temperature and heating profiles, from which snow depth and ice thickness values can be derived simultaneously (Lei et al., 2022; Perovich et al., 2023). Ground-based measurements including repeated transects (Itkin et al., 2023), snow pits (Macfarlane et al., 2023a,b), and ice cores (Angelopoulos et al., 2022; Oggier et al., 2023a,b) allowed further insight into snow depth and sea ice thickness, snow bulk density, and core-based IBD data. Moreover, airborne laser scanning (ALS) measurements within the MOSAiC L-site scale

yielded high-resolution along-track total freeboard data (Hutter et al., 2023a,b), uncovering sea ice elevation characteristics across scales of tens of kilometers. From space, ICESat-2 equipped with the Advanced Topographic Laser Altimeter System (ATLAS), also delivered track-based total freeboard measurements with relatively high resolution (Kwok et al., 2019a,b). Therefore, with appropriate adjustments to the sea ice and snow data collected during the MOSAiC expedition, it is





theoretically possible to estimate the IBD of MOSAiC ice floes using the hydrostatic equilibrium method over different
spatial scales.

In this study, we combined sea ice thickness and snow depth data from an IMB array, along-track total freeboard data
from ALS and ICESat-2, and snow bulk density data from snow pits to retrieve IBD (Section 3.1). Ice density data from spot
core sampling and snow depth and sea ice thickness measurements from repeated transects were also included for
comparison and evaluation. We aimed to investigate the seasonal variability of IBD during the Arctic freezing season from
October to April (Section 3.2), a period considered to be the most reliable for retrieving ice thickness from satellites (Ricker
et al., 2014; Petty et al., 2020), and to develop updated parameterizations for IBD to improve its seasonal and spatial
representativeness (Section 3.3). Additionally, we discussed the uncertainties and limitations associated with the IBD
retrieval process (Section 4.1), compared the parameterizations derived from this study with previous findings (Section 4.2),
and investigated the potential impact of the updated parameterizations on satellite-derived sea ice thickness (Section 4.3).

**2 Data and Methods**

**2.1 Data**

In the following sections, we present sea ice and snow data collected from the IMB array, ground-based observations, as well
as airborne and satellite measurements during the MOSAiC expedition from October 2019 to April 2020. To ensure the
robustness of the IBD retrieval, the input data were filtered only from the level ice portions.

**2.1.1 MOSAiC observations**

The MOSAiC expedition was conducted in 2019−2020 and focused on studying the atmosphere, sea ice, ocean, and
ecosystems in the Arctic Ocean (Nicolaus et al., 2022; Rabe et al., 2022; Shupe et al., 2022; Fong et al., 2024). The research
vessel *Polarstern* was anchored to an ice floe, facilitating a comprehensive year-long Lagrangian study along the Transpolar
Drift (see Fig. 1a for the trajectory during the freezing season). A distributed network (DN) of autonomous measurement
platforms, including IMBs, was deployed within a 30−40 km radius of the ship and the Central Observatory (CO) (Rabe et
al., 2024). Measurements were primarily performed on the residual ice that survived the summer of 2019 and transformed
into second-year ice (SYI) by January 2020 (Krumpen et al., 2020; Krumpen et al., 2021).

**IMB data.** Within the MOSAiC DN, the Snow Ice Mass Balance Apparatus (SIMBA, abbreviated as T) and the Seasonal
Ice Mass Balance Buoy (SIMB, abbreviated as I) serve as the principal instruments for automated measurements of snow
and sea ice mass balance (Lei et al., 2022; Perovich et al., 2023). The SIMBAs and SIMBs were deployed on level ice, free
from melt ponds, to ensure the reliability and representativeness of the buoy data. The SIMBAs record the vertical
temperature profile within the snow/ice system and detect thermal changes in the vicinity of thermistors following pulse
heating events. The integration of these measurements enables the determination of ice thickness and snow depth at SIMBA





sites (Provost et al., 2017; Liao et al., 2019). The SIMBs include a thermistor string similar to that of the SIMBAs, as well as ranging sonar and meteorological sensors (Planck et al., 2019). The accuracy of IMB-derived snow depth and sea ice thickness data is estimated to be 0.02 m, also with a vertical resolution of 0.02 m (Lei et al., 2022). In this study, we used daily resampled snow depth and ice thickness data from 12 SIMBAs (Lei et al., 2021) and 3 SIMBs (Perovich et al., 2022) to estimate IBD (Table S1 and Fig. 1b). We selected these buoys from a larger set due to their prolonged operational duration

and greater spatial overlap with the ICESat-2 data. These measurements included 4 buoys deployed in the FYI and 11 in the SYI, with all buoys remaining within 50 km of the CO throughout the study period (Fig. 1c). In particular, exceptionally large ice growth was observed at some sites (e.g., T72 and I2) due to the formation of snow-ice or platelet ice (Fig. S1). Katlein et al. (2020) provided direct evidence for platelet ice formation during MOSAiC based on remotely operated vehicle observations.


**Repeated transect data.** During the MOSAiC expedition, snow depth was systematically measured along repeated transects approximately once or twice per week using a *Magnaprobe* (Sturm and Holmgren, 2018), resulting in an extensive snow depth dataset over the CO floe (Itkin et al., 2021, 2023). The uncertainty of the *Magnaprobe* measurements is expected to be less than 0.01 m in winter (Itkin et al., 2023). Snow measurements were spaced horizontally between 1 and 3 m and varied

with surface roughness and ice type along the transect lines. Most measurements were conducted on the SYI for safety and repeatability considerations. Moreover, the surveys were often conducted in conjunction with electromagnetic induction sounding measurements from the GEM-2 instrument, which measures the total thickness from the snow–air interface to the ice–water interface, providing further information on sea ice thickness (Hendricks et al., 2022; Itkin et al., 2023). The GEM-2 can resolve sea ice layers up to 3 m thick with an accuracy of about 0.1 m, with the main source of measurement

uncertainty coming from the calibration process (Hunkeler et al., 2015; Itkin et al., 2023). The GEM-2 sensors were set to operate at five logarithmically spaced frequencies (from 1.525 to 93.075 kHz) during MOSAiC. We only used the total ice thickness data measured by the 18 kHz channel due to its higher signal-to-noise ratio, which detects the conductivity variation more sensitively (Itkin et al., 2023). To obtain sea ice thickness data, a grid of 1 m horizontal spacing was constructed over the CO floe, from which snow depth and total thickness were derived for each grid cell using nearest

neighbour interpolation (Itkin et al., 2021). Of these transects, the snow and ice thickness of the *Nloop* is more representative of the SYI/MYI, while the *Sloop* would be more typical of the FYI (Itkin et al., 2023). Here, daily mean snow depth and modal sea ice thickness data were compiled from all repeated transects except those marked *Ridge* on a given day, and then compared with buoy measurements to account for the spatial heterogeneity of MOSAiC ice floes.

**Snow pit data.** Numerous snow pit measurements were obtained on the MOSAiC CO floe, including a variety of ice types such as level seasonal ice, level second year ice, ridged areas, and refrozen leads (Macfarlane et al., 2022). Snow density measurements were carried out in 3 cm vertical intervals using a density cutter with a fixed volume of 100 cm$^3$ (Macfarlane et al., 2023a,b). The difference between the average snow density measured by the cutter and the actual value of the





snowpack was estimated to be between 2 and 7 % (Conger and McClung, 2009). In this study, the bulk density of each
snow pit was calculated by weighting the densities of the individual layers according to their respective thicknesses. Only the
snow bulk densities obtained from the level ice sections were used as ancillary data for the IBD retrieval (i.e., excluding
records from the ridge survey area). The snow pit records used contained approximately 12 % refrozen lead, 43 % FYI, and
45 % SYI, according to the original snow pit metadata and MOSAiC ice floe characteristics (Macfarlane et al., 2021;
Nicolaus et al., 2022), which are well representative of the snowpack characteristics of the study area.


**Ice core data.** During the MOSAiC expedition, ice core measurements were conducted specifically at designated coring
sites known as the main first-year (MCS-FYI, Oggier et al. (2023a)) and second-year (MCS-SYI, Oggier et al. (2023b)) ice
coring sites. These core sampling sites were meticulously selected and situated on the MOSAiC CO floe to guarantee
consistent sampling under uniform ice conditions, and were sufficiently representative of the FYI and SYI properties during
MOSAiC. Ice cores were extracted using 9 cm and 7.25 cm diameter corers, with 14 site revisits for FYI and 11 for SYI,
from October 2019 to April 2020. These datasets provided a detailed insight into seasonal variations in ice temperature,
salinity, density, and porosity. Of these, ice core densities were measured in the freezer laboratory using the hydrostatic
weighing method (Pustogvar and Kulyakhtin, 2016) and further interpolated to match the depth of salinity measurements,
providing a continuous density profile throughout the core sample. Pustogvar and Kulyakhtin (2016) found that the
hydrostatic weighing method has a relatively low uncertainty of only 0.2 % with high confidence in capturing natural
variations in ice density. In this study, the bulk density of each core was estimated by weighting the densities of the
individual layers according to their respective thicknesses, and then compared with the results from the DN and L-site scales
(see Section 3.2). To ensure relative accuracy and reasonable comparability of IBD estimates, core-based density records of
rafted ice were excluded as they have significantly higher densities than others.


**ALS data.** Total freeboard data collected by ALS during the MOSAiC expedition were also utilized (Hutter et al., 2023a,b).
The ALS system consists of a near-infrared, line-scanning Riegl VQ-580 airborne laser scanner mounted on a helicopter.
The precision and accuracy of the laser scanner are relatively high, both estimated at 25 mm (Hutter et al., 2023b). The
airborne flight path ranged from local scale grids surrounding the CO floe to extensive regional scale transects, extending up
to several tens of kilometers from the CO (L-site scale; Fig. 3a). The ALS total freeboard data were derived from a high-
resolution dataset, originally consisting of geolocated sea ice or snow surface elevation point clouds (Jutila et al., 2023).
These point cloud data were segmented into 30 second intervals of which atmospheric backscatter and other potential errors
were filtered out. These data were then processed to calculate total freeboard using an automatic open water detection
scheme and projected onto a regular grid with a high spatial resolution of 0.5 m. In this study, only the ALS total freeboard
data collected at the L-site scale (12 records in total) were utilized, aimed at integrating them with ICESat-2 data and
deriving the IBD within the same region (~25 km radius from CO). A detailed timeline of the airborne/satellite data used at
different scales is shown in Fig. 3b. Notably, the original dataset did not provide the total freeboard variable for all survey





flights due to the reduced accuracy of the Global Navigation Satellite System at high latitudes (approximately above 85° N)
and the lack of open water points in mid-winter pack ice. Therefore, we estimated ALS total freeboard following the
methods outlined in Hutter et al. (2023b). Despite the expected high uncertainty in the estimated total freeboard data, we are
confident in the modal values identified from the ALS total freeboard distribution over tens of kilometers, as reflected by the
relatively small differences between ALS and ICESat-2 modal values over the same range (i.e., cross-validation of data
accuracy; Fig. 3a).

**Figure 1. (a)** Drift trajectory of the MOSAiC CO (black line), with the starting site marked by a red pentagram (4 October 2019), and the
coloured bands around the trajectory indicate the monthly segments of the drift. The red circle marks the measurement limit of ICESat-2
(88° N). **(b)** Locations of the 12 SIMBAs and 3 SIMBs in the vicinity of the MOSAiC CO on 20 November 2019. The buoy sites marked
by circles with black edge color denote deployments in SYI, while circles with orange edge color mark sites on FYI. **(c)** Variations in the
distance of different buoys from the MOSAiC CO during the study period.



### 2.1.2 ICESat-2 data

ICESat-2 is equipped with the ATLAS instrument, which uses a low pulse energy laser configured into six beams; these beams are grouped into three pairs, with each pair consisting of one strong and one weak beam, facilitating precise surface mapping (Kwok et al., 2019a,b). The strong beams have approximately four times the pulse energy of the weak beams, thereby achieving higher spatial resolution. The vertical accuracy of ICESat-2 elevation measurements ranges from approximately 7 to 10 cm, with a ground footprint diameter of about 17 m (Markus et al., 2017; Kwok et al., 2019a). This study utilized the recently released ATL10 freeboard dataset (L3A, version 6; Kwok et al., 2023) from the National Snow and Ice Data Center (NSIDC) to retrieve IBDs at the DN and L-site scales, providing along-track total freeboard measurements from both strong and weak beams. The uncertainty of the ICESat-2 total freeboard data is estimated to be 2–4 cm (Kwok et al., 2019a,b). For each ground track, data from the strong beams were utilized exclusively owing to their enhanced along-track resolution in segment length (e.g., Petty et al. (2020) and Koo et al. (2021)).

### 2.1.3 Ancillary data

Ancillary data included airborne multi-sensor data from the Alfred Wegener Institute's IceBird (AWI IceBird) campaigns, collected in April 2017 and April 2019, and the AWI CryoSat-2 sea ice thickness product for the period October 2023 to April 2024, accompanied by gridded sea ice age data from NSIDC for the same period.

**AWI IceBird airborne data.** In April 2017 and April 2019, airborne data were collected by AWI over a total distance of 3410 km in the western Arctic Ocean, offering detailed insights into the distribution of snow depth and sea ice thickness, characterized by different ice surface roughness and age (Jutila et al., 2022a). The primary instruments onboard included an electromagnetic induction sounding instrument (EM-Bird; Haas et al., 2009), an ALS, and an ultra-wideband microwave snow radar (Jutila et al., 2022b), each designed to measure total thickness (sea ice thickness plus snow depth), total freeboard, and snow depth, respectively. The nominal measurement interval in the combined data product range from 5 to 6 m, with a footprint diameter of 40 m. Based on these measurements, Jutila et al. (2022a) estimated the IBD for the airborne trajectories using the hydrostatic equilibrium method and developed the first sea ice freeboard-dependent parameterization for IBD. Here, the IceBird-derived total freeboard and IBD data from Jutila et al. (2024a,b) were used to validate our modal approach (Text S1) and to compare with the IBD results during MOSAiC (Section 3.2), respectively. To ensure data quality, total freeboard data with negative values and IBD data with uncertainties greater than 50 kg m$^{-3}$ were excluded.

**AWI CryoSat-2 sea ice thickness product.** The European Space Agency's CryoSat-2 satellite mission incorporates the Ku-band Synthetic Aperture Interferometric Radar Altimeter (SIRAL), enabling long-term records of ice elevation and freeboard (Drinkwater et al., 2004; Wingham et al., 2006). To investigate the impact of the updated IBD parameterizations on the sea ice thickness retrievals (see details in Section 4.3), we utilized monthly gridded sea ice data from the AWI *CryoSat-2 Sea Ice*



*Thickness Product* (v2p6) (see details in Hendricks and Paul (2023)), spanning the period from October 2023 to April 2024. Sea ice parameters from the product were projected onto the Equal-Area Scalable Earth (EASE)-Grid 2.0 at a spatial resolution of 25 km × 25 km. To retrieve sea ice thickness, this product also integrates pan-Arctic IBD estimates that are

estimated by weighting the relative proportions of FYI and MYI based on the A10 climatology.

**Sea ice age product.** To identify the pan-Arctic SYI field, we used weekly sea ice age quicklook data from NSIDC (Tschudi et al., 2019; Tschudi et al., 2020) with a spatial resolution of 12.5 km × 12.5 km. Together with the annually updated final product, the data coverage extends from January 1984 to the present, providing a comprehensive sea ice age field spanning

four decades. Sea ice age is determined using a Lagrangian tracking method and is delineated by the number of summer melt seasons it has undergone (Tschudi et al., 2020). For data quality control, grid cells with ice concentrations below 15 % were considered as open water. To match the AWI CryoSat-2 data used in this study, we re-gridded the sea ice age data using an inverse distance weighting interpolation scheme and processed them into monthly averages.

## 2.2 Methods

Figure 2 illustrates the detailed steps and datasets involved in the IBD retrieval, which are elaborated in the subsequent sections. The key process involves matching along-track measurements from airborne and satellite data with observations from the IMB array. Specifically, by integrating and pre-processing all datasets collected during MOSAiC, we aim to characterize the IBD at three spatial scales: DN (within a 50 km radius of CO), L-site (within a 25 km radius of CO), and MCS (at the CO floe).

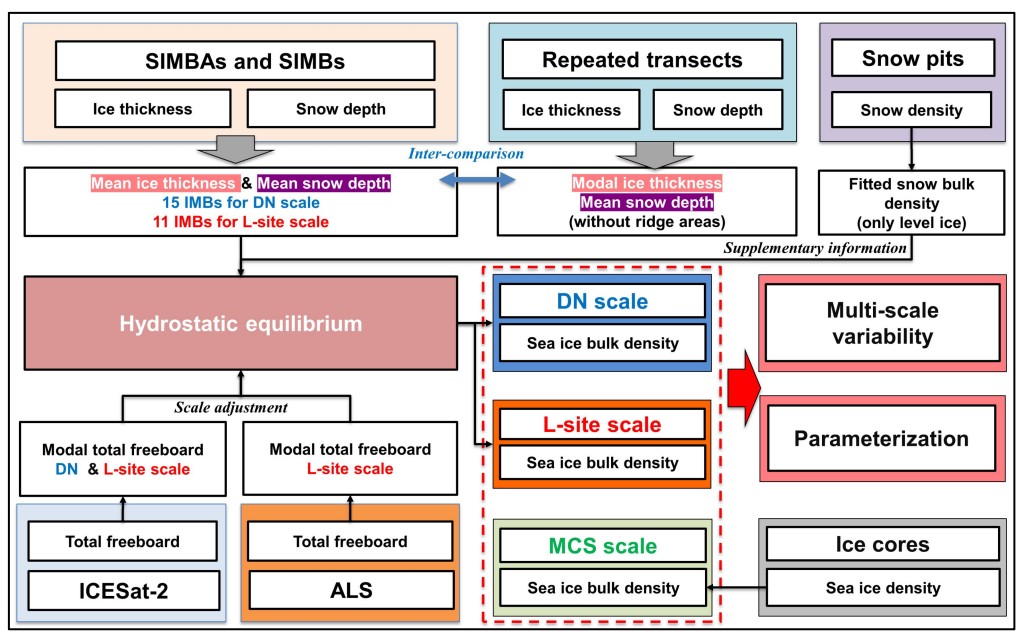


**Figure 2.** Flowchart of the IBD retrieval process.



### 2.2.1 Modal total freeboard derived from ALS and ICESat-2 measurements

To estimate the IBD accurately, it is crucial to obtain the mean total freeboard that is representative of the level ice portions to ensure compatibility with the MOSAiC IMB array, which was deployed exclusively on level ice. This objective was achieved through a simple modal approach, similar to the methods employed by Ricker et al. (2015) and Koo et al. (2021). The procedure involves: 1) identifying valid ICESat-2 and ALS along-track total freeboards within a designated search radius from the CO (25 and 50 km for the L-site and the DN scales, respectively), 2) determining the total freeboard distributions with a bin size of 0.001 m, and 3) ascertaining the modal total freeboard, which is derived from the mean of the five highest frequency peaks in the total freeboard distribution, with the standard deviation quantifying the uncertainty (Type I) introduced by these peaks. Extensive airborne and satellite measurements support a bin size of 0.001 m for the freeboard distribution, but we ultimately present the data in 0.01 m units due to measurement uncertainties. The same method was used to calculate the modal sea ice thickness of the transects, but the bin size was set to 0.05 m due to the relatively small number of thickness samples. The feasibility of using modal values from the regional freeboard/thickness distribution to accurately represent the mean freeboard/thickness of level ice sections is demonstrated in Text S1.

To ensure spatial representativeness, modal total freeboards for ICESat-2 were only calculated when the number of total freeboard segments (1 segment corresponds to 150 ICESat-2 photon aggregates) exceeded 20,000 for DN and 10,000 for L-site scales. During the study period, four records simultaneously captured the total freeboard characteristics of ALS at the L-site scale and ICESat-2 at both L-site and DN scales (Fig. 3a). The mean difference between the modal total freeboards of ALS and ICESat-2 from the same region (i.e., L-site scale) was 0.01 m (or ~5 %), which was attributed to differences in spatial coverage, as well as measurement sensors and footprints. This mean difference was also considered to represent the uncertainty of the modal total freeboard, stemming from the measurement methods and the spatial heterogeneity of the MOSAiC ice floes (Type II). Only ALS results were retained during overlapping periods due to their higher resolution and more structured measurement coverage (triangular network) compared to ICESat-2. Notably, brief data gaps occurred from February to early March, when the MOSAiC ice floes drifted beyond the spatial coverage of ICESat-2 (north of 88° N; Fig. 1a and Fig. 3b), and in April, when there was a lack of available ICESat-2 tracks around the MOSAiC CO due to satellite revisit cycle constraints (Fig. 3b). Crucially, we have applied a systematic increase of ~0.07 m to both scales of modal total freeboard to align with buoy site measurements (see Text S2 for details). This helps to ensure compatibility of the non-overlapping datasets used to retrieve IBD via the hydrostatic equilibrium method, and mitigates the intrinsic differences between the observational methods (Fig. S4). We also suggest that the snow depth difference between satellite/airborne measurements and buoy array sites, mainly caused by snow redistribution, accounts for most of the adjustment term (see Section 3.1).





**Figure 3. (a)** Four daily records illustrating the total freeboard distribution from ALS at L-site scale and ICESat-2 (IS2) at both L-site and DN scales during the study period. The symbols used for each histogram panel include: **PDF** for probability density function (note the varying scale on the vertical axes), **Mode** for modal total freeboard, **Unc.** for uncertainty introduced by the quasi-peak region, and **Num** for number of samples. **(b)** Timeline of ALS and IS2 data acquisition throughout the freezing season, with yellow bands indicating the four dates shown in panel **(a)** and green bands marking two periods of unavailable IS2 tracks around the MOSAiC CO due to limitations in measurement latitude (88° N) and satellite revisit cycle, respectively.



### 2.2.2 Retrieval of IBD

The hydrostatic equilibrium of the snow-covered sea ice is maintained by combining ice thickness, snow depth, and snow bulk density data obtained from IMBs and snow pits, along with the adjusted modal total freeboard from ALS and ICESat-2 (Alexandrov et al., 2010; Jutila et al., 2022a):

$$\rho_i h_i + \rho_s h_s = \rho_w (h_i - h_f + h_s),$$   (1)

where $\rho_i$ is the IBD, $\rho_w$ and $\rho_s$ represent the seawater and snow bulk densities, respectively; $h_i$, $h_f$, and $h_s$ are the sea ice thickness, total freeboard, and snow depth, respectively. Thus, the IBD can be estimated as:

$$\rho_i = \rho_w \left(1 - \frac{h_f}{h_i} + \frac{h_s}{h_i}\right) - \rho_s \frac{h_s}{h_i}.$$   (2)

In this study, $h_i$ and $h_s$ were both obtained from the SIMBAs and SIMBs; $h_f$ was the adjusted modal total freeboard;
$\rho_w$ was set to 1024 kg m$^{-3}$ following Wadhams et al. (1992), and $\rho_s$ was obtained from the snow pit measurements (Macfarlane et al., 2022), see details in Fig. 4d. This method provides an estimate of the level ice density mixed with SYI (a large fraction) and FYI (a small fraction) within the MOSAiC DN. Also, our estimation of IBD focused on the Arctic freezing season of relatively young sea ice less than 16 months old (Krumpen et al., 2020).

### 2.2.3 Uncertainty in IBD retrieval

The uncertainty in the retrieved IBD ($\sigma_{\rho_i}$) was determined using the Gaussian error propagation method, assuming that the uncertainties of the individual variables in Eq. (2) are independent (Taylor, 1997). It is represented as a combination of the partial derivatives of these variables and their corresponding individual error terms:

$$\sigma_{\rho_i} = \sqrt{(1 - \frac{h_f}{h_i} + \frac{h_s}{h_i})^2 \times \sigma_{\rho_w}^2 + (-\frac{h_s}{h_i})^2 \times \sigma_{\rho_s}^2 + (\frac{\rho_w h_f - \rho_w h_s + \rho_s h_s}{h_i^2})^2 \times \sigma_{h_i}^2 + (\frac{\rho_w - \rho_s}{h_i})^2 \times \sigma_{h_s}^2 + (-\frac{\rho_w}{h_i})^2 \times \sigma_{h_f}^2}.$$   (3)

where $\sigma_{\rho_w}$ and $\sigma_{\rho_s}$ are the seawater and snow bulk density uncertainties, and $\sigma_{h_i}$, $\sigma_{h_s}$, and $\sigma_{h_f}$ are the ice thickness,
snow depth, and total freeboard uncertainties.

In this study, $\sigma_{\rho_w}$ was set to 0.5 kg m$^{-3}$ according to Wadhams et al. (1992), and $\sigma_{\rho_s}$ was set to half the 95 % confidence interval of the fitted snow bulk density (see red shaded band in Fig. 4d), with a mean value (± 1 standard deviation) of 20 ± 5 kg m$^{-3}$ over the study period. Given the inherent resolution limitations of the IMBs (0.02 m) and differences in processing methods (Liao et al., 2019; Koo et al., 2021; Lei et al., 2022), $\sigma_{h_i}$ and $\sigma_{h_s}$ were both set to 0.04 m. $\sigma_{h_f}$ was regarded as
the modal total freeboard uncertainty (sum of Type I and Type II uncertainties). Throughout the freezing season, the mean uncertainty (± 1 standard deviation) for the modal total freeboard at the L-site and DN scales was estimated to be 0.04 ± 0.01 and 0.02 ± 0.01 m, respectively. It should be noted that the IBD uncertainty ($\sigma_{\rho_i}$) calculated here represents the total random uncertainty. Given the expected significant impact of episodic meteorological events on the regional IBD estimates,





we also performed a quality control of the retrieved IBDs, including 1) eliminating records with more than twice the standard deviation of the original IBDs retrieved over the study period, and 2) eliminating records with a deviation of more than 30 kg m$^{-3}$ from the previous value within a seven-day window. These data judged to be anomalous accounted for about 10 % of valid IBD records.

### 2.2.4 Parameterization for IBD

This study evaluated the potential of several univariate and bivariate sea ice parameters to serve as indicators for the
parameterization of IBD. The univariate sea ice parameters included sea ice thickness, total freeboard, sea ice draft, and sea ice freeboard. Meanwhile, the bivariate parameters included total thickness (ice thickness + snow depth), and the ratios of ice freeboard to total freeboard, ice freeboard to thickness, and snow depth to ice thickness – parameters that are potentially indicative of sea ice stratification, porosity, and permeability. The relationships between IBD and each sea ice parameter were investigated using regression analyses. The training dataset for the regression models spanned the main ice-growth
season from October to April and included measurements from all 15 deployed buoys within the MOSAiC DN, ensuring comprehensive spatial and temporal representativeness.

## 3 Results

### 3.1 Integrated observations of sea ice and snow during MOSAiC

Figure 4 depicts the seasonal variation in sea ice thickness, snow depth, total freeboard (i.e., adjusted modal total freeboard),
and snow bulk density during the MOSAiC freezing season. Note that the buoy deployment sites included both SYI and FYI for level ice only, successfully recording thermodynamically driven ice growth (Koo et al., 2021). In this study, at least 10 buoys at the DN scale and 7 buoys at the L-site scale were consistently maintained daily, and their mean sea ice thickness and snow depth served as the main inputs to the IBD retrieval. The snow depths of the transects were only used to assess the sufficiency of the buoy array to capture the regional snow accumulation process. The sea ice thickness of the transects was
included in the comparisons with the ice cores and the buoy array to investigate the spatial heterogeneity of the MOSAiC ice floes. Recall that spatial scale adjustments were made to the total freeboard data from the broader airborne and satellite observations to match the buoy array sites.

The mean ice thicknesses recorded by the buoys at the DN and L-site scales were 1.41 ± 0.32 and 1.30 ± 0.33 m, respectively (Fig. 4a and Table 1), whereas the average values for individual buoys ranged from ~0.6 to 2 m with adequate
regional representativeness (Fig. S1). The relative differences between the DN and L-site buoy averages are closely related to the FYI/SYI ratios at their deployment sites. The ice thicknesses of the MCS-SYI closely matched those of the two regional scales, whereas the MCS-FYI was consistently lower, mainly due to the predominance of SYI in the MOSAiC ice floes at the DN scale (Fig. 4a). In addition, seasonal variations in ice thickness derived from spot ice cores, repeated transects, and regional buoy averaging all demonstrated similar increasing trends over the study period (Table 1). Notably, buoy-



derived thicknesses at the DN and L-site scales were generally thicker than the transect observations conducted on the CO floe; however, considering only the two buoys deployed near the CO (T62 and T66) there was strong agreement with the transect observations. For the initial buoy records in the early stages (late October to mid-November), the average thicknesses for SYI and FYI were 0.93 ± 0.22 and 0.45 ± 0.07 m, respectively, which closely corresponded to the MCS-SYI and MCS-FYI estimates for the same period (0.81 ± 0.06 and 0.41 ± 0.03 m, respectively), confirming the effectiveness of

using ice cores as initial ice freeboard references for spatial scale adjustments (see Text S2 for details). Collectively, comparisons of the different ice thicknesses observed during MOSAiC revealed inherent discrepancies in ice thickness extending from the CO floe to the DN scale, further suggesting that there may be significant spatial heterogeneity among MOSAiC ice floes. However, the relative differences between these measurements are also related to deployment strategy, measurement coverage, and instrumental uncertainties.

In terms of snow depth, the transects recorded the thickest snow layer (0.24 m on average), attributed to the proximity of level ice segments to deformed ice that promotes snow accumulation (Itkin et al., 2023) (Fig. 4b). In contrast, ice cores documented thinner snow depths, possibly due to specific ice conditions. Snow depths at the DN and L-site scales obtained from the buoys were closely matched, averaging ~0.19 m over the study period, but varied considerably between buoy sites (Fig. S1). In particular, the variance of snow depth between buoy sites increased significantly after the strong snow drift

period on 24–25 February 2020 (Wagner et al., 2022; Fig. S1). Throughout the freezing season, the regional buoy averages, as well as the transects and MCS-FYI, all exhibited significant trends of increasing snow depth, with the buoys and transects being well aligned and having higher increasing rates than the MCS-FYI. It is important to emphasize that regional variations in snow depth require a sufficient range of measurements to be accurately represented. Therefore, comparisons with transect data, which have a more comprehensive measurement coverage (Itkin et al., 2023), show that buoy array

measurements are sufficiently representative to capture snow accumulation on level ice over a wider area. Furthermore, although ice thickness measurements from spot ice cores and larger scale transects were similar on the same CO floe, snow depth varied considerably. This implies that the spatial heterogeneity of snow cover may be more pronounced compared to ice thickness during MOSAiC, partially explaining the mismatch between the buoy-derived snow depth and modal total freeboard from satellite/airborne measurements (Text S2).

Measurements from airborne, satellite, and ice core data consistently demonstrated an increasing trend in total freeboard with ice growth, and the magnitudes of the trends were relatively close, ranging from ~0.021 to 0.029 m per month (Fig. 4c and Table 1). The total freeboard of level ice over the study period for the DN scale (adjusted to buoy array sites), the L-site scale (adjusted to buoy array sites), MCS-SYI, and MCS-FYI was 0.30, 0.29, 0.24, and 0.17 m, respectively. Episodic storm events, which are often associated with pack ice deformation, had no discernible impact on the changes in the total freeboard of level ice. The observed smooth and pronounced linear trend in this parameter indicates that the primary factors

influencing its variation are thermodynamic sea ice growth and snow accumulation.

Significant temporal variations in snow bulk density were also observed during MOSAiC, attributable to new snowfall and snow metamorphism (Macfarlane et al., 2023a,b) (Fig. 4d). A statistically significant increasing trend in snow bulk



density of 9 kg m$^{-3}$ per month was identified, potentially indicating snow stratification and compaction processes (Wagner et

al., 2022). To extend the temporal coverage of snow bulk density data, fitted estimates were used as ancillary information for the IBD retrieval (Fig. 2). However, it should be clarified that the impact of snow bulk density on the IBD retrieval is very small, with the associated uncertainties contributing less than 5 % to the total uncertainty of the retrieved IBDs (see Section 4.1). The significant seasonal variations of these sea ice- and snow-related parameters provide critical data support for establishing IBD parameterizations that rely on these parameters.

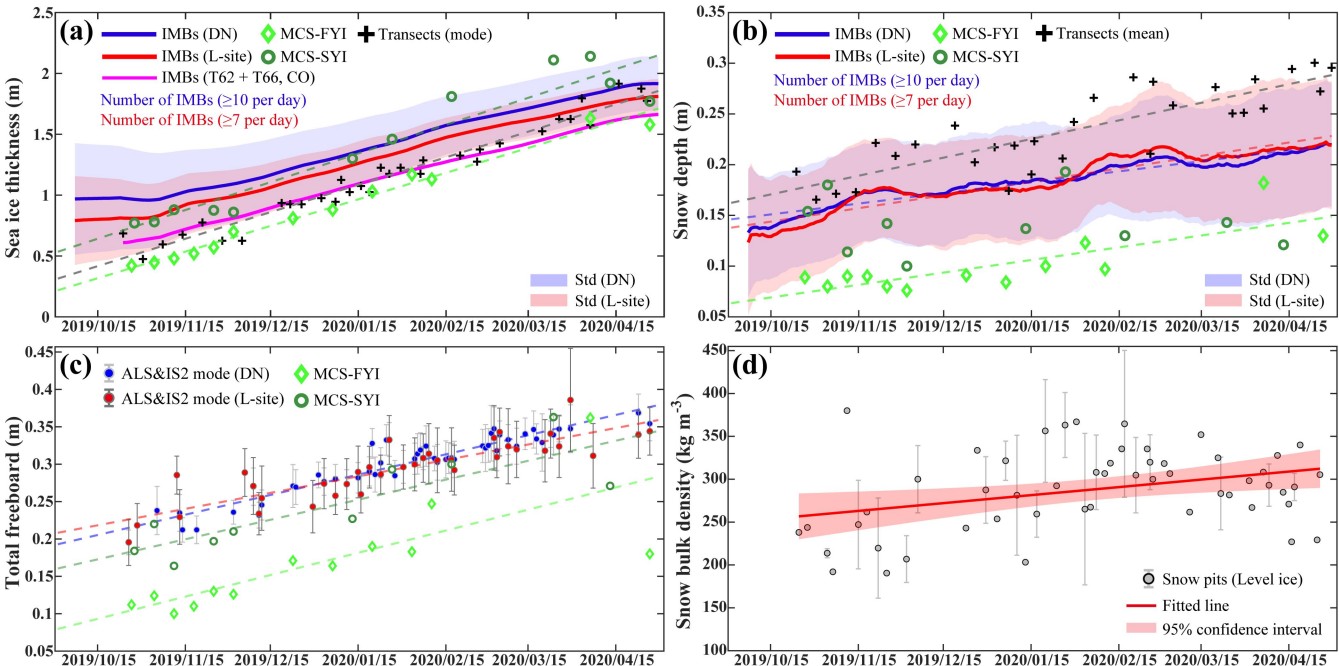


**Figure 4.** Seasonal variations of sea ice and snow parameters from level ice components during the MOSAiC freezing season. **(a)** Sea ice thickness measurements obtained from buoys (blue, red, and magenta lines for DN, L-site, and CO scales, respectively), transects (black crosses), and ice cores (green symbols). **(b)** Snow depth measurements recorded from buoys (blue and red lines for DN and L-site scales, respectively), transects (black crosses), and ice cores (green symbols). **(c)** Total freeboard measurements derived from ALS & ICESat-2

(blue and red dots for DN and L-site scales, respectively) and ice cores (green symbols). The error bars represent the uncertainties of the modal total freeboard. **(d)** Snow bulk density measurements ascertained from snow pits (grey dots), with error bars denoting one standard deviation from all daily estimates. The red line indicates a linear fit and the shaded band labels the 95 % confidence interval. **Note:** The dashed lines in panels **(a)**, **(b)**, and **(c)** represent seasonal linear trends, with all statistical *P*-values less than 0.01.






**Table 1.** Statistical characteristics of sea ice and snow parameters, including means (Mean), standard deviations (Std), and seasonal linear trends (Trend, $P < 0.01$) during the MOSAiC freezing season. The asterisk (*) denotes snow bulk density results obtained from snow pits over the CO floe.

| Data (scale) | Sea ice thickness (m) | | Snow depth (m) | | Total freeboard (m) | |
|---|---|---|---|---|---|---|
| | Mean ± Std (m) | Trend (m month$^{-1}$) | Mean ± Std (m) | Trend (m month$^{-1}$) | Mean ± Std (m) | Trend (m month$^{-1}$) |
| IMBs (DN) | 1.41 ± 0.32 | 0.16 | 0.19 ± 0.02 | 0.010 | n/a | n/a |
| IMBs (L-site) | 1.30 ± 0.33 | 0.17 | 0.19 ± 0.03 | 0.013 | n/a | n/a |
| ALS&IS2 (DN) | n/a | n/a | n/a | n/a | 0.30 ± 0.04 | 0.026 |
| ALS&IS2 (L-site) | n/a | n/a | n/a | n/a | 0.29 ± 0.04 | 0.021 |
| IMBs (T62+T66, CO) | 1.17 ± 0.31 | 0.18 | n/a | n/a | n/a | n/a |
| Transects (CO) | 1.17 ± 0.40 | 0.22 | 0.24 ± 0.04 | 0.018 | n/a | n/a |
| Ice cores (MCS-FYI) | 0.87 ± 0.41 | 0.21 | 0.10 ± 0.03 | 0.012 | 0.17 ± 0.07 | 0.029 |
| Ice cores (MCS-SYI) | 1.39 ± 0.54 | 0.23 | 0.14 ± 0.03 | Not significant | 0.24 ± 0.06 | 0.026 |

**\*Snow bulk density:** Snow pits [Mean ± Std] = 288 ± 47 kg m$^{-3}$; Snow pits [Trend] = 9 kg m$^{-3}$ month$^{-1}$.

n/a: not applicable

## 3.2 Seasonal variability of IBD during MOSAiC

Figure 5 depicts the scale-related seasonal variation of IBD during MOSAiC. The regional IBD estimates represent the ice properties at the buoy deployment sites. Overall, the variations in regional mean IBD were broadly consistent with the spot
ice core results, but there were significant relative differences in magnitude (Fig. 5a). The mean IBD at the DN and L-site scales for the entire study period was 910 and 908 kg m$^{-3}$, respectively. In this context, the average IBD at the DN scale closely matched that of the MCS-SYI (912 kg m$^{-3}$), highlighting the predominance of the SYI type within the DN spatial extent; whereas the L-site estimate mainly fell between that of the MCS-FYI (905 kg m$^{-3}$) and the MCS-SYI, possibly related to the sea ice characteristics of the buoy deployment sites at this scale. In terms of the range of IBD estimates, the
MCS-SYI remained relatively stable throughout the freezing season, associated with lower ice porosity and brine content as sea ice ages. In contrast, both the regional scale estimates and the MCS-FYI were more variable. Note also that the characteristics of the IBD variability at the DN and L-site scales were generally consistent with those of the MCS-FYI, which, combined with the relative stability of the MCS-SYI, further demonstrates that the FYI component dominates the IBD variability of the MOSAiC ice floes. The scale-related seasonal variation of IBD during MOSAiC can be divided into
two main phases from late October to late April by combining results from ice cores and two regional estimates (Fig. 5a). Nevertheless, within the two distinct phases, there was still considerable sub-weekly variability in IBD estimates at the DN and L-site scales, highlighting the complexity of larger scale IBD changes.

From late October to mid-December (phase 1), IBDs of the MCS-FYI and regional scales increased significantly from ~890 to 910 kg m$^{-3}$ (Fig. 5a), accompanied by a rapid thickening of the thinner first-year level ice during this period and a
relatively high density variability within the ice layer due to the newly formed FYI and SYI fractions at the ice bottom (Figs. S1–S2). These rapidly formed new ice layers, driven mainly by thermodynamic processes, may contain more brine and less air pockets than the older ice fractions (Petrich and Eicken, 2017), thus increasing the overall IBD. In addition, the new ice




layer will gradually transition from a loose granular structure to a dense columnar structure, resulting in a more compact layer (Oggier and Eicken, 2022). However, there is evidence that a warm air intrusion event occurred in mid-November
during MOSAiC (Angelopoulos et al., 2022), which would have increased brine discharge accordingly. Such a process may have partially offset the increase in IBD that followed the newly formed ice layer, contributing to the decrease in the rate of IBD increase (Fig. 5a). From mid-December to late April (phase 2), regional estimates along with MCS-FYI and MCS-SYI all demonstrated relatively stable IBDs over the major ice growth period of the SYI, corresponding to relatively low ice porosity with peak IBD values nearing 910 kg m$^{-3}$ (Fig. 5a, right panel). Meanwhile, the estimated standard deviation of the
IBD from MCS-FYI gradually aligned with the results from MCS-SYI (Fig. S2), suggesting that sea ice aging leads to homogenization of internal ice properties.

Notably, no clear evidence of desalination was observed during the early study period, which is typically associated with a significant decrease in IBD, as the initially salt-saturated sea ice pores were gradually filled with air/seawater content (Petrich and Eicken, 2017). This suggests that such a process was likely to have occurred during early autumn (e.g.
Angelopoulos et al., 2022), explaining the relatively low IBD observed at the beginning of our study period. The strong agreement between the regional scale IBD results and the ice core data provided sufficient confidence to elucidate and interpret the scale-related seasonal variations in IBD of the MOSAiC ice floes. This also supported the spatial scale adjustments applied to the modal total freeboard to match the buoy array sites. However, we recognized that the uncertainty associated with the IBDs at the DN and L-site scales was generally higher in October and November compared to later
months of the freezing season (Fig. 5b). Nevertheless, our analysis clearly indicated that the scale-related seasonal variation in IBD across MOSAiC ice floes is significant and needs to be considered in the IBD parameterization.

Figure 6 illustrates the comparison of the MOSAiC-derived IBD estimates with historical records. For the comparative analysis with historical data, the MOSAiC-derived IBD estimates including both ice core and regional scale results, were converted to monthly averages. In addition, the IceBird-derived IBD (2017 and 2019) was recalculated to include only level
ice consisting of FYI and SYI, resulting in an average of 917 ± 36 kg m$^{-3}$ (Jutila et al., 2022a). Individual IceBird-derived IBD estimates of FYI (921 ± 35 kg m$^{-3}$), SYI (899 ± 36 kg m$^{-3}$), and MYI (897 ± 29 kg m$^{-3}$) for level ice were also presented to show the variation in IBD for different ice types and thicknesses. The recalculated IBDs for the Arctic Sever Expedition (from 1980 to 1989) reported by Shi et al. (2023) were identified as modified A10 (mA10). Overall, Figure 6 shows good agreement of the MOSAiC results with historical data, also including scattered on-site measurements from 2000
to 2015 (Ji et al., 2021).

More specifically, the variation of monthly regional scale IBD estimates during the MOSAiC expedition were in good agreement with on-site measurements for all ice types from January to April, with a mean difference (MOSAiC minus on-site) and root-mean-square difference of 8 and 12 kg m$^{-3}$, respectively. In March and April, the MOSAiC regional estimates were positioned between the IBDs of FYI and MYI for mA10 and IceBird during the same period. In particular, the
MOSAiC regional IBD in April was almost identical to the IBD derived from IceBird (FYI + SYI), with a difference of only 2 kg m$^{-3}$, although the proportion of SYI measured by IceBird is relatively small (only 7% in 2019 and none in 2017). We





propose that the consistency of the FYI-biased IceBird results (FYI + SYI) with the SYI-dominated MOSAiC ice floes may be due to the higher proportion of relatively thicker FYI in the IceBird measurements, as well as the entirely different geographical coverage of the two measurements (western vs. central Arctic Ocean). Notably, both the MOSAiC-derived IBD

and other historical records of MYI density consistently showed higher values than the A10-MYI (up to ~30 kg m$^{-3}$), indicating that the reference density used for the A10-MYI was inappropriate.

Subsequently, we referred to the MOSAiC ice cores for IBD comparisons with historically recorded FYI and SYI estimates. We found that the ice cores during MOSAiC had lower FYI densities (MCS-FYI) compared to A10, mA10, and IceBird, but all were within one standard deviation of these results. Moreover, MCS-SYI had a higher IBD than the IceBird-

derived SYI density in April, with a difference of 11 kg m$^{-3}$. These variations are intimately linked to the ice thickness/freeboard extent, the study area, and the measurement techniques and footprints employed by various observation campaigns. Based on the MOSAiC results, we highlight the significant uncertainty associated with using fixed or typical IBD values to retrieve ice thickness from space, as these values do not account for the substantial scale-related temporal variability of IBD. We also call for an update of the MYI density of A10 used in current sea ice thickness retrievals, as

several lines of comparative evidence consistently suggest a significant underestimation.

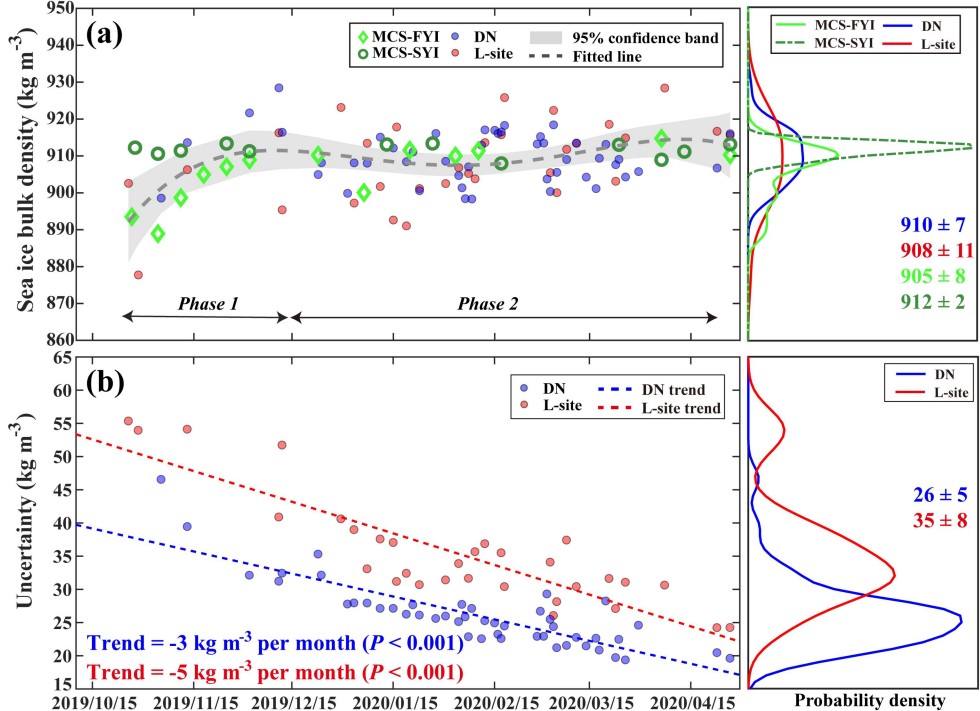

**Figure 5.** Seasonal variability of IBD during the MOSAiC freezing season. The right subplot of each panel shows the corresponding probability density distribution, with labelled values showing the mean ± standard deviation. **(a)** IBD estimates at the DN (blue dots), L-site (red dots) and MCS (green symbols) scales, respectively. The grey dashed line represents the quartic polynomial fit ($P < 0.01$) for the

DN and L-site IBDs and the shaded band indicates the 95 % confidence interval. The arrows mark two distinct stages of IBD evolution. **(b)** Uncertainty of IBD estimates for the DN (blue dots) and L-site (red dots) scales, with dashed lines showing their linear trends ($P < 0.001$).





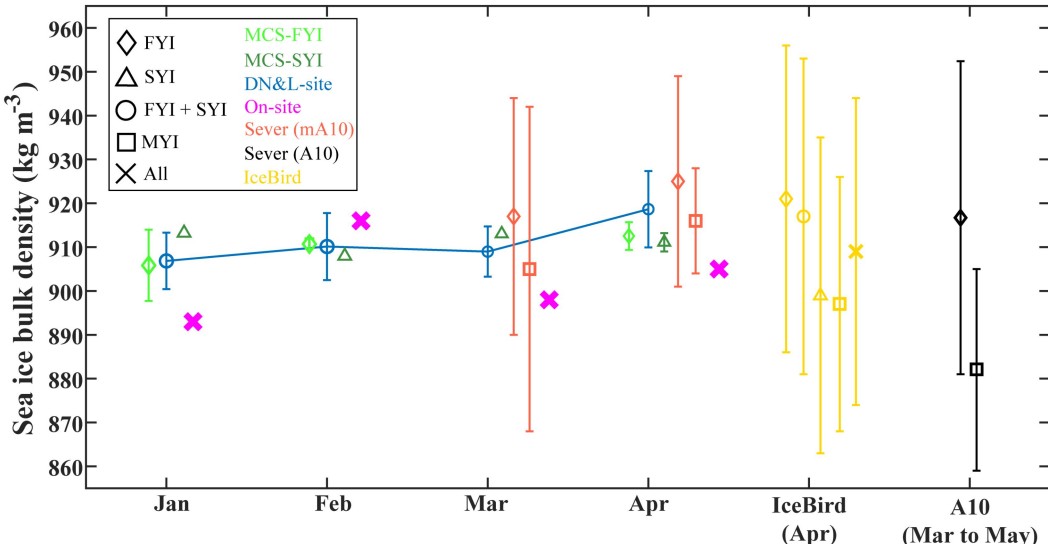

**Figure 6.** Comparison of IBD estimates during MOSAiC with historical records, including the A10 climatology from the Arctic Sever expedition from 1980 to 1989 (Alexandrov et al., 2010), the modified A10 climatology (mA10) for March and April (Shi et al., 2023), scattered on-site measurements from 2000 to 2015 (Ji et al., 2021), and AWI IceBird airborne measurements in 2017 and 2019 (Jutila et al., 2022a). Error bars represent one standard deviation of the respective IBD data. **Note:** The IBD estimates from historical measurements correspond to specific months, independent of years.

## 3.3 Parameterization of IBD

The results of analyzing the relationships between IBD and different sea ice parameters are illustrated in Fig. 7 and show significant variations across different scales and ice types. At the DN scale, all sea ice parameters showed negative correlations with IBD, excluding the ratio of snow depth to sea ice thickness. The MCS-SYI demonstrated the same statistical relationships as the DN scale, but none were statistically significant, highlighting the challenges of parameterizing IBD from sparse fixed-point measurements. However, the relative stability of the IBD from MCS-SYI and its limited sample size (only 10) may partially explain the observed lack of statistical significance. MCS-FYI showed an opposite statistical relationship compared to the DN scale and MCS-SYI estimates, emphasizing that FYI is not the dominant sea ice type within the MOSAiC DN range. At the L-site scale, the relationships between different sea ice parameters and IBD showed similar characteristics to both the DN and MCS-FYI results, indicating a more complex sea ice component in this region, consistent with the analysis in Fig. 5a. Notably, at the regional scale, only the sea ice freeboard among the univariate parameters attained consistency between the DN and L-site scales, although the L-site result was not statistically significant. For the bivariate parameters of ice freeboard-to-total freeboard ratio and ice freeboard-to-thickness ratio agreed well at the DN and L-site scales with relatively high correlation coefficients.



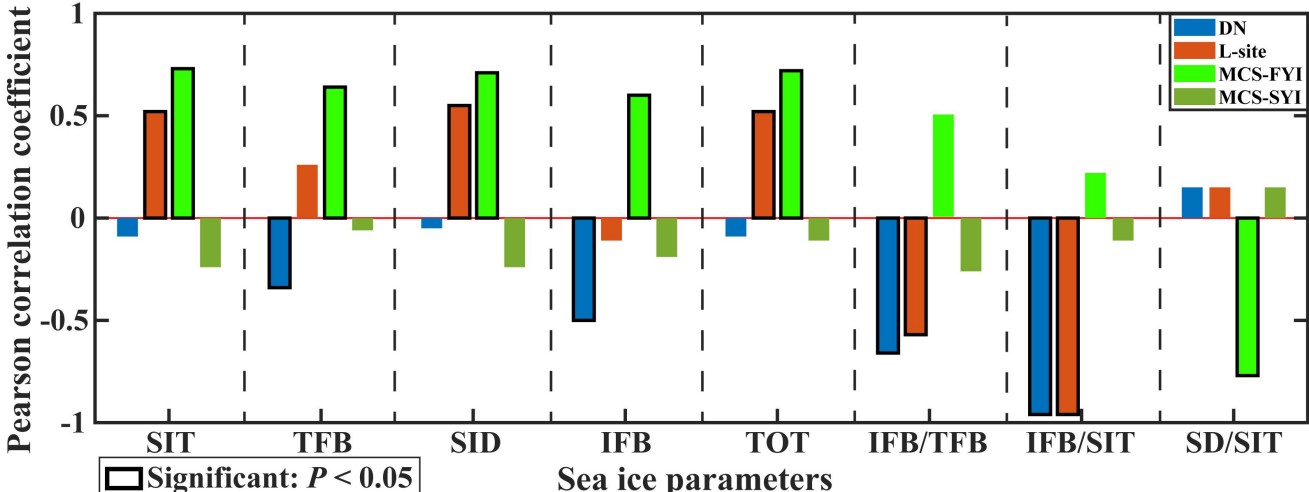

**Figure 7.** Pearson correlation coefficients of univariate and bivariate sea ice parameters against IBD at different spatial scales. The symbols used include: **SIT** for sea ice thickness, **TFB** for total freeboard, **SID** for sea ice draft, **IFB** for sea ice freeboard, **TOT** for total thickness, **IFB/TFB** for the ice freeboard-to-total freeboard ratio, **IFB/SIT** for the ice freeboard-to-thickness ratio, and **SD/SIT** for the snow depth-to-ice thickness ratio. The significance level is set at 95 %.

We then investigate potential IBD parameterization schemes at the regional scale via regression analysis, combining both DN and L-site scale estimates (Fig. 8), and also include results from MCS-FYI and MCS-SYI for comparative analysis (Fig. 9). At the regional scale (Fig. 8), sea ice freeboard showed a statistically significant negative linear correlation with IBD ($R^2$ = 0.13, $P < 0.01$), which agrees well with the results of Jutila et al. (2022a) (hereafter J22) from airborne footprints, although they identified a negative exponential model from a wider range of sea ice freeboard than this study (more details in Section 4.2). Consequently, sea ice freeboard can be considered as an effective single parameter for indirectly determining IBD, applicable both in late spring (J22) and throughout the freezing season (this study). In contrast, sea ice thickness, total thickness, and sea ice draft all showed positive linear variations with IBD, but their fitting performance was relatively poor and unsuitable for IBD parameterization. Total freeboard and the ratio of snow depth to ice thickness were not statistically significant. Analysis of the bivariate parameters indicated a robust linear decrease in IBD with increasing ratios of ice freeboard to total freeboard and ice freeboard to thickness, with respective $R^2$ values of 0.27 and 0.87. Therefore, these two ratios help to characterize different sea ice layers, such as the significant differences in ice porosity between the exposed and submerged portions, which determine the overall state of ice density (Pustogvar and Kulyakhtin, 2016). The strong linear relationship between ice freeboard-to-thickness ratio and IBD derived from MOSAiC observations further supports the use of hypothetical models based on density stratification to calculate IBD (e.g., Alexandrov et al., 2010; Shi et al., 2023). Together with the correlation analysis at both DN and L-site scales, we propose that sea ice freeboard, ice freeboard-to-total freeboard ratio, and ice freeboard-to-thickness ratio can serve as effective schemes for IBD parameterizations at the regional scale.





**Figure 8.** Regression analyses of IBD at the regional scale, including regression models using **(a)** sea ice thickness, **(b)** total thickness, **(c)** total freeboard, **(d)** ice freeboard-to-total freeboard ratio, **(e)** sea ice draft, **(f)** ice freeboard-to-thickness ratio, **(g)** sea ice freeboard, and **(h)** snow depth-to-ice thickness ratio. Each panel shows model fit metrics, including the coefficient of determination ($R^2$) and the statistical test $P$-value (significance level set at 95 %). The blue and red dots represent DN and L-site results, respectively. The total number of samples (Num) is 78.



Regression analyses combining the SYI and FYI cores demonstrated significant differences from the regional scale results (Fig. 9), aligning with the correlation analyses described in Fig. 7. The results indicated that all sea ice parameters were significantly related to IBD, except for the ratio of ice freeboard to total freeboard and the ratio of ice freeboard to thickness. All univariate parameters and total thickness showed a positive power function characteristic, with IBD first showing a
significant increase as these parameters increased, corresponding to the rapid thickening of the FYI, followed by a very weak increase associated with relatively low ice porosity. In contrast, the analysis of ice cores (including FYI and MYI) collected from the Beaufort Sea, as conducted by Kovacs (1997), revealed a significant negative linear relationship between the arithmetic square root of sea ice thickness and IBD (more details in Section 4.2). This suggests that IBD from in situ sampling could be notably influenced by local ice conditions. However, the different methods of calculating ice core
density used by the MOSAiC coring team (Oggier et al., 2023a,b) and Kovacs (1997) are also a potential factor in their opposite relationships between ice thickness and IBD (i.e., hydrostatic weighing method vs. theoretical equation). Moreover, within the range of sea ice freeboard (0.05−0.15 m) at the regional scale, core-based IBDs remain almost stable with increasing ice freeboard, highlighting the strong influence of spatial scale on IBD parameterizations.

The updated IBD parameterizations with simple linear equations (Eq. 4), are designed for use throughout the freezing
season, covering areas up to tens of kilometers dominated by SYI. Furthermore, these parameterizations align well to the grid scales used by most satellite products and models. It is worth noting that the fitting performance of the different parameterizations does not indicate their absolute robustness, as they are not independent of IBD. According to the typical atmospheric and oceanic conditions observed during the MOSAiC expedition (Rinke et al., 2021; Schulz et al., 2024), these parameterizations are expected to be sufficiently representative to be extrapolated to other years. We also argue that the
range of ice freeboard (0.05−0.15 m) and thickness (0.8−1.8 m) are particularly representative of ice regions dominated by SYI, the predominant ice type in the current Arctic Ocean after FYI. In particular, under the influence of global warming, the Arctic MYI has shown a gradual decline in recent decades, while SYI is gradually becoming the more common ice type – a trend that is expected to become even more pronounced in the future (Babb et al., 2023).

The parameterized equations are shown as follows.

$$\overline{\rho_i(x, y, t)} = a1 \times \overline{X(x, y, t)} + a2. \tag{4}$$

where the $\overline{\rho_i(x, y, t)}$ and $\overline{X(x, y, t)}$ represent the IBD and other sea ice parameters at any given location $(x, y)$ and time $(t)$, respectively. Table 2 lists the regression coefficients for Eq. (4). It should be noted that these parameterized equations are not independent. For specific applications, it is recommended to select the appropriate parameterized equation or combination based on the available sea ice parameters.






**Figure 9.** Regression analyses of IBD at the MCS scale, including regression models using **(a)** sea ice thickness, **(b)** total thickness, **(c)** total freeboard, **(d)** ice freeboard-to-total freeboard ratio, **(e)** sea ice draft, **(f)** ice freeboard-to-thickness ratio, **(g)** sea ice freeboard, and **(h)** snow depth-to-ice thickness ratio. Each panel shows model fit metrics, including the coefficient of determination ($R^2$) and the statistical test $P$-value (significance level set at 95 %). The light and dark green dots represent MCS-FYI and MCS-SYI results, respectively. The total number of samples (Num) is 23.




**Table 2.** Regression coefficients of the parameterized equations (kg m⁻³), and parentheses represent 95 % confidence intervals for the regression coefficients.

| $\overline{X(x, y, t)}$ | Variable type | a1 | a2 | Input range |
|---|---|---|---|---|
| Sea ice freeboard (m) | *Univariate* | −127 (−204, −51) | 923 (915, 932) | 0.05−0.15 m |
| Ice freeboard-to-total freeboard ratio | *Bivariate* | −115 (−158, −71) | 950 (935, 966) | 0.26−0.43 |
| Ice freeboard-to-thickness ratio | *Bivariate* | −953 (−1036, −870) | 980 (974, 986) | 0.05−0.1 |

**4 Discussion**

**4.1 Uncertainties and limitations of IBD retrieval**

Recognition of the complexities involved in aligning satellite and airborne measurements with MOSAiC data is essential, given the challenges posed by temporal mismatches, which can vary from several hours to half a day, and the limited track coverage over the MOSAiC CO. These problems also affect the main in situ data in a similar way. Furthermore, IBD

retrieval is constrained by the spatial and temporal limitations of snow pit measurements, along with inherent uncertainties in buoy-derived ice thickness and snow depth. Here, we quantified the relative contributions (RCs) of different parameters to the IBD uncertainty (Fig. 10 and Text S3). The results indicate that snow depth and total freeboard are the primary sources of uncertainty for the IBD estimation at the DN (L-site) scale, contributing 62 % (41 %) and 36 % (57 %) to the total uncertainty, respectively. In contrast, the combined contributions of other factors of seawater density, snow bulk density, and

ice thickness were less than 5 %. Since both parameters can be obtained through aerial observations, even at larger scales of several hundred kilometers, in the future, strengthening the aerial observations of these two parameters will further optimize the IBD parameterizations.

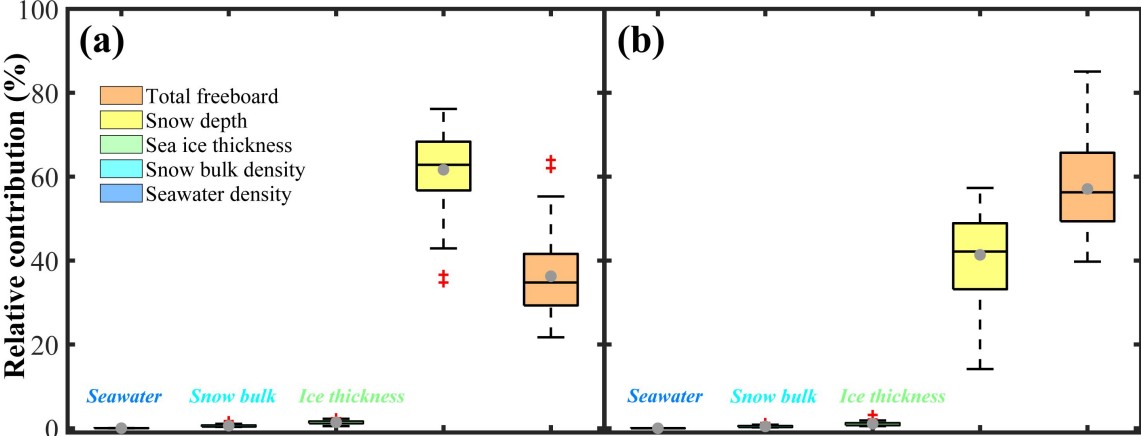

**Figure 10.** Relative contributions of uncertainties in input parameters to the total IBD uncertainty throughout the study period at **(a)** DN

and **(b)** L-site scales. The boxplots depict the inter-quartile range (IQR, Q3−Q1, boxes), median (black lines), mean (grey dots), and outliers (exceeding 1.5 × IQR, red crosses).



In terms of interpretation, the IBD at DN and L-site scales primarily reflects level ice mixed with SYI and FYI, lacking
information on other ice ages and types, particularly deformed ice, which contributes to the potential limitations and
uncertainties of the parameterization scheme. Despite these limitations, we maintain that the optimized parameterization of
IBD derived from the MOSAiC observational data demonstrates broad applicability. The MOSAiC DN region is primarily
characterized by SYI, with only a small portion comprising FYI (Krumpen et al., 2020). This pattern is representative of the
dominant characteristics of the marginal ice zone in the Arctic Ocean during the early ice freezing season, constituting
approximately 30−40 % of the total ice-covered area in the Arctic Ocean. To further develop IBD parameterization for the
pan-Arctic Ocean, we recommend expanding observational data collection to the MYI regions and across the deformed ice
fields.

To match the spatial scales of the satellite and airborne data with the buoy array sites, we made the necessary spatial scale
adjustments (Text S2), although this introduced some uncertainty in the IBD retrieval. We suggest that the spatial scale
adjustments mainly depend on the variability of snow depth in the different observation areas. Due to limited sea ice
freeboard observations during MOSAiC, we relied on ice core data and buoy deployment measurements as initial references
for sea ice freeboard at the DN and L-site scale. The representativeness of the reference values is affected by the spatial
heterogeneity of the MOSAiC ice floes, prompting an investigation into the sensitivity of these reference values in IBD
calculations. We found that a change of +1 cm in the spatial scale adjustment changes the IBD by approximately $-7$ kg m$^{-3}$
throughout the freezing season. Nonetheless, the IBD values at both scales were consistent with historical data and aligned
well with ice core variations, supporting our adjustments. Without these adjustments, the estimated IBD values at the DN
and L-site scales were about 50 kg m$^{-3}$ higher, mainly exceeding 940 kg m$^{-3}$, which is significantly above the expected data
for the SYI. Systematic biases from the spatial adjustment term are expected to be similar at both scales and therefore do not
affect their relative differences. However, it must be acknowledged that the inherent instrumental uncertainties also
contribute to some of the discrepancies between the buoy array and the satellite/airborne measurements.

### 4.2. Intercomparison of IBD parameterizations

Figure 11 compares the IBD parameterizations from this study with existing settings from Kovacs (1997, K97), Alexandrov
et al. (2010, A10), Jutila et al. (2022a, J22), and Shi et al. (2023, S23). Within the sea ice freeboard range of this study, the
results of J22 were consistently higher than those of MOSAiC, mainly due to the inclusion of both deformed and level ice in
J22, with a wider sea ice freeboard range covering FYI, SYI, and MYI (Fig. 11a). Furthermore, the spatial scale of the IBD
has also contributed to the differences between these two parameterizations. The IBD samples used for parameterization in
this study represent an average over several tens of kilometers, whereas J22 employed the weighted average IBD along the
airborne trajectory at the 800 m length scale comparable to the footprint area of the CryoSat-2 satellite altimeter. In terms of
the variation rate of IBD with respect to sea ice freeboard, the two parameterizations were in good agreement within the
freeboard range of 0.05 to 0.15 m. However, J22 demonstrated that the variation rate of IBD gradually decreases with





increasing freeboard until it stabilizes at about 880 kg m$^{-3}$. We also suspect that incorporating IBD samples from the early period with the ice desalination (lack of observations) in the MOSAiC parameterization would result in a steeper decline in IBD at a lower freeboard range, thus corresponding to that shown in J22. Overall, the regional scale observations of drifting ice floes (mostly SYI) throughout the freezing season in the central Arctic Ocean, combined with extensive airborne

observations (FYI, SYI, and MYI) in late April in the western Arctic Ocean, have consistently validated the robustness and effectiveness of using sea ice freeboard as an indirect indicator of IBD.

Moreover, our findings on the ice freeboard-to-thickness ratio were in good agreement with the functions defined by A10 and S23; however, the slope of our parameterization was significantly greater (Fig. 11b). Within the parameter range defined in this study (indicated by the red solid line), our results generally exceeded those of the parametric equation set by A10 for

MYI, particularly at lower ratios of ice freeboard to thickness. As for S23, the differences between their parametric equations for FYI and MYI were relatively small; compared to their results, our updated parametrization showed an initial positive discrepancy, which gradually decreased to a negative value. Among these parameterizations, Alexandrov et al. (2010) first considered systematic differences in the density of MYI above and below seawater, assigning values of 550 and 920 kg m$^{-3}$, respectively, and then calculated IBDs by weighting the ice thicknesses of the upper and lower layers. Shi et al. (2023)

further extended the approach to the FYI and updated the upper layer density setting of the MYI in A10 (from 550 to 815 kg m$^{-3}$) to account for a more realistic porosity of the MYI. Thus, the IBD discrepancies between our updated parameterization and their parametric equations reflect the inherent differences between the statistical method (this study) and the physical assumption model (A10 and S23). We also suggest that the significant scale-related seasonal variations in IBD revealed by the MOSAiC observations could partially explain the much larger slope of our parameterization compared to A10 and S23.

Interestingly, compared to the K97 parameterization derived from ice cores (17 samples of FYI and 4 samples of MYI), the MCS results during MOSAiC showed completely opposite characteristics (13 samples of FYI and 10 samples of SYI), highlighting the significant uncertainties associated with the use of sampling from individual points for IBD parameterization. We found that within the studied ice thickness range, the two parameterizations initially showed a significant difference of 50 kg m$^{-3}$. As the ice thickness gradually increased, this difference decreased and the discrepancies

were essentially resolved for the ice of about 2 m thickness. We also argue that if earlier core samples were available during MOSAiC (with expected thinner ice thicknesses and higher IBDs), the core-based IBD parameterization would be subject to greater uncertainty than in the current situation. Overall, we emphasize that the IBD parameterization based on sampling from individual points has limitations for the application to satellite sea ice thickness retrievals. Nevertheless, ice core sampling remains an indispensable method for the direct acquisition of sea ice properties, providing valuable reference data

for sea ice thermodynamic modelling, biogeochemical cycling studies, climate change assessment, etc.



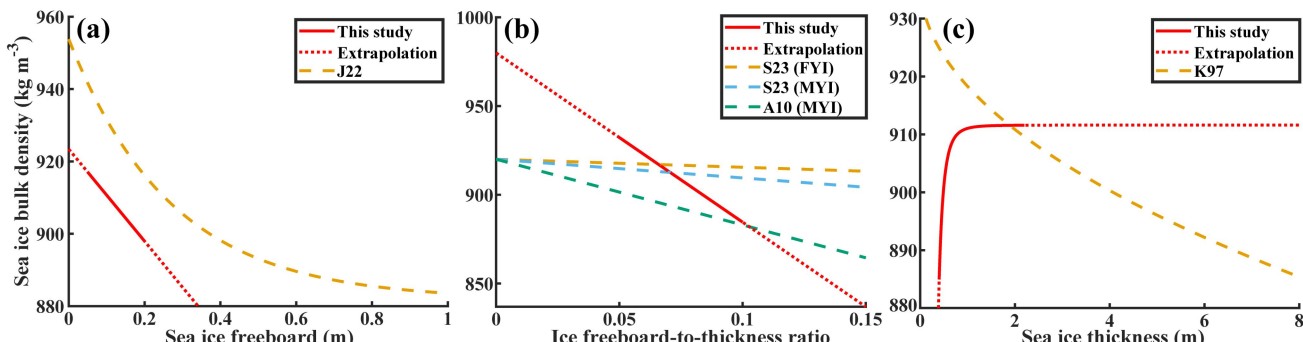

**Figure 11.** Comparison of IBD parameterizations derived from MOSAiC observations with previous studies. **(a)** Sea ice freeboard dependent parameterization including results from MOSAiC (this study) and J22 (Jutila et al., 2022a). The red solid line represents the parameter range of this study, while the dashed line indicates its potential extrapolation. The yellow dashed line represents the parameterization of J22, including FYI, SYI and MYI with both level and deformed sea ice. **(b)** Ice freeboard-to-thickness ratio dependent parameterization including results from MOSAiC (this study), A10 (Alexandrov et al., 2010), and S23 (Shi et al., 2023). The yellow and blue dashed lines show the parameterization results from S23 for FYI and MYI, respectively. The green dashed line represents the parameterization of MYI from A10. **(c)** Sea ice thickness dependent parameterization including results from MOSAiC ice cores (this study) and K97 (Kovacs, 1997). The yellow dashed line represents the parameterization of K97 incorporating both FYI and MYI cores.

### 4.3 Implications of updated IBD parameterizations for the sea ice thickness retrieval

Compared to traditional IBD climatology or representative values, the updated parameterizations proposed in this study have the potential to optimize satellite retrievals of SYI thickness throughout the freezing season. Here, we analyze the potential impact of three regional scale IBD parameterizations (Eq. 4 and Table 2) on the SYI thickness retrieval, using the latest freezing season from October 2023 to April 2024 as a case study. Specifically, we consider the processing chain of the AWI CryoSat-2 sea ice product (AWI CS2) as a benchmark, replacing its original IBD settings based on A10 with the three updated IBD parameterizations (i.e., all other parameters remain unaltered). For the parameterizations that depend on sea ice freeboard and the ratio of ice freeboard to total freeboard, IBD can be calculated directly using individual parameters from the AWI CS2 and then used to retrieve ice thickness. In contrast, the parameterization dependent on the ice freeboard-to-thickness ratio requires a combination with the CryoSat-2 radar freeboard-to-thickness conversion equation (see details in Hendricks and Paul (2023)) to estimate IBD and sea ice thickness simultaneously. The IBD and sea ice thickness results obtained using the three parameterizations are referred to as Case1, Case2 and Case3, respectively.

The SYI component accounted for about 40 % in the early stages of the freezing season and still about 20 % in the later phases, highlighting its importance in current Arctic sea ice fields (Figs. 12a–b). The different cases showed similar updated IBDs and, compared to the original AWI CS2 configuration (i.e. sea ice type-weighted A10 climatology), they were overall higher, especially in autumn significantly up to ~30 kg m$^{-3}$. In terms of IBD variation, the updated IBDs showed some decrease over time, reflecting the decay of ice porosity as the sea ice ages. In contrast, the IBD for AWI CS2 showed a gradual increase, associated with the sea ice type-weighted scheme configured to the A10 climatology used. The corresponding updated SYI thicknesses were overall systematically higher by ~0.2 m compared to the original AWI CS2 estimates throughout the freezing season. In addition, the updated IBD parameterization had reduced seasonal growth rates




for SYI thickness. These results highlight the importance of the updated IBD parameterizations for sea ice thickness retrieval, which needs to be considered in the processing chains of future sea ice thickness products. We must also emphasize that the changes in CryoSat-2 derived SYI thicknesses induced by the updated IBD parameterizations are not the final optimization results, as they are potentially biased by other parameters such as radar freeboard and snow load. In addition, our updated parameterizations may lead to systematic underestimation in grid cells with severe deformation. Nevertheless, compared to

the A10 climatology, which is also based on level ice, we are confident that the IBD results can be optimized for use in sea ice thickness retrieval. In addition, the integration of satellite remote sensing products and the updated IBD parameterizations provides a novel way to obtain IBD for the pan-Arctic SYI to support multiple scientific applications.

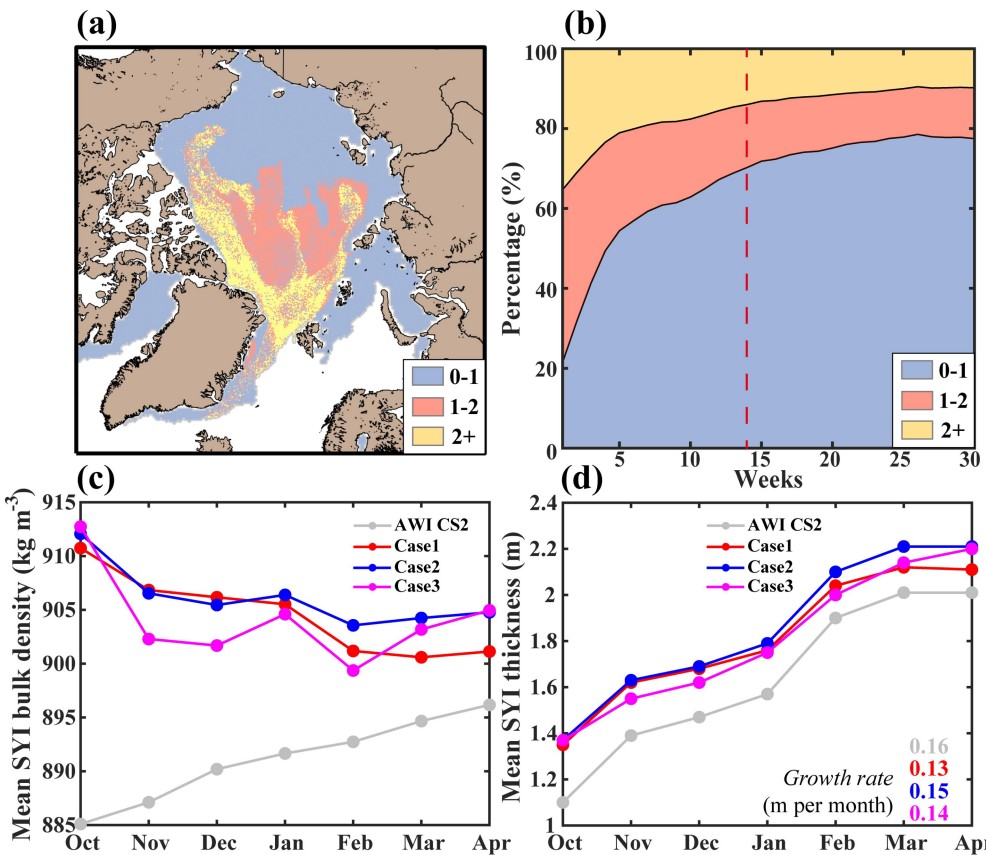

**Figure 12.** Impact of the updated IBD parameterizations on the SYI thickness retrieval based on the AWI CryoSat-2 (AWI CS2) sea ice
product. **(a)** Example of the sea ice age field from 1 January 2023 to 7 January 2023. The different colours represent the sea ice age, including blue (0–1 years, FYI), red (1–2 years, SYI) and yellow (2 + years, MYI). **(b)** Variation in the relative proportions of different sea ice ages during the freezing season. The red dashed line indicates the week shown in panel **(a)**. **(c)** Comparison of the mean pan-Arctic SYI bulk density derived from the three updated parameterizations (Case1, Case2, and Case3) with the original AWI CS2. **(d)** Comparison of the mean pan-Arctic SYI thickness derived from the three updated parameterizations (Case1, Case2, and Case3) with the original AWI
CS2. The numbers in the lower right corner indicate the growth rate of sea ice thickness (m per month, $P < 0.01$).



## 5 Conclusion

This study provides the first estimates and updated parameterizations of IBD during the Arctic freezing season using synergistic measurements from the MOSAiC expedition and ICESat-2. Our methodology proved to be useful in determining IBD over scales of tens of kilometers, thus potentially improving the retrieval of sea ice thickness using the satellite altimeter
observations from multiple satellite missions, such as CryoSat-2, ICESat-1/2, and the upcoming CRISTAL mission (Kern et al., 2020). Given the rapid change in the composition of Arctic sea ice from thicker, older MYI to thinner, younger SYI and FYI components (Babb et al., 2023; Sumata et al., 2023), our updated parameterizations of IBD, explicitly designed for Arctic SYI, could serve as a valuable reference for future satellite retrieval efforts.

In summary, the MOSAiC observations have disclosed pronounced seasonal variations in IBD across the level ice
components. The variation of IBD at the DN (~50 km) and L-site (~25 km) scales were generally agreed with the results from ice cores, but the relative magnitudes were quite different. During the study period from October 2019 to April 2020, the mean IBD values estimated at the DN, L-site, MCS-FYI, and MCS-SYI scales were $910 \pm 7$, $908 \pm 11$, $905 \pm 8$, and $912 \pm 2$ kg m$^{-3}$, respectively. We have shown that the FYI component dominates the IBD variability of the MOSAiC ice floes and can be divided into two main phases, underscoring the complex changes in the ice internal inclusions influenced by ice
ageing and desalination process. During the initial phase from late October to mid-December 2019, IBDs increased from ~890 to 910 kg m$^{-3}$, mainly accompanied by rapid thickening of the FYI and relatively high density for the newly formed ice at the bottom for both FYI and SYI. A warm air intrusion event in mid-November would reduce the ice growth rate and allowed more desalination, thereby moderating the IBD increase. In the subsequent phase from mid-December 2019 to late April 2020, IBDs relatively stabilized at ~910 kg m$^{-3}$ during the main ice growth season of SYI, closely matching historical
measurements for this period. The notable scale-related seasonal variability of IBD further highlights the significant uncertainty associated with retrieving ice thickness from space using fixed IBD values.

We also found that sea ice freeboard, along with the ratios of ice freeboard to total freeboard or thickness, serve as critical indicators to determine the IBD over scales spanning several tens of kilometers. In this context, we developed parameterizations for IBD via regression analyses incorporating both univariate and bivariate ice parameters, expected to be
effective for SYI throughout the freezing season. In contrast, the parameterization of IBD based on spot core sampling sites poses significant challenges due to the strong influence of local ice conditions and differences in the methods used to calculate ice core density. Through initial pan-Arctic IBD estimates for SYI incorporating the AWI CS2 sea ice product and our updated IBD parameterizations, we propose that the new parameterizations have the potential to improve the accuracy of satellite-based sea ice thickness retrievals. The approach proposed in this study to determine IBD throughout the freezing
season can also support interdisciplinary studies, including studies of sea ice modelling, sea ice mass balance over the pan Arctic Ocean, air–sea gas exchanges and biogeochemical cycles in the ice-covered waters, and mechanical interactions between ice and structures.



*Data availability.*

- Snow depth and sea ice thickness data derived from SIMBA buoy measurements are available from PANGAEA: https://doi.org/10.1594/PANGAEA.938244.
- Snow depth and sea ice thickness data derived from SIMB buoy measurements are available from the Arctic Data Center: https://doi.org/10.18739/A20Z70Z01.
- Snow depth and sea ice thickness data from repeated transects are available from PANGAEA:
https://doi.org/10.1594/PANGAEA.937781.
- Total thickness data measured using the GEM-2 from repeated transects are available from PANGAEA: https://doi.org/10.1594/PANGAEA.943666
- Snow pit data collected during the MOSAiC expedition are available from PANGAEA: https://doi.org/10.1594/PANGAEA.940214.
- Snow pit raw data collected during the MOSAiC expedition are available from PANGAEA: https://doi.org/10.1594/PANGAEA.935934.
- Total freeboard data obtained from airborne laser scanning measurements during MOSAiC are available from PANGAEA: https://doi.org/10.1594/PANGAEA.950896.
- Ice core data from the MOSAiC Main Coring Sites are available from PANGAEA:
https://doi.org/10.1594/PANGAEA.956732 (MCS-FYI)
  https://doi.org/10.1594/PANGAEA.959830 (MCS-SYI)
- ICESat-2 ATL10 total freeboard data (version 6, latest version) are available from NSIDC: https://doi.org/10.5067/ATLAS/ATL10.006.
- AWI IceBird airborne multi-sensor sea ice data (version 2, latest version) collected in April 2017 and 2019 are
available from PANGAEA: https://doi.org/10.1594/PANGAEA.966009 (April 2017) and https://doi.org/10.1594/PANGAEA.966057 (April 2019).
- AWI CryoSat-2 sea ice freeboard data (v2p6, latest version) are available from: ftp://ftp.awi.de/sea_ice/product/cryosat2/v2p6/nh/.
- Quicklook weekly sea ice age data are available from: https://doi.org/10.5067/2XXGZY3DUGNQ
- Scale-related bulk densities of level sea ice during the MOSAiC freezing season are available from ZENODO: https://zenodo.org/records/13690816.

*Author contributions.* **YZ (Yi Zhou)** led the study design, developed the methodology, derived the sea ice bulk density data during MOSAiC, performed the primary analysis, and drafted the manuscript. **XW** assisted in developing the methodology,
participated in the primary analysis, and collaborated in manuscript preparation. **RL** co-developed the study design and methodology, participated in the primary analysis, provided the snow depth and ice thickness data derived from SIMBAs, and participated in manuscript drafting. **AJ** analyzed ice density results, provided expertise on airborne measurements,

developed AWI IceBird sea ice bulk density data, and critically revised the manuscript. **DKP** offered expert insights on sea ice mass balance and thermodynamic processes, provided the snow depth and ice thickness data derived from SIMBs,

interpreted results, and reviewed the manuscript. **LvA and CH** aided in interpreting findings and contributed to manuscript revisions. **DVD** delivered critical insights into ice core data interpretation and manuscript revisions. **YZ (Yu Zhang)** supported the interpretation of results and meticulously revised the manuscript.

*Competing interests.* At least one of the (co-)authors is a member of the editorial board of *The Cryosphere*.


*Acknowledgements.* We are deeply indebted to our team members, collaborators, and advisors whose invaluable contributions have been pivotal to this research. We also thank Mats Granskog, Evgenii Salganik, and Hoyeon Shi for their constructive suggestions on the retrieval and analysis of the sea ice bulk density during MOSAiC, which greatly improved the manuscript quality. This study utilizes data generated by the international Multidisciplinary drifting Observatory for the

Study of Arctic Climate (MOSAiC), under the identifier MOSAiC20192020 and Project_ID: AWI_PS122_00. The authors extend their gratitude to the participants of the Research Vessel *Polarstern* (Alfred-Wegener-Institut Helmholtz-Zentrum für Polar- und Meeresforschung, 2017) during the MOSAiC project in 2019–2020, as detailed in Nixdorf et al. (2021). Special thanks are also given to NASA for providing the ICESat-2 along-track freeboard data, NSIDC for contributing the sea ice age data, and AWI for developing the CryoSat-2 sea ice thickness product. These data are critical for estimating the seasonal

bulk density of level sea ice during the MOSAiC expedition.

*Financial support.* YZ (Yi Zhou) and XW were supported by the National Natural Science Foundation of China (Grant No. 42276237). RL was supported by the National Natural Science Foundation of China (Grant No. 42325604). AJ was supported by the Research Council of Finland (Grant No. 341550). DP was supported by the National Science Foundation

(Grant No. NSF-OPP-2034919 and NSF-OPP-2138785). DVD was supported by the Research Council of Norway through project HAVOC (Grant No. 280292). YZ (Yu Zhang) was supported by the National Natural Science Foundation of China (Grant No. 42376231 and Grant No. 42130402). XW was also supported by the Oceanic Interdisciplinary Program of Shanghai Jiao Tong University (Grant No. SL2022PT205).

*Disclaimer.* Publisher's note: Copernicus Publications remains neutral with regard to jurisdictional claims made in the text, published maps, institutional affiliations, or any other geographical representation in this paper. While Copernicus Publications makes every effort to include appropriate place names, the final responsibility lies with the authors.




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
