# Peer review of "Estimating seasonal bulk density of level sea ice using the data derived from in situ and ICESat-2 synergistic observations during MOSAiC"

_EGUsphere, 2024_

## Author Comment (AC1)

**Responses to RC1**

We sincerely appreciate your constructive comments, which have substantially contributed to the improvement of our manuscript. We have incorporated detailed revisions based on your suggestions. The original RC1 comments are presented in *blue italics*, followed by our responses in standard black. Key points are emphasized in **bold black**; **Fig.** and **Table** denote the additional figure and table information, respectively.

*The study titled "Estimating seasonal bulk density of level sea ice using the data derived from in situ and ICESat-2 synergistic observations during MOSAiC" integrates observations of snow and ice thickness from 15 ice mass balance buoys, along with snow freeboard data from ICESat-2, airborne laser altimeters, and snow density data to estimate bulk ice density for a drifting ice floe and its surroundings at various scales. It also compares these estimates with manual observations of ice density and proposes several parameterizations, primarily using ice freeboard and thickness as key parameters.*

*Parametrization of ice density is a valuable task, and the study provides new data and implements known concepts to derive density estimates from large scale observations. The methods are clearly described, yet some details do not allow a reproducibility. The overview of remote methods is accurate enough, while an overview of in situ measurements is quite limited. In addition, there are a lot of inconsistencies related to the description of representative ice types and ice and snow thicknesses, related to a limited usage of existing in situ observations during the described expedition.*

*While the study offers valuable insights into the upscaling of sea ice freeboard data during the ice drift, where measurements were limited to individual floes, there are some methodological challenges that should be considered. For instance, the ice density estimates rely on a methodology with certain limitations. Some of these issues were addressed by manually adjusting the freeboard measurements, although the rationale and approach behind this adjustment could be more clearly explained. While these adjustments resulted in more realistic density estimates, there remains significant variability in the data, similar to the previously observed seasonal fluctuations in ice density.*

*Given this level of variability and the presence of an adjustment parameter that heavily influences both the density values and their seasonality, it is challenging to recommend a robust parameterization based on the current data. Additionally, the assumption that the adjustment is time-independent may need further justification. As a result, it becomes difficult to fully assess the quality of the density estimates due to both the high variability and the manual adjustments made during the analysis.*

**Overall Reply** We appreciate the reviewer's feedback on the manuscript and we have made the following improvements: 1) we clarified the regional representativeness and validity of the retrieved IBDs; 2) we added specific descriptions for the spatial scale adjustment; 3) we incorporated physical interpretations for the IBDs and developed additional parameterizations; 4) we thoroughly analyzed the uncertainties and constraints associated with the IBD process; 5) we evaluated the developed parameterizations against coring measurements from late October 2022, with a resulting relative error ranging from 0.2 to 0.9 %; 6) We estimated the pan-Arctic second-year ice bulk densities from 2010 to

2024 to demonstrate the applicability of our parameterizations; 7) we also proposed a new age-related IBD climatology, taking into account the limitations of the parameterizations.

**Representativeness** Indeed, our estimated IBDs represent the average condition of the ice layer as measured by the selected IMBs, as illustrated in **Fig. A1**. Given the considerable spatial variability in sea ice and snow thickness observed among the MOSAiC ice floes, we retract our previous statement that the results represent typical ice conditions at the DN or L-site scales. **As a result, all regional scale statements are based on the results from the buoy sites (SYI-dominated), which also motivate the so-called spatial scale adjustment.**

[Figure]

**Figure A1.** Flowchart of the IBD retrieval process.

**Compatibility of spatially non-overlapping data** The spatial scale adjustment was carefully evaluated, and the resulting values were derived from specific calculations rather than being manually determined; **we then provided additional details.** The purpose of this adjustment is to align the large-scale observations with the buoy-deployed ice layers. We explain how the correction term was derived and why it is time-independent in our response to the general comments (see details in **GC2**). Essentially, the spatially scaled modal total freeboard can be considered an approximation of the mean total freeboard across the buoy sites.

**Validity, Limitations, and Uncertainties** We argue that IBDs derived from buoy sites (regional) remain valuable, demonstrating relatively strong agreement with direct measurements (local) during MOSAiC, and illustrating a **general seasonal pattern** (**Fig. A2**, **Fig. A3**, and **Table A1**), while also falling within the range of historically reported FYI and SYI bulk densities. We discussed in **GC1** why the regional estimates were more consistent with the IBD variation of MCS-FYI ($R = 0.71$ , $P < 0.01$), despite being mostly composed of SYI. We also acknowledge that the hydrostatic equilibrium method for calculating IBD has inherent limitations and challenges, as noted by Hutchings et al. (2015) and Jutila et al. (2022), primarily related to deformation- and scale-related hydrostatic equilibrium failures. Nevertheless, it remains suitable for estimating the bulk density of level sea ice over tens of kilometers under the assumption of hydrostatic equilibrium (see details in **GC3**). Due to irregular variations in

airborne and satellite trajectories, and the limitations of the buoys in capturing daily variations in mean snow depth over their deployment area, we emphasize that the retrieval method may not fully resolve synoptic-scale IBD variations, with a relatively high uncertainty of ~3 % (**Fig. A4**).

[Figure]

**Figure A2.** **(a)** Seasonal variation of IBD estimates at the DN (blue bars), L-site (red bars), and MCS (green symbols) scales during the MOSAiC freezing season. The black dots are the regional mean of the DN and L-site estimates. The orange dashed line represents the linear fitting from late October to early December ($R = 0.62$, $P < 0.05$) and the purple dashed line indicates the mean ± one standard deviation from late December to April. **(b)** Linear interpolation (black) and smoothing (red) of regional IBD estimates. **(c)** Box plots of all IBD estimates over the study period, showing interquartile range (IQR, Q3−Q1, boxes), median (black lines), and outliers (greater than 1.5 × IQR, red crosses). **(d)** Probability density distribution (PDF) of all IBD estimates over the study period, with labelled values showing the mean ± one standard deviation.

[Figure]

**Figure A3.** Scatterplot of regional versus local IBD estimates. The regional IBD represents the smoothed result derived from both DN and L-site scale estimates, while the local IBD represents the MCS-FYI. The solid orange line indicates the fitted results and the dashed black line shows the one-to-one relationship. The statistical metrics include the correlation coefficient ($R$), the statistical test $P$-value, and the total number of samples (*Num*).

**Table A1.** Seasonal averages of IBD at regional and local scales during the MOSAiC freezing season, where ON denotes autumn (October to November), DJF denotes winter (December to February), and MA denotes spring (March to April). Statistical information is presented as mean value (Mean), standard deviation (Std) and total number of samples (*Num*).

| Season | DN & L-site (regional) | | | MCS-FYI (local) | | | MCS-SYI (local) | | |
|---|---|---|---|---|---|---|---|---|---|
| | Mean (kg m$^{-3}$) | Std (kg m$^{-3}$) | *Num* | Mean (kg m$^{-3}$) | Std (kg m$^{-3}$) | *Num* | Mean (kg m$^{-3}$) | Std (kg m$^{-3}$) | *Num* |
| ON | 900 | 14 | 5 | 899 | 8 | 5 | 912 | 1 | 4 |
| DJF | 910 | 9 | 45 | 909 | 4 | 6 | 911 | 3 | 4 |
| MA | 910 | 7 | 28 | 913 | 3 | 2 | 912 | 2 | 4 |

[Figure]

**Figure A4. (a)** Seasonal variation of IBD uncertainties at the DN (blue dots) and L-site (red dots) scales during the MOSAiC freezing season. Also shown are the relative contributions of uncertainties for different input parameters to the total IBD uncertainty over the study period at the **(a)** DN and **(b)** L-site scales, respectively. The boxplots depict the inter-quartile range (IQR, Q3−Q1, boxes), median (black lines), mean (grey dots), and outliers (exceeding 1.5 × IQR, red crosses).

*General comments*

*A parameterization of sea-ice density can indeed be a valuable tool for both modelers and observers. However, I have some concerns regarding the potential usefulness of the parameterization presented here for such applications. Firstly, it is based on a methodology with significant uncertainties, which in many cases seem to exceed the actual seasonal variability of sea-ice density. Secondly, the current parameterization focuses on correlations between density and ice freeboard or thickness, without addressing important physical factors such as region, ice salinity, ice temperature, or water temperature. Since ice freeboard is inherently linked to its density, this creates a positive feedback loop. While snow may also play a role, it is less likely to directly influence ice density. Low ice freeboard can result from both high ice density and reduced thickness, complicating the interpretation. Similarly, the relationship between ice thickness and density involves multiple factors, including the region, ice age, and type, in addition to seasonal variations. Suggesting that only very thin ice has lower density seems to oversimplify the complexity of these processes. Additionally, it may be less useful to rely on ice thickness as an input parameter, as this is typically the primary goal of many remote sensing estimates rather than an available input variable.*

**GC1 Reply** The validity of the IBD data has been discussed above and carefully addressed in **GC2** and **GC3**. We also argue that IBD variability at the sub-weekly scale does not undermine the overall correlation between IBD and other parameters. In the revision, we focus on comparing the potential IBD parameterizations derived from regional (buoy sites) and local (coring sites) estimates, for which we no longer assert absolute robustness due to inherent differences in spatial scale, measurement methods, sample sizes, and ice types.

**GC1: Clarification**

- We agree with the reviewer that predicting IBD from sea ice parameters alone is challenging. Given the complexity of the multiple processes and parameters influencing IBD, the primary objective of this study is to propose some **potential simple parameterizations** for predicting IBD throughout the freezing season, adapted to **contemporary ice conditions in the central Arctic**. We also would like to clarify that the samples used for the IBD parameterization **themselves have time-varying properties** such as ice porosity, temperature, salinity, thickness, and age.

- The IBD parameterization **involving sea ice thickness remains valuable** for satellite-based retrievals of sea ice thickness, as it can be combined with radar/laser freeboard-to-thickness conversion (when other parameters are known) to estimate both IBD and sea ice thickness simultaneously. We highlight that the regional IBD results are not independent of their sea ice parameters and are therefore subject to error propagation. Nevertheless, we suggest that these regional IBD parameterizations, which are **representative of the hydrostatic equilibrium-based IBD**, and the coefficients derived from observations, still provide valuable guidance for satellite retrievals of sea ice thickness based on the same principles.

**GC1: Comparison of IBD Parameterizations** Comparison of IBD parameterizations at regional and local scales reveals significant effects of spatial scale (buoy array vs. spot core sampling) and computational method (hydrostatic equilibrium vs. weighing) on the IBD parameterzation (see **Fig. A5** and **Fig. A6**). **Our potential parameterized equations for IBD at the regional scale support many of the previous findings.** *For example, Ackley et al. (1974) proposed a negative linear equation dependent on ice freeboard, Kovacs (1997) developed a power equation dependent on ice thickness, Alexandrov et al. (2010) and Shi et al. (2023) defined positive linear equations dependent on the ice freeboard-to-thickness ratio, and Jutila et al. (2022) developed a negative exponential function dependent on ice freeboard.*

[Figure]

**Figure A5.** Potential parameterizations for IBD at the regional scale and comparison with results at the local scale, including regression models using **(a, b)** sea ice freeboard, **(c, d)** ice freeboard-to-total freeboard ratio, and **(e, f)** ice freeboard-to-thickness ratio. Each panel shows model fit metrics, including the coefficient of determination ($R^2$) and root mean square error (*RMSE*, unit: kg m$^{-3}$). The significance level for the statistical test is set at 95%. The total number of samples *(Num)* is 78 for the regional scale and 23 for the local scale.

[Figure]

**Figure A6.** Potential parameterizations for IBD at the local scale and comparison with results at the regional scale, including regression models using **(a, b)** sea ice thickness, **(c, d)** sea ice draft, and **(e, f)** total thickness. Each panel shows model fit metrics, including the coefficient of determination ($R^2$) and root mean square error (*RMSE*, unit: kg m$^{-3}$). The significance level for the statistical test is set at 95%. The total number of samples (*Num*) is 23 for the local scale and 78 for the regional scale.

**GC1: Parameterization Considering Physical Factors** We investigated whether physical factors could enhance the established IBD parameterizations based on sea ice parameters. Here, a multiple linear regression model was used to investigate the relationships between these variables and IBD. The analysis showed that **sea ice bulk temperature significantly improved the performance of the developed regional IBD parameterizations** (see Fig. A7 and Table A2), particularly the sea ice freeboard-dependent parameterization ($R^2$ improved by ~85%), while air and seawater temperatures were not helpful suggesting that they are insufficient to explain the seasonal variability of IBD. None of these multivariate regression models suffer from multicollinearity problems, with variance inflation factors (VIFs) less than 2. In contrast, the introduction of any physical factor (including sea ice bulk temperature and salinity) into the developed local IBD parameterizations did not improve their original fitting performance, possibly due to the spatial heterogeneity of ice porosity and the potentially non-linear relationship.

[Figure]

**Figure A7.** Potential parameterizations for IBD at the regional scale using both sea ice parameters and sea ice bulk temperature, including **(a)** sea ice freeboard and sea ice bulk temperature versus IBD, **(b)** ice freeboard-to-total freeboard ratio and sea ice bulk temperature versus IBD, **(c)** ice freeboard-to-thickness ratio and sea ice bulk temperature versus IBD. Each panel shows model fit metrics, including the coefficient of determination ($R^2$) and root mean square error *(RMSE,* unit: kg m$^{-3}$). **Note:** The significance level for the statistical test is set at 95% and the total number of samples is 78.

- **The parameterized equations for IBD at the regional scale combing both sea ice parameters and physical factors are defined as follows:**

$$\overline{\rho_i(x,y,t)} = a1 \times |\overline{\delta(x,y,t)}| + a2 \times \overline{X(x,y,t)} + a3, \tag{A1}$$

where $|\overline{\delta(x,y,t)}|$ represent the absolute value of regional sea ice bulk temperature at any given grid cell over tens of kilometers scale $(x,y)$ and time $(t)$. Table A2 lists the regression coefficients and their 95 % confidence intervals for Eq. (A1).

**Table A2.** Regression coefficients of the parameterized equations based on sea ice parameters and sea ice bulk temperature at the regional scale (kg m$^{-3}$), and parentheses represent 95 % confidence intervals for the regression coefficients.

| $\overline{X(x,y,t)}$ | Variable type | a1 | a2 | a3 |
|---|---|---|---|---|
| Sea ice freeboard (m) | *Univariate* | 2.1 (0.7, 3.5) | −198 (−290, −106) | 913 (902, 924) |
| Ice freeboard-to-total freeboard ratio | *Bivariate* | 2.7 (1.5, 3.8) | −155 (−197, −112) | 942 (928, 956) |
| Ice freeboard-to-thickness ratio | *Bivariate* | 1.2 (0.8, 1.6) | −967(−1036, −899) | 970 (964, 976) |

**Figure A8** illustrates the seasonal variation in air temperature, ice bulk temperature, ice bulk salinity, and brine volume during the MOSAiC freezing season. **According to the theoretical relationship (Timco and Frederking, 1996), the recorded sea ice temperature and bulk salinity variations cannot account for the seasonality of IBD over the study period.**

- **MCS-FYI.** The significant increase in IBD during the early stages is closely related to the **significant reduction in the air fraction** (Salganik et al., 2024), as the conversion of liquid water to solid ice in closed brine pockets causes volume expansion, compressing trapped air bubbles and pushing gas into the brine solution as temperatures fall (Crabeck et al., 2016; Crabeck et al., 2019).

- **MCS-SYI.** The MCS-SYI showed a relatively stable IBD with an increase of only ~3 kg m$^{-3}$ during the early stages, **corresponding to its relatively low thickness growth rate (~9 % per month) and low porosity** with a brine volume of ~1 to 4 % (**Fig. A8**) and an air fraction of ~0.8 to 1 % (Salganik et al., 2024).

- **Regional.** The regional IBD results also showed some increasing trend (~14 kg m$^{-3}$ per month, $P < 0.05$), similar to the MCS-FYI, but the limited sample and relatively high uncertainty should be noted (average of ~30 kg m$^{-3}$). In this context, the mean ice thickness at the DN and L-site scales experienced monthly growth rates of ~15 and 21 %, respectively, with the majority of the buoy-monitored ice layers beginning to grow. **This also reveals the potential difference in ice properties between local (MCS-SYI) and regional (SYI-dominated buoy sites) scales.**

*"In particular, Hornnes et al. (2024) found that the spatial heterogeneity of SYI porosity was much greater than that of FYI during the GoNorth expedition in October 2022, and also found similar air fractions in FYI (2.5 ± 0.5 %) and SYI cores (2.8 ± 0.5 %)."*

[Figure]

**Figure A8.** Seasonal variation of **(a)** buoy-derived air temperature, **(b)** buoy-derived and core-based sea ice bulk temperature, **(c)** core-based sea ice bulk salinity, and **(d)** core-based brine volume during the MOSAiC freezing season. The red boxes in panel **(a)** mark warm air intrusion events, and the dashed line in panels **(c, d)** indicates the seasonal linear trend ($P < 0.01$).

**GC1: Updating IBD Climatology**

Given the limitations of the developed IBD parameterizations, we proposed a new age-related IBD climatology for the freezing season by combining the results from the Arctic Sever expedition (Alexandrov et al., 2010; Shi et al., 2023), the IceBird campaigns (Jutila et al., 2022), and the MOSAiC expedition (this study), as shown in Table A3. These age-related IBD values can serve as an updated IBD climatology for future basin-scale satellite sea ice thickness retrievals, providing an alternative to A10, with two key upgrades: **1) additional SYI information, and 2) mitigation of the potentially biased A10-MYI.**

**Table A3.** Age-related bulk densities of level sea ice during the freezing season derived from a combination of the Arctic Sever expedition (from 1980 to 1989), the IceBird campaigns (April 2017 and 2019), and the MOSAiC expedition (from 2019 to 2020). Uncertainty is given as the maximum difference in age-related mean IBD values across observations.

| Ice age | Mean (kg m$^{-3}$) | Standard deviation (kg m$^{-3}$) | Uncertainty (kg m$^{-3}$) |
|---|---|---|---|
| **FYI (0–1 years)** | **916.9** | 7.5 | 20 |
| **SYI (1–2 years)** | **907** | 6.8 | 13 |
| **MYI (2+ years)** | **900** | 14 | 34 |

During the GoNorth Expedition in late October 2022 (Salganik et al., 2023), we used the SYI core results to evaluate the developed IBD parameterizations within their potential applicability ranges, along with the age-related IBD climatology (Fig. A9). The FYI core was not selected because its ice parameters were not applicable to the proposed parameterizations (mean ice thickness of 0.31 m). We found that the regional IBD parameterizations maintain respectable accuracy compared to local direct measurements, even surpassing the results of parameterizations constructed from MOSAiC ice cores.

[Figure]

**Figure A9.** Evaluation of the regional and local IBD parameterizations during the GoNorth Expedition in late October 2022. The age-related IBD climatology and its uncertainty are also shown. **Note: Weighing** represents the direct measurement of IBD using the hydrostatic weighing method with an uncertainty defined as 0.2 %. **Reg.** and **Loc.** are regional and local scales, respectively. **IFB**, **Tem**, **SIT**, **SID**, and **TOT** represent ice freeboard, ice bulk temperature, ice thickness, ice draft, and total thickness, respectively. The uncertainty of the IBDs derived from the parameterizations is determined by the 95 confidence intervals of the regression coefficients.

**GC1: Application**

We combined AWI CS2 sea ice data, the ice age field, and regional IBD parameterizations to provide preliminary insights into the pan-Arctic monthly SYI bulk densities and their updated thicknesses during the 2010 to 2024 freezing seasons (Fig. A10). The final results are presented as averages derived from the three regional IBD parameterizations. The incorporation of the regional IBD parameterizations into the current AWI CS2 product introduce**d more realistic variations in SYI bulk densities**, leading to **systematically higher SYI thickness estimates and reducing their seasonal growth rates and interannual trends**. Future incorporation of pan-Arctic SYI bulk temperatures, along with more accurate sea ice ancillary parameters, will be more effective in retrieving SYI bulk densities. Notably, the IBD for AWI CS2 showed a gradual increase, associated with the sea ice type-weighted scheme configured to the A10 climatology used (i.e., non-physical variation).

In contrast to the increasing trend of IBD from late October to early December during MOSAiC, pan-Arctic SYI bulk densities showed a decrease (~6 kg m$^{-3}$) from October to November, which may be related to several factors: 1) lack of IBD samples in early October during MOSAiC (possibly main brine discharge); 2) high spatial heterogeneity of SYI porosity; 3) extrapolation of the regional parameterizations to pan-Arctic SYI; 4) inherent uncertainty in AWI CS2 sea ice parameters; 5) failure of the parameterizations for severe sea ice deformation units. **Nevertheless, the results were consistent with the expected range and evolution of SYI.**

[Figure]

**Figure A10.** IBD and thickness variation of the pan-Arctic SYI during the 2010 to 2024 freezing seasons based on the AWI CryoSat-2 (AWI CS2) sea ice product, including results using our parameterizations (Y24) and the original settings (sea ice type-weighted A10 climatology). The shaded areas denote mean ± one standard deviation and dashed lines indicate linear trends with their statistical test *P*-values. **(a)** Multi-year monthly average estimates for IBD from October to April. **(b)** Multi-year monthly average estimates for ice thickness from October to April. **(c)** Freezing season average estimates for IBD from 2010/2011 (10/11) to 2023/2024 (23/24). **(d)** Freezing season average estimates for ice thickness from 2010/2011 (10/11) to 2023/2024 (23/24).

*Another significant concern regarding the methodology of this study involves the so-called "spatial scale adjustment." First, the issue with buoys is not solely about spatial coverage or the number of buoys deployed. Rather, it arises from the fact that the buoys were installed in ice that may not be representative of the broader area. Secondly, such a substantial adjustment requires a more detailed explanation than a brief mention with a reference to supplementary materials. The adjustment of 7 cm to the freeboard, which accounts for roughly a third of the average total freeboard during the observation period, is quite significant. According to the methodology, ice freeboard is calculated as the difference between ICESat-2 snow freeboard and buoy snow thickness. Since the buoy snow thickness remains unchanged, applying a 7 cm adjustment equates to adjusting the ice freeboard by 7 cm, which corresponds to approximately 65 cm of ice thickness. This can be shown considering the initial buoy ice thickness was around 100 cm, while the modal ice thickness from electromagnetic sounding was between 40-50 cm, giving a similar difference in ice thickness, as the adjustment equivalent. Such a considerable adjustment to the interpretation of ice thickness, which ultimately impacts the density estimates, warrants a more thorough explanation of both its value and physical meaning. Additionally, it is perplexing that instead of adjusting the relatively uncertain estimates of snow and ice thickness from the buoys, the ICESat-2 and airborne laser scanner data were adjusted instead and presented in this adjusted form. This 7 cm adjustment was applied across the entire seven-month observation period, but it is not clear why this adjustment was considered time-independent. If the buoys were installed in areas with unrepresentatively thick ice, it seems unlikely that this bias in ice thickness would remain constant throughout the ice growth season. Numerous observations indicate that by the end of the cold season, initially thinner ice can grow more rapidly due to thinner snow cover, making the differences between thick and thin ice less distinguishable. Furthermore, another issue arises from the fact that buoys were installed in areas with thicker ice, which is known to correlate strongly with snow thickness. Even toward the end of the season, ice with greater surface roughness tends to accumulate more snow compared to smoother, younger ice. This raises questions about the representativeness of the snow thickness measurements from the buoys. These considerations lead to a broader question regarding the interpretation of the seasonal changes presented in the study. How might these changes be affected by the unrepresentativeness of the buoy measurements?*

**GC2 Reply** The spatial scale adjustment is designed to compensate for systematic differences in snow depth (rather than sea ice freeboard) between buoy sites and large-scale observations, as determined by **a comprehensive intercomparison** of observations from buoy, ice core, transect, and airborne/satellite measurements during the early study period (**Table A4**, **Fig. A11**, and **Fig. A12**).

In the initial phase, a comparison of MCS-FYI, MCS-SYI, and regional measurements (Krumpen et al., 2020) indicated that snow depth was the primary contributor to the total freeboard of level sea ice, and that their sea ice freeboards were very similar, despite significant differences in their ice thicknesses (up to 0.4 m) and spatial scales (local and regional), as shown in **Table A4**. Furthermore, the initial snow and ice thickness values at the SYI-dominated buoy sites were comparable to those observed in the MCS-SYI (see **Fig. A12**). This suggests that the spatial heterogeneity of level ice freeboard between the buoy sites and the large-scale airborne/satellite measurements is likely to be small, despite the absence of direct freeboard measurements for either. A subsequent comparison of buoy snow observations at the DN and L-site scales from 5−11 October 2019 with corresponding regional transect measurements showed that buoy-derived mean snow depths (0.14 m) were **approximately 7 cm**

**higher at both scales.** This highlights that snow conditions on buoy-observed ice layers may not accurately reflect the magnitude of snow depths over a broader measurement area (and vice versa). However, **the seasonal trends in mean snow depth at the buoy sites were closely aligned with the modal snow depths derived from transects over the CO floe** throughout the study period (see Fig. A12), which covered a much larger area. This finding suggests that the buoys effectively captured seasonal snow accumulation over level sea ice within their deployment area. Therefore, it can be concluded that the modal total freeboard derived from airborne/satellite measurements could be systematically adjusted by approximately 7 cm to align with buoy observations.

**The spatial adjustment term was further validated by using ice core data as the initial freeboard reference for the buoy sites.** In this context, we calculated the adjusted value at the corresponding regional scale as the sum of two terms: 1) the absolute value of the maximum negative difference between the original modal total freeboard and buoy-derived mean snow depth (**~0.4 m**, see Fig. A11), and 2) the initial freeboard reference obtained from the FYI and SYI cores in late October, determined by weighting the FYI/SYI ratios from the corresponding buoy sites (**~0.3 m**). The buoy-derived mean ice thickness and snow depth were comparable in mid-October (initial satellite record) and late October (initial ice core record), and the timing difference is not expected to significantly impact the reference freeboards. The final adjusted values were **approximately 7 cm at both spatial scales**, which aligned with those obtained from snow depth differences between the regional transects and buoy sites. This, in turn, suggests that the spatial scale difference between the two observations results from a systematic bias in snow depth rather than sea ice freeboard. Together with the ability of the buoys to capture seasonal snow variations, we ensured that the difference between the adjusted modal total freeboard and the buoy-derived mean snow depth reasonably approximated the sea ice freeboard across the buoy sites, thereby ensuring compatibility between the buoy array and the airborne/satellite measurements used in the IBD retrieval.

**Table A4.** Comparison of ice conditions in different measurement domains during the early stages of the MOSAiC expedition.

| Data (scale) | Time | Sea ice thickness (m) | Snow depth (m) | Total freeboard (m) | Sea ice freeboard (m) |
|---|---|---|---|---|---|
| Ice cores (MCS-FYI) | 27 Oct | 0.42 | 0.09 | 0.11 | **0.023** |
| Ice cores (MCS-SYI) | 28 Oct | 0.77 | 0.15 | 0.18 | **0.03** |
| Krumpen et al. (2020) (Regional transects) | 5−11 Oct | 0.36 | 0.07 | 0.10* | **0.03*** |

**Note** that data marked with an asterisk (*) are based on modal total freeboard derived from ALS and ICESat-2 on 11 Oct 2019. Sea ice thickness and snow depth from Krumpen et al. (2020) are weighted estimates according to daily observation samples or transect distances from 5 to 11 October 2019.

[Figure]

**Figure A11.** Seasonal variation of the approximate sea ice freeboard for buoy sites during the MOSAiC freezing season, calculated by subtracting the buoy-derived mean snow depth from the adjusted and original modal total freeboards at the DN (blue) and L-site (red) scales, respectively. The black dashed line indicates the zero sea ice freeboard, and the yellow bar highlights the maximum negative difference between the original modal total freeboard and the buoy-derived mean snow depth.

[Figure]

**Figure A12.** Seasonal variations of sea ice and snow parameters from level ice components during the MOSAiC freezing season. **(a)** Sea ice thickness measurements obtained from buoys (blue, red, and magenta lines for DN, L-site, and CO scales, respectively), transects (black crosses, mode), and ice cores (green symbols). **(b)** Snow depth measurements recorded from buoys (blue and red lines for DN and L-site scales, respectively), transects (black crosses, mode), and ice cores (green symbols). **(c)** Total freeboard measurements derived from ALS & ICESat-2 (blue and red dots for DN and L-site scales, respectively) and ice cores (green symbols). The error bars represent the uncertainties of the modal total freeboard. **(d)** Snow bulk density measurements ascertained from snow pits (grey dots), with error bars denoting one standard deviation from all daily estimates. The red line indicates a linear fit and the shaded band labels the 95 % confidence interval. **Note:** The dashed lines in panels **(a)**, **(b)**, and **(c)** represent seasonal linear trends, with all statistical *P*-values less than 0.01.

Given the relatively high uncertainties and sparse samples associated with the increasing trend of regional IBDs in the early stages, we introduced hydrostatic equilibrium-based modelled IBD estimates based on a simple trend assumption, to further elucidate the potential seasonal shifts in IBD. Specifically, we set the initial sea ice thickness, snow depth, and total freeboard in mid-October to values of 0.36, 0.07, and 0.10 m, respectively, which are expected to represent typical ice conditions for MOSAiC level ice (Table A4). We then calculated the average seasonal trends for these parameters during MOSAiC as ~0.20, 0.01 and 0.03 m per month according to Fig. A12, and extrapolated these initial values to April. Finally, using the hydrostatic equilibrium equation and the same snow bulk and seawater densities used for the regional IBD estimates, we calculated the so-called modelled IBD (Fig. A13). It should be noted that the modelled values can only indicate the approximate seasonal trends of IBD, completely filtering out possible IBD changes at daily and weekly scales. **Overall, the derived modelled values also supported the early increase trend of IBD depending on the seasonality of the different observed ice parameters during MOSAiC.**

[Figure]

**Figure A13.** Hydrostatic equilibrium-based modelled IBD estimates (M-IBD) during the MOSAiC freezing season.

*A third concern relates to the interpretation of in situ density measurements. Based on the abstract from the coring dataset by Oggier et al. (2023), it appears that the sea-ice density was measured at constant laboratory temperatures rather than at in situ conditions. However, this is (1) not explicitly mentioned in the study, and (2) there is no description of how density was recalculated to reflect in situ temperatures, if such a correction was applied. If this adjustment was not made, any discussion of how brine volume evolution affects sea-ice density becomes less meaningful. Furthermore, the interpretation of the measured density shows a larger standard deviation from actual weighing than from the combination of ICESat-2 freeboard data and buoy snow and ice thickness. This seems to contradict previous estimates of uncertainties in density measurements (Hutchings et al., 2015; Pustogvar and Kulyakhtin, 2016). Specifically, weighing has consistently been found to be one of the most precise methods, while density estimates based on ice freeboard and thickness measurements tend to have standard deviations several times higher—by an order of magnitude—according to Hutchings et al. (2015), even when the thickness and freeboard were measured in situ. Given this prior evidence, it is somewhat surprising that the reported standard deviations suggest the opposite pattern in this study. A clearer explanation of these discrepancies would clarify the interpretation of these measurements.*

**GC3 Reply** Although the laboratory temperature (−15 °C) varies from the in situ temperature, Pustogvar and Kulyakhtin (2016) demonstrated that the hydrostatic weighing method effectively mitigates the density errors associated with brine drainage due to ambient temperature. Even under extreme conditions, such as a 19 % brine loss, the measurement error remains as low as 2 %, while the mass/volume method can introduce errors of up to 20 %. Specifically, the hydrostatic weighing method can effectively limit the loss of brine during ice sectioning at low laboratory temperatures compared to in situ conditions, where brine losses can be greater in warm ice. The use of kerosene coating further helps to minimize brine loss by occupying the spaces at the core interfaces, thus improving the accuracy of the measurements. Salganik et al. (2024) found little impact of brine loss on hydrostatic weighing density measurements by comparing sea ice densities reconstructed from in situ temperature and freeboard/thickness, exhibiting a similar variation pattern. **Therefore, despite measurements being conducted at a constant laboratory temperature, this method provides highly reliable and accurate results, facilitating a meaningful discussion of brine volume evolution and its impact on sea ice density.**

**In the revision, we have added detailed information on ice core parameters to the text.** *"The ice temperature was measured in the field using Testo 720 thermometers at 5 cm vertical intervals; ice salinity was determined from melted cores using a YSI 30 conductivity meter at 5 cm resolution; ice density was derived in the freezing laboratory at an ambient temperature of –15 °C using the hydrostatic weighing method. The temperature and density results were further interpolated to match the depth of the salinity measurements, providing a continuous density profile throughout the core sample. Then, brine volume was calculated from measured ice salinity, temperature, and density using the equations of Cox and Weeks (1983) for cold ice and Leppäranta and Manninen (1988) for ice warmer than –2 °C."*

**Our regional IBD results did not show a lower standard deviation than those of direct measurements.** Except in the initial stages, the standard deviation of the regional IBDs exceeded twice that of MCS-FYI and MCS-SYI. The standard deviation of all IBDs at the DN scale was comparable to that of MCS-FYI, attributable to the sample difference and the early rapid increase. Considering estimates from different seasons independently (Table A3), the standard deviation at the regional scale was significantly higher compared to that of MCS-FYI and MCS-SYI. The relatively high standard deviation of the hydrostatic equilibrium-based IBDs in Hutchings et al. (2015) can be attributed to the inclusion of both level and deformed sea ice in their measurements, with a relatively high spatial sampling resolution. This may result in a failure of the hydrostatic equilibrium assumption at the local level. In contrast, the IBD results of this study represent the average state of the level ice layers across all buoy sites, resulting in significantly lower IBD variability compared to previous studies. Moreover, our analysis is focused on spatial scales of approximately 25 and 50 km radius, which facilitates the convergence of the hydrostatic equilibrium-based IBD estimation.

**For this point, the airborne observations of Jutila et al. (2022) provided a good example of the significant impact of spatial scale on IBD calculations.** Based on a ~30 km airborne measurement profile, the IBDs derived from sampling at 5−6 m spatial resolution were mainly between 800 and 1000 kg m$^{-3}$ with considerable spatial variability. When weighted averaging over the length scale of 800 m was used, the resulting IBD effectively converged to about 880 to 920 kg m$^{-3,}$ significantly reducing the potential non-physical variability of the retrieved IBDs at 5−6 m spatial resolution. We also applied the hydrostatic equilibrium method (also known as the freeboard/draft method) to calculate the IBDs of MCS-FYI and MCS-SYI using the seawater and snow bulk densities defined in Section 2.2.3. Despite the small mean differences, the results showed **considerable variability compared to those obtained by the weighing method**, with deviations of 7 ± 35 kg m$^{-3}$ for FYI and 1 ± 21 kg m$^{-3}$ for SYI, further highlighting the challenges of assuming hydrostatic equilibrium at local scales.

*Specific comments*

*Line 21: The phrasing suggests that this study considers deformations, brine, and gas inclusions in its estimates of ice density, but this does not appear to be accurate. It would be helpful to clarify this point. Impact of deformed ice, brine and air inclusions were not analysed in this study.*

**Reply:** Agreed, and as mentioned above, we have added the analysis of physical factors. **Revised to:** *"which are influenced by factors such as the age, temperature, brine, air intrusion, and growth rate of the sea ice."*

*Line 23: The term "Distributed Network scale" might not be clear to readers unfamiliar with the MOSAiC Distributed Network. Could you specify the scale or provide additional context to make it more accessible and inclusive? Line 24: The statement regarding ice density estimation is not entirely accurate. Ice density was calculated separately from a combination of buoy ice thickness and altimetry freeboard data and from ice coring data. It might be helpful to differentiate these methods more clearly.*

**Reply:** Thanks for the suggestion. **Revised to:** *" To estimate regional IBD, we combined sea ice thickness and snow depth observations from ice mass balance buoys, snow density measurements from snow pits, together with high-resolution along-track freeboard data from airborne laser scanning (ALS) and the Ice, Cloud, and land Elevation Satellite-2 (ICESat-2), at both the Distributed Network (DN) and L-site scales (~50 km and ~25 km radius from the research vessel Polarstern).Assuming hydrostatic equilibrium, regionally averaged IBD estimates were obtained for the buoy-deployed ice layers, including 11 sites in second-year ice (SYI) and 4 sites in first-year ice (FYI). The regional IBDs were then compared and evaluated with high-precision density measurements from FYI and SYI cores, the latter representing the local ice domain revisited at the Main Coring Sites (named MCS-FYI and MCS-SYI)."*

*Line 28: The assertion that MOSAiC ice floes predominantly consisted of second-year ice needs further support. (1) Several studies suggest otherwise, and (2) the term "predominantly" is vague in terms of areal fraction. Additionally, it would be useful to clarify what is meant by "MOSAiC ice floes"—how many floes are being referenced, and which ones specifically?*

**Reply:** We apologize for the misrepresentation. As mentioned above, we actually retrieved the IBDs that are representative of the buoy-deployed ice layers, which we thoroughly revised in the text. *MOSAiC ice floes* **are defined as** ice floes within 50 km of *Polarstern*.

*Line 32: The ratio of ice freeboard to ice thickness cannot serve as a direct measure of ice density since they are mathematically correlated. Density is typically used to convert ice freeboard to ice thickness. Meanwhile, such relationship may be complicated by the presence of deep snow or absence of accurate snow observations. The latter is directly related to the results of this study. Please clarify the interpretation here.*

**Reply:** The regression coefficients derived from the statistical relationships still have value in predicting hydrostatic equilibrium-based IBD, and we emphasized that parametric fit performance does not represent absolute robustness in the text, as discussed in **GC1**. At the same time, the parameterization based on ice freeboard-to-thickness ratio can be used for the so-called radar (laser) freeboard to thickness conversion to simultaneously estimate IBD and sea ice thickness from space, which can facilitate sea ice remote sensing (see **Fig. A10**). **Revised to:** *"On this basis, we developed several potential parameterizations for regional- and local-scale IBDs that are expected to be applicable throughout the freezing season, taking into account a variety of sea ice, snow, and environmental parameters."*

*Line 34: The statement that second-year ice (SYI) dominates the central Arctic Ocean contradicts existing studies, such as Tschudi et al. (2019, 10.5194/tc-14-1519-2020), which show that first-year ice (FYI) is the primary ice type in the Arctic. Could you provide more context or evidence to support this claim?*

**Reply:** Agreed, we removed this statement.

*Line 59: The referenced study from Timco and Weeks (2010) offer an overview of sea-ice properties but do not draw any specific conclusions about the heterogeneity or seasonal evolution of ice density. They only presented ice density at various temperatures (assuming zero gas volume) and discussed a broad range of density measurements without the kind of conclusions mentioned in your study. Could you clarify this?*

**Reply:** We would like to clarify that this review do discuss the internal structure and porosity of sea ice. *Timco, G. and Weeks, W.: **A review of the engineering properties of sea ice**, Cold Regions Science and Technology, 60, 107-129, https://doi.org/10.1016/j.coldregions.2009.10.003, 2010.*

*Line 63: The claim that sea ice density influences permeability, meltwater infiltration, and biogeochemistry is quite bold. The cited studies may show correlations, but it seems inaccurate to assert that density directly defines these processes. Permeability, for example, is largely governed by ice age, microstructure, and brine volume. Could this section be reframed to reflect the complexity of these relationships? Also, the term porosity is used in unconventional and inconsistent way. In most*

*cases it is used as equivalent to gas volume fraction (line 440), while typically it should combine both gas and brine volume fractions. Therefore, please explain how gas volume is influencing permeability.*

**Reply:** We rephrased the sentence, and we defined the ice porosity in the text as the sum of the brine and air volume fractions. **Revised to:** "*IBD is regarded as a pivotal parameter for thermodynamic and mechanical modelling of sea ice (Ono, 1967; Wang et al., 2021) and indirectly involved in key processes such as sea ice permeability, summer meltwater infiltration, and biogeochemical cycles in the Arctic Ocean (Perovich et al., 2021; Angelopoulos et al., 2022).*"

*Line 66: If by winter you mean a period from December to February, this is not accurate statement. Most referenced measurements were collected in winter and spring.*

**Reply:** We rephrased the sentence. **Revised to:** "*However, the scarcity of ice core sampling, particularly in the early ice growth period, limits the spatial representativeness of core-based density measurements (Timco and Frederking, 1996; Hutchings et al., 2015).*"

*Line 71: This (small errors of weighing) was known way before the referenced publication.*

**Reply:** We rephrased the sentence. **Revised to:** "*In terms of measurement accuracy and uncertainty, the hydrostatic weighing method proved to be the best at capturing natural variations in ice core density and reducing sampling constraints* **(Butkovich, 1955; Nakawo, 1983; Pustogvar and Kulyakhtin, 2016).** "

*Line 90: The robustness of the parameterizations is limited by the assumption of hydrostatic equilibrium, which has been shown to be less precise (Hutchings et al., 2015). For instance, Jutila et al. (2022) propose density values above 950 kg/m³ for relatively thin ice, but such values are unrealistic, as they have not been validated by in situ measurements. How do you account for these limitations?*

**Reply:** We would like to clarify that Hutchings et al. (2015) studied a ~100 m transect covering a large area of deformed ice. They claimed that the unrealistic density values observed could be due to the hydrostatic equilibrium assumption being inappropriate at these length scales in deformed ice. Jutila et al. (2022) also include many deformed units, which may partly explain the potentially high density of thin ice. In contrast, our focus is exclusively on level sea ice over tens of kilometers, where the hydrostatic equilibrium assumption is still valid, as details discussed in **GC1**. This explains why the standard deviation of our IBD results is significantly smaller than that of previous studies. **Revised to:** "*potential parameterizations.*" **Added:** "*In this study, we combined sea ice thickness and snow depth data from an IMB array (15 buoys), along-track total freeboard data from ALS and ICESat-2, and snow bulk density data from snow pits to retrieve the bulk density of level sea ice over tens of kilometers suitable for the hydrostatic equilibrium assumption (Section 3.1).*"

*Line 101: Most of laser scans were significantly smaller than the linear scale of tens of kilometres (Hutter et al., 10.1038/s41597-023-02565-6). Could you clarify the scale?*

**Reply:** We clarified the measurement scale of ALS data. **Revised to:** *"Moreover, repeated airborne laser scanning (ALS) measurements yielded high-resolution along-track total freeboard data (Hutter et al., 2023a,b), uncovering sea ice elevation characteristics ranging from the local floe (< 5 km from Polarstern) to regional transects of tens of kilometers, with a total of 64 helicopter fights from September 2019 and September 2020."*

*Line 132: The exclusion of melt ponds from the sampling is not straightforward, as refrozen melt ponds are a substantial part of the ice cover, covering around 20-30% area. And their smooth surface also influence the amount of snow which may accumulate above such ice type. Additionally, surface conditions are not the primary issue when it comes to hydrostatic balance. The representativeness of the ice thickness at the buoy sites is more important, and this is not adequately discussed. How does excluding ridges and deformed ice improve representativeness?*

**Reply:** According to the buoy installation reports and initial surveys, the IMBs were indeed deployed at sites initially devoid of melt ponds. In addition, temperature profiles recorded by the IMB did not show any anomalous signals that would indicate potential refrozen melt ponds. In terms of representativeness, the buoy deployment sites successfully recorded thermodynamically driven ice growth, effectively representing the level ice features in their deployment areas (Koo et al., 2021). We also acknowledge that the buoy deployment area contained many thick SYIs, and given the heterogeneity of the MOSAiC ice floes, it is difficult to fully represent DN scale ice conditions. **Therefore, we no longer claim in the text that buoy observations are representative of DN ice conditions. Revised to:** *"The SIMBAs and SIMBs were mainly deployed on level ice, free from melt ponds, to ensure the reliability of the buoy data".* **Added** *"In particular, buoy deployment sites have successfully recorded thermodynamically driven ice growth, effectively representing the level ice features in their deployment areas (Koo et al., 2021)."*

*Line 140: You state that 27% of the buoys were in FYI and 73% in SYI, but Kortum et al. (2024, 10.5194/tc-18-2207-2024) showed equal fractions of FYI and SYI based on freeboard measurements. How was this potential unrepresentativeness of buoy sites handled? Moreover, SYI is not typically associated with ice deformation when thinner FYI is present, which may influence evolution of its areal fraction.*

**Reply:** Our IBD results only represent the average sate across the buoy sites, as described above.

*Line 141: First, does this influence your results? Probably not. Also, exceptionally large ice growth should be quantified of important. Illustration shows that platelet ice presumably contributed to 10-20 cm ice growth which later disappeared. Is it relevant for this study?*

**Reply:** Agreed. **Removed** *"In particular, exceptionally large ice growth was observed at some sites (e.g., T72 and I2) due to the formation of snow-ice or platelet ice (Fig. S1). Katlein et al. (2020) provided direct evidence for platelet ice formation during MOSAiC based on remotely operated vehicle observations."*

*Line 173: There's an inconsistency here—earlier, you stated that 27% FYI was representative in an SYI-dominated area, yet here, 43% FYI is also considered representative. Meanwhile, Kortum et al. estimated this fraction as 50%. Could you clarify this discrepancy?*

**Reply:** Here, snow bulk density serves as an auxiliary data for IBD retrieval, with a relatively low error contribution (less than 5%). Our main objective is to extract its seasonal trend by analyzing all available snow pit data. **Revised to** *"The snow pit records used contained approximately 12 % refrozen lead, 43 % FYI, and 45 % SYI, according to the original snow pit metadata and MOSAiC ice floe characteristics (Macfarlane et al., 2021; Nicolaus et al., 2022),* **effectively reflecting the general variability in snow properties under different ice conditions during MOSAiC***."*

*Line 179: It would be more robust to demonstrate the representativeness of the selected sites by comparing their ice thickness or freeboard with large-scale observations, rather than just stating they were selected for representativeness.*

**Reply:** Agreed. **Removed** *"and were sufficiently representative of the FYI and SYI properties during MOSAiC".*

*Line 182: Please clarify what is meant by "ice porosity." What does it include?*

**Reply:** "Ice porosity" **replaced** with "brine volume". In this study, ice porosity was considered as the sum of brine and gas volume fractions.

*Line 183: According to Oggier et al. (2023), salinity was measured directly from the sample after weighing, not interpolated from another core. Also, could you explain how brine volume and density were estimated for the measured temperatures?*

**Reply:** We would like to clarify that the density values are indeed interpolated to the depth of salinity measurements, as described in Oggier et al. (2023). *"The depth estimates assume that the total length*

*of all core sections is equal to the measured ice thickness. Each core section has the value of its practical salinity (14), isotopic values (16, 17, 18) (Meyer et al., 2000), as well as sea ice temperature (19) and ice density (20) interpolated to the depth of salinity measurements."*

**Modified and added:** *"Of these, ice temperature was measured in the field using Testo 720 thermometers at 5 cm vertical intervals; ice salinity was determined from melted cores using a YSI 30 conductivity meter at 5 cm resolution; ice density was derived in the freezing laboratory at an ambient temperature of –15 °C using the hydrostatic weighing method. The temperature and density results were further interpolated to match the depth of the salinity measurements, providing a continuous density profile throughout the core sample. Then, brine volume was calculated from measured ice salinity, temperature, and density using the equations of Cox and Weeks (1983) for cold ice and Leppäranta and Manninen (1988) for ice warmer than –2 °C."*

*Line 189: The statement that some cores were longer and might come from rafted ice, leading to substantially different densities, is not supported by the data from Oggier et al. (2023). Please provide evidence for this assertion.*

**Reply:** We would like to clarify that the FYI cores from 2020-03-21 and 2020-04-14 are labelled as rafted ice. Our aim is to focus solely on the variations in ice properties under thermodynamic growth of the level ice and their comparison with regional results. **Here, we removed the statement of substantially different densities. Revised to** *"To ensure reasonable comparability of IBD estimates between ice cores and regional estimates for the level ice layer, core-based density records of rafted ice associated with ice deformation were excluded."*

*Line 268: The ice pack does not consist solely of level ice. Please rephrase this statement, as ridges and deformed ice make up a substantial portion of the ice cover, as shown in transect observations. Also, how can you be certain that modal snow freeboard represents the same level ice described at the buoy sites? A simple comparison of ice thickness distribution or mean, modal and median values could be also useful. What if ice consists of both FYI and SYI level ice, with a bimodal thickness distribution, which is not seen in snow freeboard data?*

**Reply:** *The ice pack does not consist solely of level ice.* **Revised to** *"The MOSAiC ice floes have considerable spatial heterogeneity, covering ice ridges, deformed ice, level ice, and refrozen melt ponds (Nicolaus et al., 2022). To estimate the IBD accurately, it is first needs to obtain the total freeboard that is representative of the level ice portions to ensure compatibility with the selected IMBs, which were deployed exclusively on level ice."*

*Also, how can you be certain that modal snow freeboard represents the same level ice described at the buoy sites?* This is why we made the spatial scale adjustment mentioned above.

*A simple comparison of ice thickness distribution or mean, modal and median values could be also useful. What if ice consists of both FYI and SYI level ice, with a bimodal thickness distribution, which is not seen in snow freeboard data?* We would like to clarify that based on the high-resolution IceBird airborne measurements, which include both level and deformed FYI, SYI and MYI, we found that the modal total freeboards of all ice types are very close to the mean total freeboards of the level ice fractions only (Fig. A14). In addition, the total freeboard distributions obtained from ICESat-2 and ALS both showed distinct unimodal characteristics, as did the ice thickness distributions derived from the transects. Several studies have found that the typical regional characteristics of sea ice freeboard/thickness conform to log-normal/negative exponential distributions (Haas, 2010; Farrell et al., 2011; Ricker et al., 2015; Petty et al., 2016; Landy et al., 2019; Koo et al., 2021), despite the inclusion of both deformed and level ice, as well as first-year and multi-year ice.

[Figure]

**Figure A14.** Comparison of the mode of the total freeboard distribution (encompassing all surface types) with the mean total freeboard of level ice, derived from AWI IceBird measurements conducted in April 2017 and April 2019. The example from 2 April 2017 illustrates: **(a)** the total freeboard distribution, **(b)** the measurement area, and **(c)** the total freeboard profile. **(d)** Results for each measurement date include the modal freeboard for all surface types, mean freeboard for level ice, and the relative percentage difference (RPD) between the two freeboard values, as well as the level ice fraction (at least 20%).

*Line 287: What are "MOSAiC ice floes, type II"? Could you provide more information on this classification? If you refer to the types of uncertainties, then they are described after this part of the manuscript.*

**Reply:** Clarified. **Revised to:** "*1) ascertaining the modal total freeboard, which is derived from the mean of the five highest frequency peaks in the total freeboard distribution, with the standard deviation quantifying the **uncertainty (Type I)** introduced by these peaks. 2) This mean difference was also considered to represent the **uncertainty (Type II)** of the modal total freeboard, stemming from the measurement methods and the spatial heterogeneity of the MOSAiC ice floes. 3) The final modal total freeboard uncertainty was then set as t**he sum of the Type I and Type II uncertainties**.*"

*Line 291: The 7 cm adjustment is a significant point, yet it is only briefly mentioned. Could you explain in more detail how this value was estimated and why it was applied as a time-independent adjustment throughout the seven-month period?*

**Reply:** We clarified this in **GC2**, see above. **Added:** "*2.2.2 Spatial scale adjustment*".

*Line 294: How is this possible if you only use a single source of snow thickness from buoys? Here you attempt to match ice freeboard calculated as difference between ICESat-2 snow freeboard and snow thickness from buoys. You do not have data about large-scale snow thickness (line 314). Therefore, you lift the snow and ice freeboards together which leads to increased estimate of ice thickness which fits better the initial buoy ice thickness. From these estimates of ice freeboard and thickness you estimate the ice density. Therefore, it is not clear what do you mean by the snow depth difference between satellite and buoys sites, as second one is unknown. Similarly, this is an important assumption which cannot even be generalized for future campaigns. It deserved not to be described in supplementary materials.*

**Reply:** We clarified this in **GC1** and **GC2.**

*Line 315: While seawater density might not be the primary source of uncertainty, why not use directly measured seawater salinity and temperature from the expedition? Why was the chosen reference used instead?*

**Reply:** The selected literature is very representative and has been widely used in the hydrostatic equilibrium equation for Arctic sea ice. We also found that the fixed value is in good agreement with the mean density estimate with the upper 10 m layer recorded by buoys and CTDs during MOSAiC (Hoppmann et al., 2022). In the range of their estimated densities (~1023.5 to 1026.5 kg m$^{-3}$), the effect on our IBD results does not exceed 0.5 kg m$^{-3}$ (0.05 %). **Overall, the use of representative value remains valid.**

*Line 317: Again, more evidence is needed to support the claim that the MOSAiC Distributed Network (DN) was dominated by SYI. How this is possible when the measured by ICESat-2 snow freeboard in autumn was close to 10 cm? Was it representative for SYI, especially SYI similar to ice where buoys were installed? Because buoys had ice freeboard around 10 cm and snow thickness of 13 cm, giving the snow freeboard way above 10 cm.*

**Reply:** Please refer to our answers in the **GC2. Revised to:** *"This method provides an estimate of the regional bulk density of level sea ice mixed with SYI (11 buoy sites) and FYI (4 buoy sites)."*

*Line 318: Krumpen et al. (2020, 10.5194/tc-14-2173-2020) report that the modal ice thickness in L-sites was only 40-50 cm, significantly thinner than at buoy sites. Why is this discrepancy not addressed?*

**Reply:** We addressed this in **GC2**. We highlighted in the text the significant differences in thickness and snow depth of level sea ice between the buoy sites and earlier transects during MOSAiC.

*Line 359: The ice thickness range of 0.6–2.0 m is stated as representative for level ice, yet Krumpen reports much thinner modal thicknesses. Early transects from Itkin et al. (2023,10.1525/elementa.2022.00048) also mainly show modal ice thickness around 0.5 m. Could you provide clarification on this?*

**Reply:** Agreed. **Revised to:** *"The mean ice thicknesses recorded by the buoys at the DN and L-site scales were 1.41 ± 0.32 and 1.30 ± 0.33 m, respectively (Fig. 5a and Table 2), whereas the average values for individual buoys ranged from ~0.6 to 2 m (Fig. S1). Due to their deployment strategy, buoy-derived sea ice thicknesses were generally thicker than the regional transects in the early stages (Krumpen et al., 2020, Table 1)"*

*Line 370: Arbitrary selection of coring sites does not necessarily make them representative in terms of ice thickness. This should be better supported.*

**Reply:** We removed that statement.

*Line 375: You obtain level ice freeboard using modal values of snow freeboard. Why is a different approach applied here to the transect snow thickness? Itkin et al. (2023) showed a snow thickness distribution with initial modal snow thickness around 10 cm. Similarly, the initial snow thickness above*

*level ice in their paper is also close to 10 cm. Therefore, what is the meaning of the initial average 24 cm from transects?*

**Reply:** We recalculated the modal snow depths for the transects and reported the differences with the buoys, see **Fig. A12**.

*Line 377: It appears that specific conditions at the coring sites resulted in shifting the ICESat-2 freeboard upwards. However, ICESat-2 and ALS initially showed freeboards close to 10-15 cm, suggesting the buoys were in areas with unrepresentative snow thickness. Could this be clarified?*

**Reply:** We highlighted and clarified in the text the differences in ice conditions between buoy sites and large-scale observations, which accounts for the spatial scale adjustment, see **GC2**. **Revised to** *"The snow depths from the transects (mode), MCS-FYI, and MCS-SYI were generally thinner than the buoy array observations, with mean values of 0.15 ± 0.02 , 0.10 ± 0.03, and 0.14 ± 0.03 m, respectively, further suggesting that thicker level ice tends to accumulate thicker snow."*

*Line 384: This is a misinterpretation of the study from Itkin et al. (2023) You claim that buoy snow depth was close to transects. Yet, in the Fig. 9 of Itkin et al. (2023) the shown snow depth above level ice is below 10 cm.*

**Reply:** We would like to clarify that our conclusion is that the buoys have similar seasonal snow depth trends to the transects, not their snow depth magnitudes. **Revised to** *" Throughout the freezing season, the regional buoy averages, as well as the transects and MCS-FYI, all exhibited significant trends of increasing snow depth, ranging from 0.010 to 0.013 m per month (Table 2)."* **Original text:** *"buoy array measurements are sufficiently representative **to capture seasonal snow accumulation on level ice over a wider area"**.*

*Line 437: You mention that the DN is dominated by SYI, but later state that MOSAiC ice floes were predominantly FYI. This contradicts previous observations (e.g., Guo et al., 10.5194/tc-17-1279-2023; Kortum et al.). Could you clarify what exactly is being referred to as "MOSAiC ice floes"?*

**Reply:** We no longer claim that any of the IBD results represent MOSAiC ice floes (which we denote as a 50 km radius centred on Polarstern), but simply the buoy-deployed ice layers, as detailed in GC1 and **GC2**. **Revised to** *"The regional IBD estimates, including both DN and L-site scales, **reflect the ice properties at the corresponding buoy deployment sites**, instead of the ice conditions over the entire regional range for DN or L-site scales."*

*Line 437: Your estimate of ice density includes 7 cm freeboard adjustment; how can you discuss these minor differences between measured FYI and SYI density considering buoy method uncertainty close to 50 kg/m³? Moreover, there is not any clear difference in winter FYI and SYI density based on weighing. How can potential difference in several kg/m³ can support whether FYI or SYI was dominating the area of investigations? You are saying that 910 kg/m³ is closer to 912 kg/m³, while 908 is closer to 905 kg/m³? A difference in 2 kg/m³ is beyond measuring uncertainties and spatial variability even for the best performing methods to measure ice density. No such claims can be made. Finally, it is not correct to average FYI density for the whole period of observation, as it includes a period of lower densities in autumn. If you exclude those lower values from the average value of 905 kg/m³, would you be able to see any measurable difference between two ice types?*

**Reply:** We fully revised this. **Revised to** *"The mean IBD values estimated at the DN, L-site, MCS-FYI, and MCS-SYI scales were 910 ± 7, 908 ± 11, 905 ± 8, and 912 ± 2 kg m$^{-3}$, respectively. The recorded bulk densities of FYI and SYI cores were not obviously different except in the early stages."*

*Line 443: in line 437 you claim that DN is dominated by SYI, while here you claim that MOSAiC ice floes were predominantly FYI. What are MOSAiC ice floes? This contradicts many other observations (Guo et al., Kortum et al.). MOSAiC Central Observatory was one of the thickest floes in the area with the largest SYI fraction, unlike surroundings (Krumpen et al., 2020), which is opposite to the presented claims.*

**Reply:** We removed this statement. As mentioned above, our regional IBD results only represent the physical properties of buoy-deployed ice layers (dominated by SYI).

*Line 446: What is the basis of the claim of substantial sub-weekly ice density variability. This is a clear artefact of spatial variability of ICESat-2 tracks and corresponding freeboards which has nearly nothing to do with ice density changes.*

**Reply:** We acknowledge that trajectory changes in ALS/IS2 can introduce non-physical variations in modal total freeboard that affect the derived IBD. It is also critical to note that the covariation between ALS/IS2 modal freeboard and buoy-derived mean snow depth demonstrates high consistency, which may also be attributable to variations in snow depth (**Fig. A15**). **Added:** *"Due to the irregular variations in airborne and satellite trajectories and the limitations of the buoys in capturing daily variations of mean snow depth over their deployment area, it must be emphasized that the retrieval method may not be able to fully resolve synoptic scale IBD variations and thus introduce some non-physical signals, which is a major challenge for estimating IBD using spatially non-overlapping data."*

[Figure]

**Figure A15.** Buoy-derived mean snow depth versus airborne/satellite-derived modal total freeboard during the MOSAiC freezing season at both DN and L site scales.

*Line 448: You are saying that FYI had newly formed SYI fractions at the ice bottom? What does it mean?*

**Reply:** Remove this statement and we added the sea ice bulk salinity and brine volume data. **Revised to** *"The bulk salinity of MCS-SYI, on the other hand, showed an increasing trend of 0.3 ppt per month, which is closely associated with the formation of new ice layers at the base"*

*Line 451: First, which layers are you referring to? These layers may contain more brine than which other layers? For FYI, how is it possible, if ice is gradually getting colder and less saline upon ice growth? The brine volume is easy to estimate, and it was estimated in several publications, there is no need for such speculations. And, following the modelling study from Griewank and Notz (2008), brine volume is decreasing in Autumn, contributing to decreasing density if gas is not considered. And if you discuss SYI, there were no new layer under SYI in autumn, only in winter, which can be clearly seen in Fig. 4a. Only since cold winter month, SYI with larger snow and ice thickness may have a substantial thickness gain.*

**Reply:** As answered in **GC1**, we re-analyzed the seasonal variations in IBD, which were attributed to gas reduction, decreasing bulk temperatures, and rapid growth of the sea ice over the same period.

*Line 461: What does it mean? There is a clear relationship between temperature, salinity, and brine volume which explain why saltier FYI has a stronger vertical gradient of brine volume. Also, most of SYI formed under the remaining 0.5 m at the surface is a new ice, with its physical properties close to FYI. Therefore, there is no surprise that FYI and SYI are eventually becoming more similar. Homogenization is a too broad description of these processes.*

**Reply:** Remove this statement. We added analyses of the seasonal evolution of sea ice bulk temperature, bulk salinity, and brine volume (see Fig. A8).

*Line 462: What is the evidence for these claims? You do not present ice salinities in this study. But if you check FYI bulk salinity from Oggier et al. (2023), it has decreasing trend very similar to the broadly used parametrization from Kovacs (1996, 10.21236/ada312027). Also, please explain in more detail the suggested process behind the initial density decrease. If you consider only brine, desalination leads to decrease of brine volume and corresponding density decrease, that is correct. But it cannot be significant only due to brine, the absolute values are less than 5 kg/m³. One can estimate brine volumes from the measured salinity and temperature and draw the same conclusion. Yet, the observed density change is much larger, and it was increasing, not decreasing (Fig. 5a).*

**Reply:** As answered in **GC1**, we re-analysed the seasonal variations in IBD, which were attributed to gas reduction, decreasing bulk temperatures, and rapid growth of the sea ice over the same period.

*Line 466: This is an example of overfitting. Density estimates from ICESat-2 has a large spread of around 30 kg/m³. The increasing trend in autumn is barely based on a single data point in Fig. 5a.*

**Reply:** We combined the regional IBD estimates from late Oct. To early Dec. with a statistical test P value of less than 0.05 and an *R* of 0.62, so there was no overfitting (see Fig. A2). For more details, please refer to **Overall Reply**. We also acknowledge the relatively high uncertainty and small sample in the early stages and therefore additionally introduced the so-called modelled IBD values (see Fig. A13).

*Line 498: Can you explain more specifically what is the mentioned scale-related density variability? You cover different areas in DN and L-site scales, but there is no clear difference between them following Fig. 5a. Also, the mean values are nearly identical, but this would not be possible without in situ measurements, providing less uncertain estimates. And what supports the well-known importance of ice density for ice thickness retrieval but based purely on your measurements? How do you validate thickness estimates?*

**Reply:** In the revised version, we highlighted the scale-related density variability between the MCS-SYI and regional buoy sites (SYI dominated). We argue that the regional IBD results are still of

value to the sea ice remote sensing community, which was attempted in the MOSAiC observations after A10 (Sever) and J22 (IceBird), albeit with significant challenges. The supplementary validation of the parameterization and its preliminary application across the pan-Arctic SYI have provided us with considerable confidence, please refer to **GC1**.

*Line 520: Repeating this statement doesn't make it more evident. Without scale adjustment, the density estimate from ICESat-2, and buoys not only show an opposite decreasing seasonal trend to in situ measurements, but they also give unrealistic density values above 950 kg/m³. Therefore, it is not clear why this difference in correlation should be compared between different methods.*

**Reply:** Removed.

*Line 548: Previously, it was mentioned that seasonality of ice density is significant. Yet, none of the suggested parameters are related to seasons. **Ice thickness has a large range from thin ice in Atlantic sector to thick ice in Pacific sector during the same sampling time.** Therefore, ice freeboard cannot be an accurate factor related to the seasonality of ice density.*

**Reply:** As we mentioned above, the samples used for IBD parameterization themselves exhibited time-varying properties, such as ice porosity, temperature, salinity, thickness, and age. Therefore, we suggest that IBDs estimated from the parameterizations are still capable of exhibiting potential seasonal variations, as shown in **Fig. A10**. We also added the sea ice bulk temperature to enhance the model interpretability. Following the potential applicability ranges, spatial scales and sea ice types of the parameterizations, we suggest that it is still possible to reasonably predict IBD for different thicker and thinner ice layers in the same season.

*Line 580: What is the basis of this claim? A typical modal ice thickness is around 1.7 m (Sumata et al., 2023) which includes both FYI and SYI. Why FYI cannot have 0.8-1.8 m thickness, only SYI? Also, there is a substantial difference in ice thicknesses wit thinner FYI in the start of Transpolar Drift and older ice closer to Canadian coast, where ice thickness is also typically larger than 0.8-1.8 m.*

**Reply:** We removed this statement.

*Line 627: If you claim that L-site ice was FYI and SYI, while DN was mainly SYI, why then snow freeboard from ICESat-2 and ALS showed such a great agreement despite ALS covering only a relatively small area around MOSAiC ice floe? Also, if most of ice was SYI, why ice density seasonality is close to the FYI in situ measurements, not SYI with no seasonality?*

**Reply:** Removed. In fact, the modal total freeboard derived from ICESat-2 at the DN scale, ICESat-2 at the L-site scale, and ALS at the L-site scale are similar to each other. In the revised version, we only highlighted that the results represent buoy-deployed ice layers, as described above. We attribute the proximity of SYI-dominated regional IBDs to FYI field measurements to the high spatial heterogeneity of SYI, as detailed in **GC1**.

*Line 675: The study from Kovacs does not properly describe methodology of density measurements. You describe it as "theoretical equation", what does it mean? There is a few better documented ice density reports, why this specific study was used?*

**Reply:** We removed this statement. The K97 parameterization has been used in several sea ice thickness retrievals (Kwok and Cunningham, 2015), so it is worth comparing our parameterizations with K97 to identify uncertainties, especially since K97 has opposite characteristics to the MOSAiC coring results.

*Line 680: What is the basis for such claims? There are in situ measurements of ice density for relatively thin ice (20-30 cm), and those values are mostly around 900-910 kg/m³ (10.1016/0165-232X(95)00007-X), which is close to MOSAiC values. Also, MOSAiC ice was quite thin, around 40 cm. Sampling ice thinner than 30 cm is typically considered unsafe, and there is a very short time when ice is that thin in Central Arctic, as thin ice growth exponentially faster.*

**Reply:** Removed.

*Line 682: Similarly, why a method based on substantial adjustments to be able to get realistic estimates of ice density can support limitations of more accurate method, based on measurements of ice density? What exactly was done to support this statement?*

**Reply:** We clarified this statement. **Revised to** *"Overall, we emphasize that the IBD parameterization based on sampling from individual points has the spatial representativeness-related limitations for the application to satellite sea ice thickness retrievals. In contrast, the regional IBD parameterisations are helpful in adapting to satellite footprint scales, but have relatively high uncertainties compared to direct measurements."*

*Line 697: The was not any seasonality in SYI density estimates. A strong seasonality and corresponding dependence of ice density from its thickness was only observed for FYI. Therefore, why the parametrization based on FYI measurements can optimize retrieval of SYI? This should be better explained.*

**Reply:** As discussed above, the IBD differences between SYI-dominated buoy sites and local SYI cores suggest a possible high degree of spatial heterogeneity in SYI porosity. Thus, the parameterizations derived from the SYI-dominated buoy sites are still capable of exhibiting seasonal variations for IBD. This, in turn, illustrates the impact of spatial scale. *In particular, Hornnes et al. (2024) found that the spatial heterogeneity of SYI porosity was much greater than that of FYI during the GoNorth expedition in October 2022, and also found similar air fractions in FYI (2.5 ± 0.5 %) and SYI cores (2.8 ± 0.5 %).*

*Line 707: This is generally surprising that SYI is treated as a separate ice type from FYI. Maybe it could be reasonable many years ago, but for MOSAiC observations with SYI having low-salinity remnant layer below 40-80 cm (Oggier et al., 2023), most of SYI thickness by the end of winter is FYI grown under SYI, which may have density values like pure FYI. Since all such data is available, it is worth discussing. In brief, MOSAiC SYI was much closer to FYI than SYI sampled by Jutila et al., with at least two times higher thicknesses.*

**Reply:** We argue that the physical properties of SYI after experiencing summer melt generally differ significantly from those of FYI, such as bulk salinity, brine volume, and thickness, and therefore remain worth distinguishing, also shown in Salganik et al. (2024). We agree with the reviewer's mention of *low-salinity remnant layer* and added descriptions of ice core salinity and brine volume accordingly. We also highlighted the difference between SYI during MOSAiC and J22.

*Line 734: You previously mentioned that without scale adjustments your estimates would be 50 kg/m³, therefore you should explain how this bias should be removed for future analysis prior to providing parametrization based on adjusted estimates.*

**Reply:** We added suggestions on how to overcome methodological challenges in the future.

*Line 738: You haven't provided a strong proof that the provided parametrization with a strong seasonality in autumn is related to SYI, not FYI. Observations suggest that this is different.*

**Reply:** We would like to clarify that the regional IBDs derived from buoy sites predominantly featuring SYI indeed exhibited variations similar to those observed in MCS-FYI. By setting representative regional values and applying a trend assumption, an early growth trend in IBD was also observed. Consequently, we believe this is related to the spatial heterogeneity of SYI porosity, which has been well-documented by Hornnes et al. (2024).

*Line 742: It is generally looks impossible that estimates from hydrostatic balance have smaller standard deviations than measurements from weighing. First, even in situ measurements of freeboard and thickness give much higher deviations of density as described by Hutchings et al. (10.3189/2015AoG69A814) Second, density of separate layers of ice should not be compared with bulk density.*

**Reply:** Please see our specific answers in **GC3** for more details.

*Line 743: This is one of the most inconsistent claims throughout the whole paper. Can you say specifically what exactly was dominated by FYI and SYI and provide some evidence for this.*

**Reply:** We comprehensively revised the manuscript to clarify that the IBD results pertain solely to the ice layers where the buoys were deployed. The spatial scale adjustment was made to align the modal total freeboard derived from ALS/IS2 with the buoy sites. Therefore, interpretations for IBD that primarily involve SYI are justified and reasonable.

*Line 748: The assertion that warm air intrusions substantially influenced future changes in ice density evolution is significant but not discussed earlier in the text. Could this be elaborated?*

**Reply:** Added discussion related to sea ice permeability.

*Line 756: The criticism of in situ density measurements requires more analysis. How can 15 buoys at the fixed sites capture a broader picture better than around 40 coring events at slightly different sites? Additionally, how does the smaller uncertainty from weighing lead to challenges, as this seems counterintuitive?*

**Reply:** In our revisions, we no longer claim superiority of any methods or parameterizations. Instead, we focus on comparing the difference between the regional (buoy sites, hydrostatic equilibrium method) and local results (coring sites, weighing). Please see our specific answers in **GC1** and **CG3** for more details.

**Reference**

Ackley, S.F., Hibler, W.B., Kugrzuk, F., Kovacs, A., Weeks, W.F., 1974. Thickness and roughness variations of arctic multi-year ice. Oceans '74, IEEE International Conference on Engineering in the Ocean Environment, vol. 1, pp. 109 – 117. Halifax, NS, Canada.

Alexandrov, V., Sandven, S., Wahlin, J., and Johannessen, O.: The relation between sea ice thickness and freeboard in the Arctic, The Cryosphere, 4, 373-380, 2010.

Angelopoulos, M., Damm, E., Simões Pereira, P., Abrahamsson, K., Bauch, D., Bowman, J., Castellani, G., Creamean, J., Divine, D. V., and Dumitrascu, A.: Deciphering the properties of different arctic ice types during the growth phase of MOSAiC: Implications for future studies on gas pathways, Frontiers in Earth Science, 10, 864523, 2022.

Butkovich, T. R.: Density of Single Crystals of Ice from a Temperate Glacier, Journal of Glaciology, 2, 553-559, 1955.

Cox, G. F. N. and Weeks, W. F.: Equations for Determining the Gas and Brine Volumes in Sea-Ice Samples, Journal of Glaciology, 29, 306-316, 1983.

Crabeck, O., Galley, R., Delille, B., Else, B., Geilfus, N. X., Lemes, M., Des Roches, M., Francus, P., Tison, J. L., and Rysgaard, S.: Imaging air volume fraction in sea ice using non-destructive X-ray tomography, The Cryosphere, 10, 1125-1145, 2016.

Crabeck, O., Galley, R. J., Mercury, L., Delille, B., Tison, J.-L., and Rysgaard, S.: Evidence of Freezing Pressure in Sea Ice Discrete Brine Inclusions and Its Impact on Aqueous-Gaseous Equilibrium, Journal of Geophysical Research: Oceans, 124, 1660-1678, 2019.

Farrell, S. L., Kurtz, N., Connor, L. N., Elder, B. C., Leuschen, C., Markus, T., McAdoo, D. C., Panzer, B., Richter-Menge, J., and Sonntag, J. G.: A first assessment of IceBridge snow and ice thickness data over Arctic sea ice, IEEE Transactions on Geoscience and Remote Sensing, 50, 2098-2111, 2011.

Haas, C.: Dynamics versus thermodynamics: The sea ice thickness distribution, Sea ice, 82, 113-152, 2010.

Hoppmann, M., Kuznetsov, I., Fang, Y. C., and Rabe, B.: Mesoscale observations of temperature and salinity in the Arctic Transpolar Drift: a high-resolution dataset from the MOSAiC Distributed Network, Earth Syst. Sci. Data, 14, 4901-4921, 2022.

Hornnes, V., Salganik, E., and Høyland, K. V.: Relationship of physical and mechanical properties of sea ice during the freeze-up season in Nansen Basin, Cold Regions Science and Technology, 2024. 104353, 2024.

Hornnes, V., Salganik, E., and Høyland, K. V.: Relationship of physical and mechanical properties of sea ice during the freeze-up season in Nansen Basin, Cold Regions Science and Technology, 229, 104353, 2025.

Hutchings, J. K., Heil, P., Lecomte, O., Stevens, R., Steer, A., and Lieser, J. L.: Comparing methods of measuring sea-ice density in the East Antarctic, Annals of Glaciology, 56, 77-82, 2015.

Jutila, A., Hendricks, S., Ricker, R., von Albedyll, L., Krumpen, T., and Haas, C.: Retrieval and parameterisation of sea-ice bulk density from airborne multi-sensor measurements, The Cryosphere, 16, 259-275, 2022.

Koo, Y., Lei, R., Cheng, Y., Cheng, B., Xie, H., Hoppmann, M., Kurtz, N. T., Ackley, S. F., and Mestas-Nuñez, A. M.: Estimation of thermodynamic and dynamic contributions to sea ice growth in the Central Arctic using ICESat-2 and MOSAiC SIMBA buoy data, Remote Sensing of Environment, 267, 112730, 2021.

Kovacs, A.: Estimating the full‐scale flexural and compressive strength of first‐year sea ice, Journal of Geophysical Research: Oceans, 102, 8681-8689, 1997.

Krumpen, T., Birrien, F., Kauker, F., Rackow, T., von Albedyll, L., Angelopoulos, M., Belter, H. J., Bessonov, V., Damm, E., and Dethloff, K.: The MOSAiC ice floe: sediment-laden survivor from the Siberian shelf, The Cryosphere, 14, 2173-2187, 2020.

Kwok, R. and Cunningham, G.: Variability of Arctic sea ice thickness and volume from CryoSat-2, Philosophical Transactions of the Royal Society A: Mathematical, Physical and Engineering Sciences, 373, 20140157, 2015.

Landy, J. C., Tsamados, M., and Scharien, R. K.: A Facet-Based Numerical Model for Simulating SAR Altimeter Echoes From Heterogeneous Sea Ice Surfaces, Ieee Transactions on Geoscience and Remote Sensing, 57, 4164-4180, 2019.

Leppäranta, M. and Manninen, T.: The brine and gas content of sea ice with attention to low salinities and high temperatures, 1988. 1988.

Nakawo, M.: Measurements on air porosity of sea ice, Annals of Glaciology, 4, 204-208, 1983.

Ono, N.: Specific heat and heat of fusion of sea ice, Physics of Snow and Ice: proceedings, 1, 599-610, 1967.

Perovich, D., Smith, M., Light, B., and Webster, M.: Meltwater sources and sinks for multiyear Arctic sea ice in summer, The Cryosphere, 15, 4517-4525, 2021.

Petrich, C. and Eicken, H.: Overview of sea ice growth and properties, Sea ice, 2017. 1-41, 2017.

Petty, A. A., Tsamados, M. C., Kurtz, N. T., Farrell, S. L., Newman, T., Harbeck, J. P., Feltham, D. L., and Richter-Menge, J. A.: Characterizing Arctic sea ice topography using high-resolution IceBridge data, The Cryosphere, 10, 1161-1179, 2016.

Pustogvar, A. and Kulyakhtin, A.: Sea ice density measurements. Methods and uncertainties, Cold Regions Science and Technology, 131, 46-52, 2016.

Ricker, R., Hendricks, S., Perovich, D. K., Helm, V., and Gerdes, R.: Impact of snow accumulation on CryoSat‐2 range retrievals over Arctic sea ice: An observational approach with buoy data, Geophysical Research Letters, 42, 4447-4455, 2015.

Salganik, E., Crabeck, O., Fuchs, N., Hutter, N., Anhaus, P., and Landy, J. C.: Impacts of air fraction increase on Arctic sea-ice thickness retrieval during melt season, EGUsphere, 2024, 1-30, 2024.

Salganik, E., Hornnes, V., Rübsamen-Von Döhren, P. J., Panchi, N., and Høyland, K. V.: First- and second-year sea-ice salinity, temperature, and density during GoNorth 2022 leg 1. PANGAEA, 2023.

Shi, H., Lee, S.-M., Sohn, B.-J., Gasiewski, A. J., Meier, W. N., Dybkjær, G., and Kim, S.-W.: Estimation of snow depth, sea ice thickness and bulk density, and ice freeboard in the Arctic winter by combining CryoSat-2, AVHRR, and AMSR measurements, IEEE Transactions on Geoscience and Remote Sensing, 2023. 2023.

Timco, G. and Frederking, R.: A review of sea ice density, Cold regions science and technology, 24, 1-6, 1996.

Wang, Q., Zong, Z., Lu, P., Zhang, G., and Li, Z.: Probabilistic estimation of level ice resistance on ships based on sea ice properties measured along summer Arctic cruise paths, Cold Regions Science and Technology, 189, 103336, 2021.

*"Our ultimate goal is to advance sea ice remote sensing; despite the methodological challenges, we have at least considered all available observational data and potential solutions for implementation."*

- Snow depth and sea ice thickness data derived from SIMBA buoy measurements are available from PANGAEA: https://doi.org/10.1594/PANGAEA.938244
- Snow depth, sea ice thickness, air temperature, sea ice bulk temperature, and seawater temperature data derived from SIMB buoy measurements are available from the Arctic Data Center: https://doi.org/10.18739/A20Z70Z01 (added)
- Snow depth and sea ice thickness data from repeated transects are available from PANGAEA: https://doi.org/10.1594/PANGAEA.937781
- Total thickness data measured using the GEM-2 from repeated transects are available from PANGAEA: https://doi.org/10.1594/PANGAEA.943666
- Snow pit data collected during the MOSAiC expedition are available from PANGAEA: https://doi.org/10.1594/PANGAEA.940214
- Snow pit raw data collected during the MOSAiC expedition are available from PANGAEA: https://doi.org/10.1594/PANGAEA.935934
- Total freeboard data obtained from airborne laser scanning measurements during MOSAiC are available from PANGAEA: https://doi.org/10.1594/PANGAEA.950896
- Ice core data collected from the MOSAiC Main Coring Sites are available from PANGAEA: https://doi.org/10.1594/PANGAEA.956732(MCS-FYI) (added temperature, salinity, and brine volume)
  https://doi.org/10.1594/PANGAEA.959830 (MCS-SYI) (added temperature, salinity, and brine volume)
- Ice core data collected during the GoNorth expedition in late October 2022 are available from PANGAEA: https://doi.org/10.1594/PANGAEA.962567 (added)
- ICESat-2 ATL10 total freeboard data (version 6, latest version) are available from NSIDC: https://doi.org/10.5067/ATLAS/ATL10.006
- AWI IceBird airborne multi-sensor sea ice data (version 2, latest version) collected in April 2017 and 2019 are available from PANGAEA: https://doi.org/10.1594/PANGAEA.966009 (April 2017) and https://doi.org/10.1594/PANGAEA.966057 (April 2019).
- AWI CryoSat-2 gridded sea ice product from 2010 to 2024 (v2p6, latest version) are available from AWI: ftp://ftp.awi.de/sea_ice/product/cryosat2/v2p6/nh/ (added)
- Weekly gridded sea ice age data are available from NSIDC (version 4, latest version): https://doi.org/10.5067/UTAV7490FEPB (added)
  https://doi.org/10.5067/2XXGZY3DUGNQ (quick look)
- Temperature profiles derived from SIMBA T47 are available from PANGAEA: https://doi.pangaea.de/10.1594/PANGAEA.940387 (added)
- Temperature profiles derived from SIMBA T58 are available from PANGAEA: https://doi.pangaea.de/10.1594/PANGAEA.940393 (added)
- Temperature profiles derived from SIMBA T62 are available from PANGAEA: https://dx.doi.org/10.1594/PANGAEA.940231 (added)
- Temperature profiles derived from SIMBA T63 are available from PANGAEA: https://doi.pangaea.de/10.1594/PANGAEA.940593 (added)
- Temperature profiles derived from SIMBA T64 are available from PANGAEA:

https://doi.pangaea.de/10.1594/PANGAEA.940617 (added)

- Temperature profiles derived from SIMBA T65 are available from PANGAEA: https://doi.pangaea.de/10.1594/PANGAEA.940634 (added)
- Temperature profiles derived from SIMBA T66 are available from PANGAEA: https://doi.pangaea.de/10.1594/PANGAEA.938134 (added)
- Temperature profiles derived from SIMBA T67 are available from PANGAEA: https://doi.pangaea.de/10.1594/PANGAEA.938128 (added)
- Temperature profiles derived from SIMBA T68 are available from PANGAEA: https://doi.pangaea.de/10.1594/PANGAEA.940650 (added)
- Temperature profiles derived from SIMBA T69 are available from PANGAEA: https://doi.pangaea.de/10.1594/PANGAEA.938096 (added)
- Temperature profiles derived from SIMBA T70 are available from PANGAEA: https://doi.pangaea.de/10.1594/PANGAEA.940659 (added)
- Temperature profiles derived from SIMBA T72 are available from PANGAEA: https://doi.pangaea.de/10.1594/PANGAEA.940668 (added)

---

## Author Comment (AC2)

**Responses to RC2**

We sincerely appreciate the reviewer's positive and constructive comments on our manuscript. We have made careful and thorough modifications according to your suggestions in the revised manuscript. We hope that the revisions meet the reviewer's expectations. The original RC2 comments are shown in *red italics*, followed by our responses in standard black. Key points are highlighted in **bold black**; **Fig.** and **Table** represent the additional figure and table information, respectively.

*This manuscript, titled "**Estimating seasonal bulk density of level sea ice using the data derived from in situ and ICESat-2 synergistic observations during MOSAiC**", deals with the sea ice density, one of the key parameter of the sea ice. Sea ice density is indicative of the thermodynamic history of the sea ice, and it plays an essential role in estimating the sea ice thickness during satellite altimetry. However, it is also one of the most poorly observed parameter. So I consider the topic, as well as the major results, are highly relevant to this journal's scope. A large body of a variety of dataset are used for deriving the IBD, including in-situ, airborne and satellite data. Relevant issues and extensions to the work, such as the impact on satellite altimetry are also covered. I consider the majority of the work is reasonable in its current form. However, I have the following comments for major revisions before it be considered for acceptance.*

**Overall Reply** We would like to express our gratitude for your positive feedback on our manuscript and have made the following major revisions: 1) we clarified the issues of representativeness and independency involved in the regional IBD parameterizations; 2) we analyzed the potential causes of IBD seasonality; 3) We added details describing how to apply the regional IBD parameterizations to satellite retrievals of sea ice thickness, using the AWI CS2 product as an example; 4) We highlighted the improvements in the IBD parameterization itself and the "compensation effect" in the retrieval of sea ice thickness; 5) We discussed the impact of ice deformation on the application of our regional IBD parameterizations.

*First, one key issue is the independency of the estimated IBD and the contributing factors. It is important to differentiate between direct measurements of IBD and the derived IBD. Ideally, only the direct measurements (in this case, coring) should be used for the parameterization of IBD, which however, is not available at large scale. If the derived IBD is used/evaluated, one should be very careful to make sure that it is independent from other parameter used. For example in Eq. 2, IBD is estimated, but not directly observed. One notable example is that the good correlation in Fig. 8.f totally disappears in Fig. 9.f. I wonder whether this contrast can be really explained by the scale of the observation, but not by something else. Is the significant correlation in Fig. 8.f largely due to the way IBD is derived in Eq. 2? Since the ice freeboard is computed as (hf-hs) and the ratio to the ice thickness is directly written in the equation of: ((hf-hs)/hi), then is the correlation physical, or did it just arise because how you **ESTIMATE** the ice density? One should be very careful here, since uncertainty in the input values could be wrongly interpreted as correlation, since the uncertainty is transferred to the derived value. Then the correlation could be purely artificial. At least some more explanation is needed here. Especially please pay attention to representation issues for DN and L-site studies, which should be at least discussed more thoroughly in Sec. 4.*

**GC1 Reply** We agree with the reviewer's concerns about the representativeness and independency issues associated with the regional IBD parameterizations. We provided additional results to address and clarify these issues. **Overall, we shown that our regional IBD parameterizations remain valid despite the influence of error propagation, see details below.**

**GC1: Representativeness** We clarified the representativeness of the developed IBD parameterizations. Throughout the freezing season, the IBD parameterization derived from the regional buoy sites included a total of 78 samples, with 47 at the DN scale (FYI/SYI = 4/11) and 31 at the L-site scale (FYI/SYI = 4/7), as shown in **Fig. B1**. In contrast, the local scale comprised 23 samples, including 10 from FYI cores and 13 from SYI cores. The samples used for parameterization themselves exhibited time-varying properties, such as ice porosity, temperature, salinity, thickness, and age. Meanwhile, we focus on comparing *potential* IBD parameterizations derived from regional (**Fig. B2**; hydrostatic equilibrium method) and local estimates (**Fig. B3**; weighing), recognizing that absolute robustness cannot be claimed due to inherent differences in their spatial scales, measurement methods, sample sizes, and ice types.

[Figure]

**Figure B1.** Flowchart of the IBD retrieval process.

The parameterized equations for IBD at the regional scale based on sea ice parameters are defined as follows:

$$\overline{\rho_i(x, y, t)} = a1 \times \overline{X(x, y, t)} + a2, \tag{B1}$$

where the $\overline{\rho_i(x, y, t)}$ and $\overline{X(x, y, t)}$ represent the regional IBD and other sea ice parameters at any given grid cell over tens of kilometers $(x, y)$ and time $(t)$, respectively. **Table B1** lists the regression coefficients and their 95 % confidence intervals for Eq. (B1).

The parameterized equations for IBD at the local scale based on sea ice parameters are defined as follows:

$$\rho_i(x, y, t) = a1 \times \left(\frac{1}{X(x,y,t)}\right)^{a2} + a3, \tag{B2}$$

where the $\rho_i(x, y, t)$ and $X(x, y, t)$ represent the local IBD and other sea ice parameters at any given site $(x, y)$ and time $(t)$, respectively. **Table B2** lists the regression coefficients and their 95 % confidence intervals for Eq. (B2).

**Table B1.** Regression coefficients of the parameterized equations based on sea ice parameters at the regional scale $(\text{kg m}^{-3})$, and parentheses represent 95 % confidence intervals for the regression coefficients.

| $\overline{X(x, y, t)}$ | Variable type | a1 | a2 | Input range |
|---|---|---|---|---|
| Sea ice freeboard (m) | *Univariate* | −127 (−204, −51) | 923 (915, 932) | 0.05−0.15 m |
| Ice freeboard-to-total freeboard ratio | *Bivariate* | −115 (−158, −71) | 950 (935, 966) | 0.25−0.45 |
| Ice freeboard-to-thickness ratio | *Bivariate* | −953 (−1036, −870) | 980 (974, 986) | 0.05−0.09 |

**Table B2.** Regression coefficients of the parameterized equations based on sea ice parameters at the local scale $(\text{kg m}^{-3})$, and parentheses represent 95 % confidence intervals for the regression coefficients.

| $X(x, y, t)$ | Variable type | a1 | a2 | a3 | Input range |
|---|---|---|---|---|---|
| Sea ice thickness (m) | *Univariate* | −0.6 (−1.7, 0.6) | 4 (2, 7) | 912 (910, 914) | 0.4−2.2 m |
| Sea ice draft (m) | *Univariate* | −0.4 (−1.2, 0.4) | 4 (2, 7) | 912 (910, 913) | 0.4−2.0 m |
| Total thickness (m) | *Bivariate* | −0.7 (−2.1, 0.7) | 5 (2, 8) | 912 (910, 914) | 0.5−2.3 m |

[Figure]

**Figure B2.** Potential parameterizations for IBD at the regional scale and comparison with results at the local scale, including regression models using **(a, b)** sea ice freeboard, **(c, d)** ice freeboard-to-total freeboard ratio, and **(e, f)** ice freeboard-to-thickness ratio. Each panel shows model fit metrics, including the coefficient of determination ($R^2$) and root mean square error (*RMSE*, unit: kg m$^{-3}$). The significance level for the statistical test is set at 95%. The total number of samples *(Num)* is 78 for the regional scale and 23 for the local scale.

[Figure]

**Figure B3.** Potential parameterizations for IBD at the local scale and comparison with results at the regional scale, including regression models using **(a, b)** sea ice thickness, **(c, d)** sea ice draft, and **(e, f)** total thickness. Each panel shows model fit metrics, including the coefficient of determination ($R^2$) and root mean square error (*RMSE*, unit: kg m$^{-3}$). The significance level for the statistical test is set at 95%. The total number of samples (*Num*) is 23 for the local scale and 78 for the regional scale.

**GC1: independency issues** We presented several pieces of evidence that the error propagation from different input parameters in Eq. (2) **has not obscured the potential physical phenomena** revealed by the regional IBD parameterizations. Overall, we suggest that these regional IBD parameterizations, which are representative of the hydrostatic equilibrium-based IBD, and the regression coefficients derived from observations, still provide valuable guidance for satellite retrievals of sea ice thickness. **Four main arguments are presented below:**

1) The input parameters in Eq. (2) were independent of each other and exhibited relatively high measurement precision (Section 2.1). Meanwhile, the distinct seasonality observed in these parameters, together with the good agreement in both magnitude and seasonality between the derived IBDs and direct measurements (**Fig. B4**), effectively supports the validity of the developed parameterizations based on these parameters.

[Figure]

**Figure B4. (a)** Seasonal variation of IBD estimates at the DN (blue bars), L-site (red bars), and MCS (green symbols) scales during the MOSAiC freezing season. The black dots are the regional mean of the DN and L-site estimates. The orange dashed line represents the linear fitting from late October to early December ($R = 0.62$, $P < 0.05$) and the purple dashed line indicates the mean ± one standard deviation from late December to April. **(b)** Linear interpolation (black) and smoothing (red) of regional IBD estimates. **(c)** Box plots of all IBD estimates over the study period, showing interquartile range (IQR, Q3−Q1, boxes), median (black lines), and outliers (greater than 1.5 × IQR, red crosses). **(d)** Probability density distribution (PDF) of all IBD estimates over the study period, with labelled values showing the mean ± one standard deviation.

2) We used high-precision, spatially coincident airborne measurements from AWI IceBird to further investigate the relationship between ice freeboard-to-thickness ratio and IBD at a spatial resolution of 25 km × 25 km. We identified **a similar negative linear function** (Fig. B5) in the same range as Eq. (B1), which is expected to be less affected by error propagation, thus strengthening our conclusions.

[Figure]

**Figure B5.** Relationship between ice freeboard-to-thickness ratio and IBD at a spatial resolution of 25 km × 25 km based on AWI IceBird measurements. The model fit metrics include the coefficient of determination ($R^2$) and root mean square error (*RMSE*, unit: kg m$^{-3}$). **Note:** The significance level for the statistical test is set at 95% and the total number of samples is 448.

3) By incorporating the physical factor of sea ice bulk temperature into the regional IBD parameterizations based on sea ice parameters, we found significant improvements in fitting performance compared to the original models (Fig. B6). This suggests that the initial good fit between different sea ice parameters and IBD is **unlikely to be dominated by computational redundancy or spurious correlations**; otherwise the inclusion of independent physical information would tend to weaken model performance.

[Figure]

**Figure B6.** Potential parameterizations for IBD at the regional scale using both sea ice parameters and sea ice bulk temperature, including **(a)** sea ice freeboard and sea ice bulk temperature versus IBD, **(b)** ice freeboard-to-total freeboard ratio and sea ice bulk temperature versus IBD, **(c)** ice freeboard-to-thickness ratio and sea ice bulk temperature versus IBD. Each panel shows model fit metrics, including the coefficient of determination ($R^2$) and root mean square error *(RMSE,* unit: kg m$^{-3}$). **Note:** The significance level for the statistical test is set at 95% and the total number of samples is 78.

4) Independent validation with coring samples from the 2022 GoNorth expedition showed that the relative error of our parameterizations was between 0.2% and 0.9% within their applicability range (**Fig. B7**), and that **the regional IBD parameterizations even outperformed the local parameterizations developed from the MOSAiC ice cores**. Furthermore, the A10 configuration (882 kg m⁻³) significantly underestimated the observed SYI bulk density, while our newly proposed age-related IBD climatology (907 kg m⁻³) was significantly improved (see details in GC3 Reply).

[Figure]

**Figure B7.** Evaluation of the regional and local IBD parameterizations during the GoNorth Expedition in late October 2022. The age-related IBD climatology and its uncertainty are also shown. **Note: Weighing** represents the direct measurement of IBD using the hydrostatic weighing method with an uncertainty defined as 0.2 %. **Reg.** and **Loc.** are regional and local scales, respectively. **IFB**, **Tem**, **SIT**, **SID**, and **TOT** represent ice freeboard, ice bulk temperature, ice thickness, ice draft, and total thickness, respectively. The uncertainty of the IBDs derived from the parameterizations is determined by the 95 confidence intervals of their regression coefficients.

*Besides, it really surpised me that in Fig. 9.a (also in 5.a) that the coring based FYI bulk density increases since October to December. **Is this physical or due to sampling representations?***

**GC2 Reply** **We discussed in detail the possible causes of the increasing trend in IBD from late October to early December, which is mainly influenced by the air fraction (physical).**

According to the theoretical relationship between the density of gas-free sea ice and its temperature and salinity (Timco and Frederking, 1996), the observed variations in bulk temperature (from about −4 to −8 °C) and bulk salinity (from about 6 to 5 ppt) of the FYI cores would hardly contribute to an increase in their bulk density from late October to early December (**Fig. B8**). **Thus, the significant increase in IBD during the early stages could be closely related to the significant reduction in the air fraction (Salganik et al., 2024)**, as the conversion of liquid water to solid ice in closed brine pockets causes volume expansion, compressing trapped air bubbles and pushing gas into the brine solution as temperatures fall (Crabeck et al., 2016; Crabeck et al., 2019).

**The rapid thickening of the sea ice may also have significantly regulated the brine volume and air fraction within the ice layer.** For example, the newly formed ice layer will gradually transition from a loose granular structure to a dense columnar structure, resulting in a more compact layer (Oggier and Eicken, 2022). Indeed, the air fraction has been shown to be the most important factor influencing IBD, as it is much less dense than sea ice and brine (Timco and Frederking, 1996). Furthermore, no clear evidence of main desalination processes (Petrich and Eicken, 2017) was observed during the early phase of the study, which partly explains the relatively low IBD observed in late October. The warm air intrusion events were not shown to significantly affect sea ice bulk salinity and brine volume, although they were expected to enhance sea ice permeability, which may be related to their shorter duration (Fig. B8).

[Figure]

**Figure B8.** Seasonal variation of **(a)** buoy-derived air temperature, **(b)** buoy-derived and core-based sea ice bulk temperature, **(c)** core-based sea ice bulk salinity, and **(d)** core-based brine volume during the MOSAiC freezing season. The red boxes in panel **(a)** mark warm air intrusion events, and the dashed line in panels **(c, d)** indicates the seasonal linear trend ($P < 0.01$).

**Our results also suggest a potential difference in ice porosity between local (MCS-SYI) and regional (SYI-dominated buoy sites) scales.** During the early period, buoy-derived mean ice thicknesses at the DN and L-site scales showed monthly growth rates of ~15 and 21 %, respectively, with the majority of buoy-monitored ice layers beginning to grow. In contrast, the MCS-SYI showed a relatively low thickness growth rate (~9 % per month) and low porosity with a brine volume of ~1 to 4 % (Fig. B8) and an air fraction of ~0.8 to 1 % (Salganik et al., 2024). In particular, Hornnes et al. (2024) found that the spatial heterogeneity of SYI porosity was greater than that of FYI during the GoNorth expedition in October 2022, and observed **similar air fractions in FYI (2.5 ± 0.5 %) and SYI cores (2.8 ± 0.5 %)**.

*Second, I suggest the authors add some more discussion on the satellite altimetry. During the derivation of ice thickness with buoyancy relationship, ice bulk density is one of the model parameters. However, the IBD proposed in this work is parameterized with ice freeboard, freeboard-thickness ratio, or ice thickness. **Neither of these 3 parameters are directly available for altimetry retrieval. Rather, they should be estimated with the retrieved ice thickness.** Even for ice freeboard, a conversion from radar freeboard and snow data is needed, which in itself is a large unresolved issue. So I suggest that at least the detailed treatment be introduced. For example, how radar freeboard is converted into ice freeboard. The bulk density estimations (Fig. 12.c) seem to be more reasonable than AWI's default setting. **But any other proof that the new IBD scheme is better than the default scheme? For example, any proof of better draft or ice thickness retrievals?** Surrounding altimetry I want to mention that according to the authors the IBD scheme is developed for level ice. Although CryoSat-2 currently is not used for ice topography retrieval, ridged ice certainly plays important roles in modulating the CS2 waveform and retrieved freeboard. **Is there a potential problem for applying it to the whole basin, especially given that a lot of deformed ice is present and the density estimation has potential limitations?** Some discussion is suggested to be added in Sec. 4 on this issue.*

GC3 Reply We agree with the reviewers' suggestion and included more details on the satellite altimetry.

**1) Extended Application.** We further quantified the impact of regional IBD parameterizations (including three equations) on satellite retrievals of sea ice thickness. For this purpose, we combined satellite altimetry data, ice age field and regional IBD parameterizations (Eq. B1 and Table B1) to provide preliminary insights into the pan-Arctic SYI bulk densities and their updated thicknesses during the 2010 to 2024 freezing seasons (Fig. B9).

**2) Processing Chain.** We added details describing how to apply the regional IBD parameterizations to satellite retrievals of sea ice thickness, using the AWI CryoSat-2 sea ice thickness product (AWI CS2) as an example. Specifically, we considered the processing chain of the AWI CS2 as a benchmark, replacing its original IBD settings based on A10 climatology with the three regional IBD parameterizations (Eq. B1). **Please see details below.**

**Given the length of the main text, we provided specific details in the supplementary material**

CryoSat-2 initially provided radar backscatter data from the surface target, which can be processed to derive radar freeboard ($h_{fr}$) by differentiating between ice floes and leads, as well as by retracking radar waveforms to identify the main scattering interface. Following the processing chain outlined by Hendricks and Paul (2023), the AWI CS2 algorithm includes a velocity correction term to convert radar freeboard to sea ice freeboard, taking into account the delayed radar velocity in the snow layer and assuming that the main scattering interface aligns with the snow-ice interface.

First, the CryoSat-2 sea ice freeboard can be estimated as follows:

$$h_{fi}^{cor} = h_{fr} + h_c, \tag{B3}$$

$$h_c = \left(\frac{c_v}{c_s} - 1\right) h_s, \tag{B4}$$

where $h_{fi}^{cor}$ is the radar velocity-corrected sea ice freeboard, $h_{fr}$ is the radar freeboard derived using the Threshold First Maximum Retracker Algorithm (TFMRA), $h_c$ represents the velocity correction term; $c_v$ is the velocity of light in a vacuum ($3 \times 10^8$ m s$^{-1}$) and $c_s$ is the radar propagation velocity in the snow layer. Here, $c_s$ was obtained from a snow bulk density-dependent parameterization: $c_s = c_v \times [1 + (5.1 \times 10^{-4})\rho_s]^{-1.5}$ m s$^{-1}$ (Ulaby et al., 1982); $\rho_s$ was derived from a time-dependent equation: $\rho_s = 6.50t + 274.51$ kg m$^{-3}$ ($t$ ranges from 0 to 6 representing October to April) (Mallett et al., 2020).

Assuming hydrostatic equilibrium, the sea ice thickness can then be determined:

$$h_i = \left(\frac{\rho_w}{\rho_w - \rho_i}\right)h_{fi}^{cor} + \left(\frac{\rho_s}{\rho_w - \rho_i}\right)h_s. \tag{B5}$$

where $h_i$, $h_{fi}^{cor}$, and $h_s$ represent the sea ice thickness, sea ice freeboard, and snow depth, respectively. $\rho_w$, $\rho_i$, and $\rho_s$ are the seawater density, sea ice bulk density, and snow bulk density, respectively. Here, $\rho_w$ was set to 1024 kg m$^{-3}$, $\rho_s$ derived from the time-dependent equation mentioned above, and $h_s$ was obtained from a merged snow depth product by combing W99 climatology and passive microwave measurements (MW99/AMSR2). $\rho_i$ was configured according to the regional IBD parameterizations and the sea ice type-weighted A10 climatology (AWI original settings), respectively.

**Taking the above equations together, the unknown variables are only $\rho_i$ and $h_i$ when applying the regional IBD parameterizations.** For the parameterizations that depend on the sea ice freeboard or the ratio of ice freeboard to total freeboard, the IBD can be calculated directly using the individual parameters of $h_{fi}^{cor}$ and $h_s$ from AWI CS2. In contrast, the parameterization depending on the ratio of ice freeboard to thickness requires a combination with Eqs. (B3–5) to estimate IBD and sea ice thickness simultaneously. The final results were calculated as the mean IBD and thickness of the pan-Arctic SYI derived using all three regional IBD parameterizations (referred to as updated; Fig. B9). **Our processing details are also applicable to other CryoSat-2/ICESat-2 sea ice thickness products.**

[Figure]

**Figure B9.** IBD and thickness variation of the pan-Arctic SYI during the 2010 to 2024 freezing seasons based on the AWI CryoSat-2 (AWI CS2) sea ice product, including results using our parameterizations (updated) and the original settings (sea ice type-weighted A10 climatology). The shaded areas denote mean ± one standard deviation and dashed lines indicate linear trends with their statistical test *P*-values. **(a)** Multi-year monthly average estimates for IBD from October to April. **(b)** Multi-year monthly average estimates for ice thickness from October to April. **(c)** Freezing season average estimates for IBD from 2010/2011 (10/11) to 2023/2024 (23/24). **(d)** Freezing season average estimates for ice thickness from 2010/2011 (10/11) to 2023/2024 (23/24).

**3) Potential Improvement.** We agree with the reviewer's comment that the IBD parameterization should improve the accuracy of satellite-based sea ice thickness retrievals. However, at this stage we aim to demonstrate the validity and improvement of the regional IBD parameterization itself compared to the A10 climatology used in most sea ice thickness products, as detailed in **GC1 Reply**.

In the context of the CryoSat-2 sea ice thickness retrieval, Landy et al. (2020) demonstrated the uncertainty and contributions of various parameters in the processing chain, with IBD being only one of the key parameters. **This suggests that improvements in sea ice thickness retrieval by applying the regional IBD parameterization may be significantly influenced by other factors,** such as surface roughness, radar penetration, snow depth, and snow bulk density, leading to a bias compensation effect. In particular, both the W99-based snow depth and the assumption of full radar penetration in the snow layer used in the AWI CS2 processing chain may have contributed to an overestimation of sea ice thickness. **Therefore, we focus on reporting the impact of the regional IBD parameterization on the retrieval of sea ice thickness, highlighting the "directional changes" in its relative magnitude, seasonality and interannual variations. More importantly, the retrieved SYI bulk densities were in good agreement with the expected range and seasonal evolution.**

**In the revision, we highlight that** future incorporation of pan-Arctic SYI bulk temperatures, combined with more accurate sea ice ancillary parameters, will be more helpful in retrieving SYI bulk densities. Overall, the incorporation of regional IBD parameterizations into the current AWI CS2 product introduced more realistic variations in IBD, resulting in **systematically higher SYI thickness, lower seasonal growth rate, and reduced interannual trend.**

**4) Caveat.** We carefully discussed the impact of ice deformation on the application of our regional IBD parameterizations, which were developed solely from level ice observations.

**Added:** "*In terms of interpretation, the IBD at DN and L-site scales primarily reflects level ice mixed with SYI (11 buoy sites) and FYI (4 buoy sites), lacking information on other ice ages and types, particularly deformed ice, which contributes to the potential limitations and uncertainties of the parameterization scheme. Notably, deformed and unconsolidated sea ice generally exhibits a higher bulk density due to the sea water trapped within it, and applying parameterizations derived from level sea ice to satellite retrievals of sea ice thickness may consequently result in* **an underestimation for deformed grid cells.**"

**We also summarized the limitations and caveats of this study at the end of the revised manuscript:** "*1) the hydrostatic equilibrium calculations of the IBD have relatively high uncertainties and sub-weekly variability compared to direct measurements; 2) the different MOSAiC observations used do not overlap spatially; 3) the regional IBD parameterizations are not strictly independent of the input parameters; 4) the IBD results represent only level sea ice and lack information on deformed ice. We can also foresee that the establishment of a regional coring network spanning different seasons and ice types will greatly advance the parameterization of IBD.*"

**5) New Climatology.** Notably, both the MOSAiC and other historical records of IBD all reported higher values than the A10-MYI (up to ~30 kg m$^{-3}$), suggesting that the reference density used for the A10-MYI was inappropriate and may also be related to the increasing youth of Arctic sea ice. **Therefore, we proposed a new age-related IBD climatology** during the freezing season by combining the mean IBD values from the Arctic Sever expedition(Alexandrov et al., 2010; Shi et al., 2023), the IceBird campaigns (Jutila et al., 2022), and the MOSAiC expedition (this study), as shown in Table B3.

**Table B3** Age-related bulk densities of level sea ice during the freezing season derived from a combination of the Arctic Sever expedition (from 1980 to 1989), the IceBird campaigns (April 2017 and 2019), and the MOSAiC expedition (from 2019 to 2020). Uncertainty is given as the maximum difference in age-related mean IBD values across observations.

| Ice age | Mean (kg m$^{-3}$) | Standard deviation (kg m$^{-3}$) | Uncertainty (kg m$^{-3}$) |
|---|---|---|---|
| FYI (0–1 years) | 916.9 | 7.5 | 20 |
| SYI (1–2 years) | 907 | 6.8 | 13 |
| MYI (2+ years) | 900 | 14 | 34 |

**Reference**

Alexandrov, V., Sandven, S., Wahlin, J., and Johannessen, O.: The relation between sea ice thickness and freeboard in the Arctic, The Cryosphere, 4, 373-380, 2010.

Crabeck, O., Galley, R., Delille, B., Else, B., Geilfus, N. X., Lemes, M., Des Roches, M., Francus, P., Tison, J. L., and Rysgaard, S.: Imaging air volume fraction in sea ice using non-destructive X-ray tomography, The Cryosphere, 10, 1125-1145, 2016.

Crabeck, O., Galley, R. J., Mercury, L., Delille, B., Tison, J.-L., and Rysgaard, S.: Evidence of Freezing Pressure in Sea Ice Discrete Brine Inclusions and Its Impact on Aqueous-Gaseous Equilibrium, Journal of Geophysical Research: Oceans, 124, 1660-1678, 2019.

Hendricks, S. and Paul, S.: Product User Guide & Algorithm Specification - AWI CryoSat-2 Sea Ice Thickness (version 2.6) Issued by (v2.6), Zenodo, https://doi.org/10.5281/zenodo.10044554, 2023.

Hornnes, V., Salganik, E., and Høyland, K. V.: Relationship of physical and mechanical properties of sea ice during the freeze-up season in Nansen Basin, Cold Regions Science and Technology, 2024. 104353, 2024.

Jutila, A., Hendricks, S., Ricker, R., von Albedyll, L., Krumpen, T., and Haas, C.: Retrieval and parameterisation of sea-ice bulk density from airborne multi-sensor measurements, The Cryosphere, 16, 259-275, 2022.

Landy, J. C., Petty, A. A., Tsamados, M., and Stroeve, J. C.: Sea Ice Roughness Overlooked as a Key Source of Uncertainty in CryoSat-2 Ice Freeboard Retrievals, Journal of Geophysical Research-Oceans, 125, 2020.

Mallett, R. D. C., Lawrence, I. R., Stroeve, J. C., Landy, J. C., and Tsamados, M.: Brief communication: Conventional assumptions involving the speed of radar waves in snow introduce systematic underestimates to sea ice thickness and seasonal growth rate estimates, Cryosphere, 14, 251-260, 2020.

Oggier, M. and Eicken, H.: Seasonal evolution of granular and columnar sea ice pore microstructure and pore network connectivity, Journal of Glaciology, 68, 833-848, 2022.

Petrich, C. and Eicken, H.: Overview of sea ice growth and properties, Sea ice, 2017. 1-41, 2017.

Salganik, E., Crabeck, O., Fuchs, N., Hutter, N., Anhaus, P., and Landy, J. C.: Impacts of air fraction increase on Arctic sea-ice thickness retrieval during melt season, EGUsphere, 2024, 1-30, 2024.

Shi, H., Lee, S.-M., Sohn, B.-J., Gasiewski, A. J., Meier, W. N., Dybkjær, G., and Kim, S.-W.: Estimation of snow depth, sea ice thickness and bulk density, and ice freeboard in the Arctic winter by combining CryoSat-2, AVHRR, and AMSR measurements, IEEE Transactions on Geoscience and Remote Sensing, 2023. 2023.

Timco, G. and Frederking, R.: A review of sea ice density, Cold regions science and technology, 24, 1-6, 1996.

Ulaby, F., Moore, R., and Fung, A.: Microwave remote sensing: Active and passive. Volume 2-Radar remote sensing and surface scattering and emission theory. 1982.

---

## Author Comment (AC4)

**Summary of the major revision**

**Dear Editor and Reviewers,**

We would like to express our sincere appreciation to the editor and reviewers for their comments to improve our manuscript. Regarding the concerns raised about the significant changes in the IBD results after each revision, we would like to clarify that the IBD differences compared to the previous version (egusphere-2024-1240) are primarily attributable to variations in the input parameters rather than the hydrostatic equilibrium method used. In this version, we have included **physical explanations, modeled estimates, independent validation**, and **details of the spatial scale adjustment** to illustrate the feasibility of estimating the seasonal bulk density of level sea ice over tens of kilometers using the hydrostatic equilibrium method.

In response to Referee #1's suggestions, we have made the following improvements: 1) We have clarified the regional representativeness and validity of the retrieved IBDs; 2) We have provided specific descriptions of the spatial scale adjustment; 3) We have incorporated physical interpretations of the IBD results and developed additional parameterizations that integrate physical factors; 4) We have thoroughly discussed the uncertainties and constraints associated with the IBD process; 5) We have evaluated the developed parameterizations against independent coring measurements, with relative biases ranging from ~0 to 0.9% for SYI and from 0.1% to 2.1% for FYI; 6) We have estimated the pan-Arctic second-year ice bulk densities from 2010 to 2024 to demonstrate the applicability of our regional IBD parameterizations; 7) We have also proposed a new age-related IBD climatology, taking into account the limitations of the parameterizations.

In response to the suggestions of Referee #2, we have implemented the following major revisions: 1) We have clarified the issues of representativeness and independence associated with the regional IBD parameterizations; 2) We have analyzed the potential causes of IBD seasonality and developed additional parameterizations incorporating physical factors; 3) We have provided details on the application of the regional IBD parameterizations to satellite retrievals of sea ice thickness, using the AWI CS2 product as an illustrative example; 4) We have highlighted the improvements in the IBD parameterizations themselves (independent validation), as well as the compensation effect in satellite retrievals of sea ice thickness; 5) We have added a detailed description of the applicability and limitations of the IBD parameterizations, particularly the lack of information on deformed sea ice.

Taken together, our findings provide preliminary evidence of seasonal variability in IBD during the MOSAiC freezing season, derived from both point observations and regional integration. Some concerns and limitations of this study are also summarized in the *Conclusion*. The proposed IBD parameterizations, along with the newly introduced ice age-related IBD climatology, can serve as valuable references for future satellite retrieval efforts (e.g., ESA - CRISTAL). We hope that you are satisfying with the revision.

Best regards in the New Year, Yi Zhou on behalf of the authors.

**Responses to the Referee 1#**

We sincerely appreciate your constructive comments, which have significantly contributed to the improvement of our manuscript. We have made detailed revisions or clarifications according to your suggestions. The original reviewer comments are presented in *blue italics*, followed by our responses in black. Fig. A and Table A denote the additional figure and table information, respectively. Furthermore, specific locations of the modifications in the revised manuscript are highlighted in purple.

The study titled "Estimating seasonal bulk density of level sea ice using the data derived from in situ and ICESat-2 synergistic observations during MOSAiC" integrates observations of snow and ice thickness from 15 ice mass balance buoys, along with snow freeboard data from ICESat-2, airborne laser altimeters, and snow density data to estimate bulk ice density for a drifting ice floe and its surroundings at various scales. It also compares these estimates with manual observations of ice density and proposes several parameterizations, primarily using ice freeboard and thickness as key parameters.

Parametrization of ice density is a valuable task, and the study provides new data and implements known concepts to derive density estimates from large scale observations. The methods are clearly described, yet some details do not allow a reproducibility. The overview of remote methods is accurate enough, while an overview of in situ measurements is quite limited. In addition, there are a lot of inconsistencies related to the description of representative ice types and ice and snow thicknesses, related to a limited usage of existing in situ observations during the described expedition.

While the study offers valuable insights into the upscaling of sea ice freeboard data during the ice drift, where measurements were limited to individual floes, there are some methodological challenges that should be considered. For instance, the ice density estimates rely on a methodology with certain limitations. Some of these issues were addressed by manually adjusting the freeboard measurements, although the rationale and approach behind this adjustment could be more clearly explained. While these adjustments resulted in more realistic density estimates, there remains significant variability in the data, similar to the previously observed seasonal fluctuations in ice density.

Given this level of variability and the presence of an adjustment parameter that heavily influences both the density values and their seasonality, it is challenging to recommend a robust parameterization based on the current data. Additionally, the assumption that the adjustment is time-independent may need further justification. As a result, it becomes difficult to fully assess the quality of the density estimates due to both the high variability and the manual adjustments made during the analysis.

**Overall Reply** We gratefully acknowledge the reviewers' feedback on the manuscript and have made significant improvements or clarifications in the **representativeness, compatibility, variability, validity, limitations, and uncertainties** of the IBD results.

**Representativeness**

In fact, the estimated IBDs represent the average condition of the ice cover as measured by the selected IMBs, as shown in **Fig. A1**. As a result, all regional scale statements are based on the results from the buoy sites (SYI-dominated), which also motivates the so-called spatial scale adjustment. Please refer to "2.2 Methods" for specific revisions.

Figure A1. Flowchart of the IBD retrieval process.

**Compatibility**

The spatial adjustment term was derived from specific calculations, rather than being manually determined; additional details have been provided in the revised manuscript. The purpose of this adjustment is to align the large-scale observations with the ice layers deployed by the buoys. We explain how the correction term was derived and why it is time-independent in our response to the general comments (**GC2**). Essentially, the adjusted modal total freeboard can be regarded as an approximation of the mean total freeboard across the buoy sites. Please refer to "2.2.2 Spatial scale adjustment" and "Text S2. Feasibility of the spatial scale adjustment" for specific revisions.

**Variability**

We also attempted to derive regional IBDs by combining sea ice thickness and snow depth from the local transects, ALS/IS2-derived total freeboard (unadjusted), along with the seawater and snow densities used in this study. However, this approach resulted in significantly greater variability and higher IBD values compared to those derived from the buoy array and ice cores, highlighting a more pronounced spatial and temporal inconsistency problem. Truthfully, the current IBD results represent the most appropriate solution after evaluating several combinations of input parameters in the IBD retrieval during MOSAiC at the regional scale.

Compared to previous studies that used the hydrostatic equilibrium method to estimate IBD, the variability of our retrieved IBDs was significantly lower, with a standard deviation of approximately 10 kg m-3. Meanwhile, the compression of the time axis and the sparse data points at different spatial scales increased the visual variability of the IBD results. Given the expected high spatial heterogeneity of ice porosity across the buoy sites (11 SYI + 4 FYI), we also argue that the IBD variation, which primarily ranges between 900 and 920 kg m-3, is generally

reasonable despite the presence of some data noise. At the very least, if the hydrostatic equilibrium-based IBDs in this study are invalid, the predictive performance of the regional IBD parameterizations based on these results is unlikely to exceed that of the local IBD parameterizations derived from the MOSAiC ice cores (GC1: Independent Validation).

---

## Author Comment (AC5)

**Summary of the major revision**

**Dear Editor and Reviewers,**

We would like to express our sincere appreciation to the editor and reviewers for their comments to improve our manuscript. Regarding the concerns raised about the significant changes in the IBD results after each revision, we would like to clarify that the IBD differences compared to the previous version (egusphere-2024-1240) are primarily attributable to variations in the input parameters rather than the hydrostatic equilibrium method used. In this version, we have included **physical explanations**, **modeled estimates**, **independent validation**, and **details of the spatial scale adjustment** to illustrate the feasibility of estimating the seasonal bulk density of level sea ice over tens of kilometers using the hydrostatic equilibrium method.

**In response to Referee #1's suggestions, we have made the following improvements:** 1) We have clarified the regional representativeness and validity of the retrieved IBDs; 2) We have provided specific descriptions of the spatial scale adjustment; 3) We have incorporated physical interpretations of the IBD results and developed additional parameterizations that integrate physical factors; 4) We have thoroughly discussed the uncertainties and constraints associated with the IBD process; 5) We have evaluated the developed parameterizations against independent coring measurements, with relative biases ranging from ~0 to 0.9% for SYI and from 0.1% to 2.1% for FYI; 6) We have estimated the pan-Arctic second-year ice bulk densities from 2010 to 2024 to demonstrate the applicability of our regional IBD parameterizations; 7) We have also proposed a new age-related IBD climatology, taking into account the limitations of the parameterizations.

**In response to the suggestions of Referee #2, we have implemented the following major revisions:** 1) We have clarified the issues of representativeness and independence associated with the regional IBD parameterizations; 2) We have analyzed the potential causes of IBD seasonality and developed additional parameterizations incorporating physical factors; 3) We have provided details on the application of the regional IBD parameterizations to satellite retrievals of sea ice thickness, using the AWI CS2 product as an illustrative example; 4) We have highlighted the improvements in the IBD parameterizations themselves (independent validation), as well as the compensation effect in satellite retrievals of sea ice thickness; 5) We have added a detailed description of the applicability and limitations of the IBD parameterizations, particularly the lack of information on deformed sea ice.

Taken together, our findings provide preliminary evidence of seasonal variability in IBD during the MOSAiC freezing season, derived from both point observations and regional integration. Some concerns and limitations of this study are also summarized in the *Conclusion*. The proposed IBD parameterizations, along with the newly introduced ice age-related IBD climatology, can serve as valuable references for future satellite retrieval efforts (e.g., ESA - CRISTAL). We hope that you are satisfying with the revision.

**Best regards in the New Year,**
**Yi Zhou on behalf of the authors.**

**Responses to the Referee 1**

We sincerely appreciate your constructive comments, which have significantly contributed to the improvement of our manuscript. We have made detailed revisions or clarifications according to your suggestions. The original reviewer comments are presented in *blue italics*, followed by our responses in black. Fig. A and Table A denote the additional figure and table information, respectively. Furthermore, specific locations of the modifications in the revised manuscript are highlighted in purple.

*The study titled "Estimating seasonal bulk density of level sea ice using the data derived from in situ and ICESat-2 synergistic observations during MOSAiC" integrates observations of snow and ice thickness from 15 ice mass balance buoys, along with snow freeboard data from ICESat-2, airborne laser altimeters, and snow density data to estimate bulk ice density for a drifting ice floe and its surroundings at various scales. It also compares these estimates with manual observations of ice density and proposes several parameterizations, primarily using ice freeboard and thickness as key parameters.*

*Parametrization of ice density is a valuable task, and the study provides new data and implements known concepts to derive density estimates from large scale observations. The methods are clearly described, yet some details do not allow a reproducibility. The overview of remote methods is accurate enough, while an overview of in situ measurements is quite limited. In addition, there are a lot of inconsistencies related to the description of representative ice types and ice and snow thicknesses, related to a limited usage of existing in situ observations during the described expedition.*

*While the study offers valuable insights into the upscaling of sea ice freeboard data during the ice drift, where measurements were limited to individual floes, there are some methodological challenges that should be considered. For instance, the ice density estimates rely on a methodology with certain limitations. Some of these issues were addressed by manually adjusting the freeboard measurements, although the rationale and approach behind this adjustment could be more clearly explained. While these adjustments resulted in more realistic density estimates, there remains significant variability in the data, similar to the previously observed seasonal fluctuations in ice density.*

*Given this level of variability and the presence of an adjustment parameter that heavily influences both the density values and their seasonality, it is challenging to recommend a robust parameterization based on the current data. Additionally, the assumption that the adjustment is time-independent may need further justification. As a result, it becomes difficult to fully assess the quality of the density estimates due to both the high variability and the manual adjustments made during the analysis.*

**Overall Reply** We gratefully acknowledge the reviewers' feedback on the manuscript and have made significant improvements or clarifications in the **representativeness, compatibility, variability, validity, limitations, and uncertainties** of the IBD results.

**Representativeness**

In fact, the estimated IBDs represent the average condition of the ice cover as measured by the selected IMBs, as shown in Fig. A1. As a result, all regional scale statements are based on the results from the buoy sites (SYI-dominated), which also motivates the so-called spatial scale adjustment. Please refer to "2.2 Methods" for specific revisions.

[Figure]

**Figure A1.** Flowchart of the IBD retrieval process.

**Compatibility**

The spatial adjustment term was derived from specific calculations, rather than being manually determined; additional details have been provided in the revised manuscript. The purpose of this adjustment is to align the large-scale observations with the ice layers deployed by the buoys. We explain how the correction term was derived and why it is time-independent in our response to the general comments (**GC2**). Essentially, the adjusted modal total freeboard can be regarded as an approximation of the mean total freeboard across the buoy sites. Please refer to "2.2.2 Spatial scale adjustment" and "Text S2. Feasibility of the spatial scale adjustment" for specific revisions.

**Variability**

We also attempted to derive regional IBDs by combining sea ice thickness and snow depth from the local transects, ALS/IS2-derived total freeboard (unadjusted), along with the seawater and snow densities used in this study. However, this approach resulted in significantly greater variability and higher IBD values compared to those derived from the buoy array and ice cores, highlighting a more pronounced spatial and temporal inconsistency problem. Truthfully, the current IBD results represent the most appropriate solution after evaluating several combinations of input parameters in the IBD retrieval during MOSAiC at the regional scale.

Compared to previous studies that used the hydrostatic equilibrium method to estimate IBD, the variability of our retrieved IBDs was significantly lower, with a standard deviation of approximately 10 kg m$^{-3}$. Meanwhile, the compression of the time axis and the sparse data points at different spatial scales increased the visual variability of the IBD results. Given the expected high spatial heterogeneity of ice porosity across the buoy sites (11 SYI + 4 FYI), we also argue that the IBD variation, which primarily ranges between 900 and 920 kg m$^{-3}$, is generally

reasonable despite the presence of some data noise. At the very least, if the hydrostatic equilibrium-based IBDs in this study are invalid, the predictive performance of the regional IBD parameterizations based on these results is unlikely to exceed that of the local IBD parameterizations derived from the MOSAiC ice cores (**GC1: Independent Validation**).

[Figure]

Please refer to "3.2 Seasonal variability of IBD during MOSAiC", "4.1 Uncertainties and limitations of IBD retrieval", and "4.3 Evaluation of IBD parameterizations" for specific revisions.

**Validity**

We provided several lines of evidence supporting the validity of the hydrostatic equilibrium-based IBDs derived from the selected buoy sites. Although notable differences in IBD variability were observed at daily and weekly intervals, the seasonal variations of the regional IBD estimates were generally in good agreement with direct coring measurements (**Fig. A2** and **Table A1**), thereby demonstrating the feasibility of using the hydrostatic equilibrium method to estimate the seasonal bulk density of level sea ice over tens of kilometers. In this regard, the monthly and seasonal averages of regional IBDs estimates, together with the segment features for different periods, effectively reduced some of the IBD noise and revealed a clear seasonal pattern. Please refer to "3.2 Seasonal variability of IBD during MOSAiC" for specific revisions.

Given the relatively high uncertainties and sparse sampling associated with the early increasing trend of regional IBD estimates, we also introduced hydrostatic equilibrium-based modelled IBDs and observed a similar increasing trend from late October to early December (**GC3: Regional Modeled IBD**). Furthermore, an independent validation of the regional IBD parameterizations against coring measurements well supported the validity of our IBD calculations (**GC1: Independent Validation**). Please refer to "Text S4. Hydrostatic equilibrium-based modeled IBD estimates during MOSAiC" and "4.3 Evaluation of IBD parameterizations" for specific revisions.

Meanwhile, we have also added analyses of the main drivers of IBD seasonality and the possible reason why the variation in regional IBD estimates is more consistent with the MCS-FYI, despite being predominantly composed of SYI (**GC3: Physical Explanation**). Please refer to "4.2 Possible factors contributing to the IBD seasonality" for specific revisions.

[Figure]

**Figure A2. (a)** Seasonal variation of IBD estimates at the DN scale (blue bars), L-site scale (red bars), and MCS (green symbols) during the MOSAiC freezing season. The black dots represent the mean values of the regional IBD estimates at the DN and L-site scales. The orange dashed line shows the linear fit for regional IBD estimates from late October to early December ($R = 0.62$, $P < 0.05$), while the purple dashed line indicates the mean ± one standard deviation from late December to April. **(b)** Comparison of regional modeled values, monthly mean of regional buoy estimates, and linearly interpolated MCS results. The error bars for regional buoy estimates represent one standard deviation. **(c)** Box plots of all IBD estimates over the study period, showing the interquartile range (IQR, Q3−Q1, boxes), median (black lines) and outliers (values greater than 1.5 × IQR, red crosses). **(d)** Probability density distribution (PDF) of all IBD estimates over the study period, with labelled values indicating the mean ± one standard deviation.

**Table A1.** Seasonal averages of IBD at regional and local scales during the MOSAiC freezing season, where ON denotes autumn (October to November), DJF denotes winter (December to February), and MA denotes spring (March to April). Statistical information is presented as mean value (Mean), standard deviation (Std) and total number of samples (*Num*).

| Season | DN & L-site (buoy sites, 4 FYI + 11 SYIl) | | | MCS-FYI (sampling point) | | | MCS-SYI (sampling point) | | |
|---|---|---|---|---|---|---|---|---|---|
| | Mean (kg m⁻³) | Std (kg m⁻³) | *Num* | Mean (kg m⁻³) | Std (kg m⁻³) | *Num* | Mean (kg m⁻³) | Std (kg m⁻³) | *Num* |
| ON | 900 | 14 | 5 | 899 | 8 | 5 | 912 | 1 | 4 |
| DJF | 910 | 9 | 45 | 909 | 4 | 6 | 911 | 3 | 4 |
| MA | 910 | 7 | 28 | 913 | 3 | 2 | 912 | 2 | 4 |

**Limitations and Uncertainties**

Due to irregular variations in airborne and satellite trajectories, as well as the limitations of the buoys in capturing variations in mean snow depth over their deployment area, we have emphasized that the retrieval method may not fully resolve synoptic-scale IBD variations with a relatively high uncertainty of ~3 %, which is the inherent limitation of the retrieval method (Fig. A3). Please refer to "4.1 Uncertainties and limitations of IBD retrieval" for specific revisions.

[Figure]

**Figure A3. (a)** Seasonal variation of IBD uncertainties at the DN (blue dots) and L-site (red dots) scales during the MOSAiC freezing season. Also shown are the relative contributions of uncertainties for different input parameters to the total IBD uncertainty over the study period at the **(b)** DN and **(c)** L-site scales, respectively. The boxplots depict the inter-quartile range (IQR, Q3−Q1, boxes), median (black lines), mean (grey dots), and outliers (exceeding 1.5 × IQR, red crosses).

**GC1:** *A parameterization of sea-ice density can indeed be a valuable tool for both modelers and observers. However, I have some concerns regarding the potential usefulness of the parameterization presented here for such applications. Firstly, it is based on a methodology with significant uncertainties, which in many cases seem to exceed the actual seasonal variability of sea-ice density. Secondly, the current parameterization focuses on correlations between density and ice freeboard or thickness, without addressing important physical factors such as region, ice salinity, ice temperature, or water temperature. Since ice freeboard is inherently linked to its density, this creates a positive feedback loop. While snow may also play a role, it is less likely to directly influence ice density. Low ice freeboard can result from both high ice density and reduced thickness, complicating the interpretation. Similarly, the relationship between ice thickness and density involves multiple factors, including the region, ice age, and type, in addition to seasonal variations. Suggesting that only very thin ice has lower density seems to oversimplify the complexity of these processes. Additionally, it may be less useful to rely on ice thickness as an input parameter, as this is typically the primary goal of many remote sensing estimates rather than an available input variable.*

**GC1 Reply** **We agree and appreciate the reviewer's suggestion.** In the revision, we focus on comparing potential IBD parameterizations derived from buoy sites (regional) and MOSAiC coring sites (local), considering the inherent differences in spatial scale (regional vs. local), measurement methods (hydrostatic equilibrium vs. weighing), sample sizes or footprints, and ice types. Throughout the freezing season, the regional IBD parameterizations included a total of 78 samples, with 47 at the DN scale (FYI/SYI = 4/11) and 31 at the L-site scale (FYI/SYI = 4/7). In contrast, the local results consisted of 23 samples, including 10 from FYI cores and 13 from SYI cores.

**Five major revisions or clarifications are presented as follows:**
1) We would like to clarify that the primary objective of this study is to propose several potential parameterizations for predicting IBD throughout the freezing season, adapted to contemporary ice conditions in the central Arctic. **The samples used for the IBD parameterization themselves exhibited seasonal properties** including ice porosity, temperature, salinity, thickness, and age. We also recommend applying the parameterizations within their potential applicability ranges. Please refer to "3.4 Potential parameterization for IBD" for specific revisions.

2) **We agree with the reviewer's comment that predicting IBD based solely on sea ice parameters is challenging and fully acknowledge the logical circularity issue raised.** In the revised manuscript, we have proposed several IBD parameterizations using physical factors (sea ice bulk temperature, bulk salinity, brine volume fraction, and air volume fraction), as well as those incorporating both sea ice parameters and physical factors, which significantly enhance physical interpretability. Please refer to "3.4 Potential parameterization for IBD" for specific revisions.

3) **An independent validation using core samples collected during the 2022 GoNorth expedition indicated that the relative biases of the proposed parameterizations ranged from approximately 0 to 0.9 % for SYI and from 0.1 to 2.1 % for FYI, within their potential**

**respective applicability ranges.** Encouragingly, the predictive accuracy of the regional IBD parameterizations based on sea ice parameters even outperformed those derived from the MOSAiC core samples, further supporting the validity of our IBD calculations. The regional IBD parameterizations incorporating both sea ice parameters and physical factors demonstrated good accuracy for both FYI and SYI. Please refer to "4.3 Evaluation of IBD parameterizations" for specific revisions.

4) **We proposed a new age-related IBD climatology by considering the limitations of the developed IBD parameterizations.** These age-related IBD values can serve as an updated IBD climatology for future basin-scale satellite sea ice thickness retrievals, providing an alternative to A10, with two key upgrades: 1) additional SYI information, and 2) mitigation of the potentially biased A10-MYI. Please refer to "3.3 Comparison of IBD during MOSAiC with historical measurements" and "4.3 Evaluation of IBD parameterizations" for specific revisions.

5) We combined satellite altimetry data, ice age field, and three regional IBD parameterizations to provide preliminary insights into pan-Arctic SYI bulk densities and their updated thicknesses during the 2010 to 2024 freezing seasons. In this context, **the IBD parameterizations involving sea ice thickness remains valid for satellite-based retrievals of sea ice thickness**, as it can be combined with radar/laser freeboard-to-thickness conversion to estimate both IBD and sea ice thickness simultaneously, as detailed in **Text S5**. Please refer to "4.5 Implications of IBD parameterizations for the sea ice thickness retrieval" for specific revisions.

**GC1: Comparison of IBD Parameterizations**

We highlighted significant impacts of spatial scale and computational method on the IBD parameterization using sea ice parameters (Figs. A4−5). The specific forms, regression coefficients, and uncertainty estimates for these parameterized equations are provided in this revision, all of which represent empirical statistical models. Meawhiled, we provided several lines of evidence that the differences between regional and local IBD parameterizations are primarily due to **spatial scale (physical phenomena)** rather than computational methods, please see our response to general comments in **RC2** for details.

Our potential parameterized equations for IBD at the regional scale support many of the previous findings. "*For example, Ackley et al. (1974) proposed a negative linear equation dependent on ice freeboard, Kovacs (1997) developed a power equation dependent on ice thickness, Alexandrov et al. (2010) and Shi et al. (2023) defined positive linear equations dependent on the ice freeboard-to-thickness ratio, and Jutila et al. (2022) developed a negative exponential function dependent on ice freeboard.*"

[Figure]

**Figure A4.** Potential parameterizations for IBD at the regional scale and comparison with results at the local scale, including regression models using **(a, b)** sea ice freeboard, **(c, d)** ice freeboard-to-total freeboard ratio, and **(e, f)** ice freeboard-to-thickness ratio. Each panel shows model fit metrics, including the coefficient of determination ($R^2$) and root mean square error (*RMSE*, kg m$^{-3}$). The significance level for the statistical test is set at 95 %. The total number of samples *(Num)* is 78 for the regional scale and 23 for the local scale.

[Figure]

**Figure A5.** Potential parameterizations for IBD at the local scale and comparison with results at the regional scale, including regression models using **(a, b)** sea ice thickness, **(c, d)** sea ice draft, and **(e, f)** total thickness. Each panel shows model fit metrics, including the coefficient of determination ($R^2$) and root mean square error ($RMSE$, kg m$^{-3}$). The significance level for the statistical test is set at 95 %. The total number of samples ($Num$) is 23 for the local scale and 78 for the regional scale.

**GC1: Physical Factors**

We investigated whether physical factors could improve the established IBD parameterizations based on sea ice parameters. Here, a multivariate linear regression model was used to investigate the relationships between these variables and IBD. Physical factors at the regional scale included air and ice bulk temperatures derived from buoy sites, while at the sampling site sea ice bulk temperature, bulk salinity, brine volume, and air fraction derived from ice core samples were included.

The analysis showed that **sea ice bulk temperature significantly improved the performance of the developed regional IBD parameterizations** (Fig. A6), particularly the sea ice freeboard-dependent parameterization ($R^2$ improved by ~85%).

[Figure]

**Figure A6.** Potential parameterizations for IBD at the regional scale using using both sea ice parameters and the physical factor of ice bulk temperature, including **(a)** sea ice freeboard and sea ice bulk temperature, **(b)** ice freeboard-to-total freeboard ratio and sea ice bulk temperature, **(c)** ice freeboard-to-thickness ratio and sea ice bulk temperature. Each panel shows model fit metrics, including the coefficient of determination ($R^2$) and root mean square error *(RMSE*, kg m$^{-3}$). **Note:** The significance level for the statistical test is set at 95 % and the total number of samples is 78.

We also found significant correlations between some physical factors and IBD data from local coring samples (Fig. A7). With the exception of air volmue fraction, incorporating any other physical factor into the local IBD parameterizations based on sea ice parameters did not improve the original fitting performance (Table A2).

[Figure]

**Figure A7.** Potential parameterizations for IBD at the local scale using physical factors, including regression models using **(a)** sea ice bulk temperature, **(b)** air volume fraction, **(c)** sea ice bulk salinity, and **(d)** brine volume fraction. Each panel shows model fit metrics, including the coefficient of determination ($R^2$) and root mean square error *(RMSE*, kg m$^{-3}$). **Note:** The significance level for the statistical test is set at 95 % and the total number of samples (*Num*) is 23.

**Table A2.** Fitting performance of different parameter combinations with IBD based on a multivariate linear regression model. The variance inflation factor (VIF) of these equations is less than 2. **Note** that **SID**, **SIT**, and **TOT** represent ice draft, ice thickness, and total thickness, respectively. **Tem**, **AV**, **Sal**, and **BV** are ice bulk temperature, air volume fraction, ice bulk salinity, and brine volume fraction, respectively.

| Combination | $R^2$ | *RMSE* ( kg m$^{-3}$ ) |
|---|---|---|
| SID + Tem | 0.30 | 5.3 |
| SID + AF | 0.95 | 1.4 |
| SID + Sal | 0.49 | 4.6 |
| SID + BV | 0.56 | 4.2 |
| TOT + Tem | 0.35 | 5.3 |
| TOT + AF | 0.95 | 1.4 |
| TOT + Sal | 0.52 | 4.5 |
| TOT + BV | 0.58 | 4.2 |
| SIT + Tem | 0.30 | 5.3 |
| SIT + AF | 0.95 | 1.4 |
| SIT + Sal | 0.56 | 4.2 |
| SIT + BV | 0.56 | 4.2 |

**GC1: Independent Validation**

During the GoNorth expedition in late October 2022 (Salganik et al., 2023), we used the collected ice cores to evaluate our developed IBD parameterizations within their potential applicability ranges, as well as the newly proposed age-related IBD climatology (**Fig. A8**).

[Figure]

**Figure A8.** Evaluation of the regional and local IBD parameterizations within their respective applicability ranges during the GoNorth expedition in late October 2022, including the results for **(a)** SYI and **(b)** FYI, respectively. The updated age-related IBD climatology is also shown. **Note** that **Weighing** represents the direct measurement of IBD using the hydrostatic weighing method. **Reg.** and **Loc.** are the regional and local scales, respectively. **IFB**, **SIT**, **SID**, **TOT**, and **IFB/SIT** represent ice freeboard, ice thickness, ice draft, total thickness, and ice freeboard-to-thickness ratio, respectively. **Tem**, **AV**, **Sal**, and **BV** are ice bulk temperature, air volume fraction, ice bulk salinity, and brine volume fraction, respectively.

**GC1: Pan-Arctic Application**

We combined AWI CS2 sea ice thickness product, the ice age field, and three regional IBD parameterizations based on sea ice parameters to provide preliminary insights into the pan-Arctic monthly SYI bulk densities and their updated thicknesses during the 2010 to 2024 freezing seasons (Fig. A9).

The incorporation of regional IBD parameterizations into the current AWI CS2 product resulted in more realistic variations in SYI bulk densities, leading to systematically higher SYI thickness estimates and a reduction in their seasonal growth rate and interannual trend. Future incorporation of pan-Arctic SYI bulk temperatures, along with more accurate sea ice ancillary parameters, will be more effective in retrieving SYI bulk densities and thicknesses.

[Figure]

**Figure A9.** Retrieved IBDs and thicknesses of the pan-Arctic SYI during the 2010 to 2024 freezing seasons based on the AWI CryoSat-2 sea ice thickness product (AWI CS2), including results using the regional IBD parameterizations (updated) and the original settings (sea ice type-weighted A10 climatology). The shaded areas denote mean ± one standard deviation and dashed lines indicate linear trends. **(a)** Multi-year monthly average estimates for IBD from October to April. **(b)** Multi-year monthly average estimates for ice thickness from October to April. **(c)** Freezing season average estimates for IBD from 2010/2011 (10/11) to 2023/2024 (23/24). **(d)** Freezing season average estimates for ice thickness from 2010/2011 (10/11) to 2023/2024 (23/24).

*GC2: Another significant concern regarding the methodology of this study involves the so-called "spatial scale adjustment." First, the issue with buoys is not solely about spatial coverage or the number of buoys deployed. Rather, it arises from the fact that the buoys were installed in ice that may not be representative of the broader area. Secondly, such a substantial adjustment requires a more detailed explanation than a brief mention with a reference to supplementary materials. The adjustment of 7 cm to the freeboard, which accounts for roughly a third of the average total freeboard during the observation period, is quite significant. According to the methodology, ice freeboard is calculated as the difference between ICESat-2 snow freeboard and buoy snow thickness. Since the buoy snow thickness remains unchanged, applying a 7 cm adjustment equates to adjusting the ice freeboard by 7 cm, which corresponds to approximately 65 cm of ice thickness. This can be shown considering the initial buoy ice thickness was around 100 cm, while the modal ice thickness from electromagnetic sounding was between 40-50 cm, giving a similar difference in ice thickness, as the adjustment equivalent. Such a considerable adjustment to the interpretation of ice thickness, which ultimately impacts the density estimates, warrants a more thorough explanation of both its value and physical meaning. Additionally, it is perplexing that instead of adjusting the relatively uncertain estimates of snow and ice thickness from the buoys, the ICESat-2 and airborne laser scanner data were adjusted instead and presented in this adjusted form. This 7 cm adjustment was applied across the entire seven-month observation period, but it is not clear why this adjustment was considered time-independent. If the buoys were installed in areas with unrepresentatively thick ice, it seems unlikely that this bias in ice thickness would remain constant throughout the ice growth season. Numerous observations indicate that by the end of the cold season, initially thinner ice can grow more rapidly due to thinner snow cover, making the differences between thick and thin ice less distinguishable. Furthermore, another issue arises from the fact that buoys were installed in areas with thicker ice, which is known to correlate strongly with snow thickness. Even toward the end of the season, ice with greater surface roughness tends to accumulate more snow compared to smoother, younger ice. This raises questions about the representativeness of the snow thickness measurements from the buoys. These considerations lead to a broader question regarding the interpretation of the seasonal changes presented in the study. How might these changes be affected by the unrepresentativeness of the buoy measurements?*

**GC2 Reply** **We have added more details on the spatial scale adjustment.** The spatial scale adjustment is designed to compensate for systematic differences in snow depth (rather than sea ice freeboard) between buoy sites and large-scale observations, as determined by **a comprehensive intercomparison** of observations from buoy, ice core, transect, and airborne/satellite measurements during the early study period (Table A3, Figs. A10−11). The primary supporting results summarized as follows: 1) The level ice freeboards from MCS-FYI, MCS-SYI, and large-scale observations were in close agreement during the early period (Table A3); 2) The buoys effectively recorded the seasonal accumulation of snow depth (i.e., linear trend) across a broad area (Figs. A11); 3) The mean snow depths derived from the buoys were ~7 cm higher than those derived from regional transects conducted between 5 and 11 October (Krumpen et al., 2020); 4) The use of ice core data as initial sea ice freeboard references for buoy sites (weighted by the SYI/FYI ratio) indicated that the values were ~7 cm higher than the initial negative differences

(**Fig. A10**). Please refer to "2.2.2 Spatial scale adjustment*"* and "Text S2. Feasibility of the spatial scale adjustment" for specific revisions.

During the initial phase of the MOSAiC expedition (October 2019), a comparison of MCS-FYI, MCS-SYI, and regional measurements (Krumpen et al., 2020) indicated that snow depth was the primary contributor to the total freeboard of level sea ice, and that their sea ice freeboards were very similar, despite significant differences in their ice thicknesses (up to 0.4 m) and spatial scales (local and regional), as shown in **Table A3**. Furthermore, the initial snow and ice thickness values at the SYI-dominated buoy sites were comparable to those observed in the MCS-SYI (**Fig. A11**). This suggests that the spatial heterogeneity of level ice freeboard between the buoy sites and the large-scale airborne/satellite measurements is likely to be small, despite the lack of direct freeboard measurements for either. A subsequent comparison of buoy snow observations at the DN and L-site scales from 5−11 October 2019 with corresponding regional transect measurements showed that buoy-derived mean snow depths (0.14 m) were **approximately 7 cm higher at both scales.** This highlights that snow conditions on buoy-observed ice layers may not accurately reflect the magnitude of snow depths over a broader measurement area (and vice versa). However, **the seasonal trends in mean snow depth at the buoy sites were closely aligned with the modal snow depths derived from transects over the CO floe** throughout the study period (**Fig. A11**), which covered a much larger area. This result indicates that the buoys effectively captured seasonal snow accumulation over level sea ice within their deployment area. Therefore, it can be concluded that the modal total freeboard derived from airborne/satellite measurements could be systematically adjusted by approximately 7 cm to align with buoy observations.

**The spatial adjustment term was further validated by using ice core data as the initial freeboard reference for the buoy sites.** In this context, we calculated the adjusted value at the corresponding regional scale as the sum of two terms: 1) the absolute value of the maximum negative difference between the original modal total freeboard and buoy-derived mean snow depth (**~0.4 m**, see **Fig. A10**), and 2) the initial freeboard reference obtained from the FYI and SYI cores in late October, determined by weighting the FYI/SYI ratios from the corresponding buoy sites (**~0.3 m**). The buoy-derived mean ice thickness and snow depth were comparable in mid-October (initial satellite record) and late October (initial ice core record), and the timing difference is not expected to significantly impact the reference freeboards. The final adjusted values were **approximately 7 cm at both spatial scales**, which aligned with those obtained from snow depth differences between the regional transects and buoy sites. This, in turn, suggests that the spatial scale difference between the two observations results from a systematic bias in snow depth rather than sea ice freeboard. Together with the ability of the buoys to capture seasonal snow variations, we ensured that the difference between the adjusted modal total freeboard and the buoy-derived mean snow depth reasonably approximated the sea ice freeboard across the buoy sites, thereby ensuring compatibility between the buoy array and the airborne/satellite measurements used in the IBD retrieval.

**Table A3.** Comparison of level ice measurements from MCS-FYI, MCS-SYI, and regional transects during the early phase of the MOSAiC expedition.

| Data | Time | Sea ice thickness (m) | Snow depth (m) | Total freeboard (m) | Sea ice freeboard (m) |
|---|---|---|---|---|---|
| Ice cores (MCS-FYI) | 27 Oct | 0.42 | 0.09 | 0.11 | **0.023** |
| Ice cores (MCS-SYI) | 28 Oct | 0.77 | 0.15 | 0.18 | **0.03** |
| Krumpen et al. (2020) | 5−11 Oct | 0.36 | 0.07 | 0.10* | **0.03*** |

**Note** that data marked with an asterisk (*) are based on modal total freeboard derived from ALS and ICESat-2 on 11 Oct 2019. Sea ice thickness and snow depth from Krumpen et al. (2020) are weighted modal estimates according to daily observation samples or transect distances from 5 to 11 October 2019.

[Figure]

**Figure A10.** Differences between modal total freeboard (adjusted and original) and buoy-derived mean snow depth at the DN (blue) and L-site scales (red), respectively. The yellow bar highlights the maximum negative difference between the original modal total freeboard and the buoy-derived mean snow depth.

[Figure]

**Figure A11.** Seasonal variations of sea ice and snow parameters from level ice components during the MOSAiC freezing season. **(a)** Sea ice thickness measurements obtained from buoys (blue, red, and magenta lines for DN, L-site, and CO scales, respectively), transects (black crosses, mode), and ice cores (green symbols). **(b)** Snow depth measurements recorded from buoys (blue and red lines for DN and L-site scales, respectively), transects (black crosses, mode), and ice cores (green symbols). **(c)** Total freeboard measurements derived from ALS & ICESat-2 (blue and red dots for DN and L-site scales, respectively) and ice cores (green symbols). The error bars represent the uncertainties of the modal total freeboard. **(d)** Snow bulk density measurements ascertained from snow pits (grey dots), with error bars denoting one standard deviation from all daily estimates. The red line indicates a linear fit and the shaded band labels the 95 % confidence interval. **Note:** The dashed lines in panels **(a)**, **(b)**, and **(c)** represent seasonal linear trends, with all statistical *P*-values less than 0.01.

*GC3: A third concern relates to the interpretation of in situ density measurements. Based on the abstract from the coring dataset by Oggier et al. (2023), it appears that the sea-ice density was measured at constant laboratory temperatures rather than at in situ conditions. However, this is (1) not explicitly mentioned in the study, and (2) there is no description of how density was recalculated to reflect in situ temperatures, if such a correction was applied. If this adjustment was not made, any discussion of how brine volume evolution affects sea-ice density becomes less meaningful. Furthermore, the interpretation of the measured density shows a larger standard deviation from actual weighing than from the combination of ICESat-2 freeboard data and buoy snow and ice thickness. This seems to contradict previous estimates of uncertainties in density measurements (Hutchings et al., 2015; Pustogvar and Kulyakhtin, 2016). Specifically, weighing has consistently been found to be one of the most precise methods, while density estimates based on ice freeboard and thickness measurements tend to have standard deviations several times higher—by an order of magnitude—according to Hutchings et al. (2015), even when the thickness and freeboard were measured in situ. Given this prior evidence, it is somewhat surprising that the reported standard deviations suggest the opposite pattern in this study. A clearer explanation of these discrepancies would clarify the interpretation of these measurements.*

**GC3 Reply** In the revision, we have added detailed information on the MOSAiC coring measurements, included the physical interpretation of the IBD variations and the corresponding parameterization, and clarified the smaller standard deviation of our derived IBDs compared to previous studies. Please refer to "2.1.1 MOSAiC observations (ice core data)", "3.2 Seasonal variability of IBD during MOSAiC", "4.1 Uncertainties and limitations of IBD retrieval", and "4.2 Possible factors contributing to the IBD seasonality" for specific revisions.

**GC3: Details of the Coring Measurement**
Although the laboratory temperature (−15 °C) varies from the in situ temperature, Pustogvar and Kulyakhtin (2016) demonstrated that the hydrostatic weighing method effectively mitigates the density errors associated with brine drainage due to ambient temperature. Even under extreme conditions, such as a 19 % brine loss, the measurement error remains as low as 2 %, while the mass/volume method can introduce errors of up to 20 %. Specifically, the hydrostatic weighing method can effectively limit the loss of brine during ice sectioning at low laboratory temperatures compared to in situ conditions, where brine losses can be greater in warm ice. The use of kerosene coating further helps to minimize brine loss by occupying the spaces at the core interfaces, thus improving the accuracy of the measurements. Salganik et al. (2024) found little impact of brine loss on hydrostatic weighing density measurements by comparing sea ice densities reconstructed from in situ temperature and freeboard/thickness, also exhibiting a similar variation pattern. **Therefore, despite measurements being conducted at a constant laboratory temperature, this method provides highly reliable and accurate results, facilitating a meaningful discussion of brine volume evolution and its impact on sea ice density.** Please refer to "2.1.1 MOSAiC observations (ice core data)" for specific revisions.

**GC3: Standard deviation of IBD results**

**We would like to clarify that the regional IBD results did not show a lower standard deviation than those of direct measurements.** Except in the initial stages, the standard deviation of the regional IBDs exceeded twice that of MCS-FYI and MCS-SYI. The standard deviation of all IBDs at the DN scale was comparable to that of MCS-FYI, attributable to the sample difference and the early rapid increase. Considering estimates from different seasons independently (**Table A1**), the standard deviation at the regional scale was significantly higher compared to that of MCS-FYI and MCS-SYI.

The relatively high standard deviation of the hydrostatic equilibrium-based IBDs in Hutchings et al. (2015) can be attributed to the inclusion of mostly deformed sea ice in their measurements, which, combined with a relatively high spatial sampling resolution, may result in a failure of the hydrostatic equilibrium assumption at the local scale. The explanation they give in the text is: *"Bulk densities estimated from ice and snow measurements along 100 m transects were high, and likely unrealistic as the assumption of isostatic balance is not suitable over these length scales in deformed ice"*. In contrast, the IBD results of this study represent the **average state of the level ice layers across the buoy sites**, resulting in significantly lower IBD variability compared to previous studies. Moreover, our analysis is focused on spatial scales of approximately 25 and 50 km radius, which facilitates the convergence of the hydrostatic equilibrium-based IBD estimation.

**For this point, the airborne observations of Jutila et al. (2022) provided a good example of the significant impact of spatial scale on IBD calculations.** Based on a ~30 km airborne measurement profile, the IBDs derived from sampling at 5−6 m spatial resolution were mainly between 800 and 1000 kg m$^{-3}$ with considerable spatial variability. When weighted averaging over the length scale of 800 m was used, the resulting IBD effectively converged to about 880 to 920 kg m$^{-3}$, significantly reducing the potential non-physical variability of the retrieved IBDs.

We also applied the hydrostatic equilibrium method (also known as the freeboard/draft method) to calculate the IBDs of MCS-FYI and MCS-SYI using the seawater and snow bulk densities defined in Section 2.2.3. Despite the small mean differences, the results showed **considerable variability compared to those obtained by the weighing method**, with deviations of $7 \pm 35$ kg m$^{-3}$ for FYI and $1 \pm 21$ kg m$^{-3}$ for SYI, further highlighting the challenges of assuming hydrostatic equilibrium at local scales.

Please refer to "4.1 Uncertainties and limitations of IBD retrieval" for specific revisions.

**GC3: Physical Explanation**

**Figure A12** illustrates the seasonal variation in air temperature, ice bulk temperature, ice bulk salinity, brine volume fraction, and air volume fraction during the MOSAiC freezing season. According to the theoretical relationship (Timco and Frederking, 1996), the recorded sea ice temperature and bulk salinity variations cannot account for the seasonality of IBD over the study period (IBD variation < 5 kg m$^{-3}$).

**MCS-FYI.** The significant increase in IBD during the early stages is closely related to the **reduction in the air volume fraction within the ice layer (Fig. A12**), as the conversion of liquid water to solid ice in closed brine pockets causes volume expansion, compressing trapped air bubbles and pushing gas into the brine solution as temperatures fall (Crabeck et al., 2016; Crabeck et al., 2019). The rapid thickening of the sea ice may also have significantly regulated the brine volume and air volume fractions within the ice layer. For example, the newly formed ice layer will gradually transition from a loose granular structure to a dense columnar structure, resulting in a more compact layer (Oggier and Eicken, 2022). Indeed, the air volume fraction has been shown to be the most important factor influencing IBD, as it is much less dense than sea ice and brine (e.g., Timco and Frederking, 1996; Pustogvar and Kulyakhtin, 2016).

**MCS-SYI.** The MCS-SYI showed a relatively stable IBD with an increase of only ~3 kg m$^{-3}$ during the early stages, **corresponding to its relatively low thickness growth rate (~9 % per month) and low porosity** with a brine volume of ~1 to 4 % and an air fraction of ~0.8 to 1 %.

**Regional buoy sites.** The regional IBD results also showed an increasing trend during this period (~14 kg m$^{-3}$ per month), similar to the MCS-FYI, but the limited sample and relatively high uncertainty should be noted (average of ~3 %). Thus, we further introduced hydrostatic equilibrium-based modeled IBD estimates based on the seasonality of various ice parameters observed during MOSAiC (**GC4: Regional Modeled IBD**).

The results further indicate a significant difference in ice porosity between local (MCS-SYI) and regional (SYI-dominated buoy sites) scales. In particular, Hornnes et al. (2024) found that the spatial heterogeneity of SYI porosity was much greater than that of FYI during the GoNorth expedition in October 2022, and observed similar air volume fractions in FYI (2.5 ± 0.5 %) and SYI cores (2.8 ± 0.5 %).

Please refer to "3.2 Seasonal variability of IBD during MOSAiC" and "4.2 Possible factors contributing to the IBD seasonality" for specific revisions.

[Figure]

**Figure A12.** Seasonal variation of **(a)** sea ice bulk salinity, **(b)** air temperature, **(c)** brine volume fraction, **(d)** sea ice bulk temperature, and **(e)** air volume fraction during the MOSAiC freezing season. The red boxes in panel **(b)** indicate the typical warm air intrusion events with air temperature increasing to near or above −5 °C. The dashed line in panels **(a, c)** represents the seasonal linear trend ($P < 0.01$), while that in panel **(e)** denotes the power function fit ($R^2 = 0.78$, $P < 0.01$).

**GC4: Regional Modeled IBD**

Given the relatively high uncertainties and sparse sampling associated with the increasing trend of regional IBDs in the early stage of MOSAiC, we introduced hydrostatic equilibrium-based modeled IBDs based on a simple trend assumption to further elucidate IBD seasonality. Specifically, we set the initial sea ice thickness, snow depth, and total freeboard in mid-October to values of 0.36, 0.07, and 0.10 m, respectively, which are expected to represent typical ice conditions for MOSAiC level ice (Table 1). We then calculated the average seasonal trends for these parameters during MOSAiC as ~0.20, 0.01, and 0.03 m per month according to Fig. A11 and extrapolated these initial values to late April. Finally, we calculated the so-called modeled IBD using the hydrostatic equilibrium equation and the same snow bulk and seawater densities as defined in this study. It should be noted that the modeled values can only give the approximate seasonal signal of the IBD, completely filtering out possible IBD changes on daily and weekly scales. Overall, the derived modeled values also support the early increasing trend of IBD depending on the seasonality of the different observed ice parameters during MOSAiC.

[Figure]

**Figure A13.** Hydrostatic equilibrium-based modeled IBD estimates (M-IBD) during the MOSAiC freezing season.

Please refer to "Text S4. Hydrostatic equilibrium-based modeled IBD estimates during MOSAiC" and "3.2 Seasonal variability of IBD during MOSAiC" for specific revisions.

**Specific comments**

*Line 21: The phrasing suggests that this study considers deformations, brine, and gas inclusions in its estimates of ice density, but this does not appear to be accurate. It would be helpful to clarify this point. Impact of deformed ice, brine and air inclusions were not analysed in this study.*

**Reply:** As mentioned in **GC3: Physical Explanation**, we have added several physical factors to the analysis. **Revised to:** *"which are influenced by factors such as the age, temperature, brine content, air intrusion, and growth rate of sea ice."*

*Line 23: The term "Distributed Network scale" might not be clear to readers unfamiliar with the MOSAiC Distributed Network. Could you specify the scale or provide additional context to make it more accessible and inclusive?*

**Reply:** Thanks for the suggestion. **Revised to:** *" To estimate regional IBD, we integrated sea ice thickness and snow depth observations from ice mass balance buoys, snow density measurements from snow pits, along with high-resolution along-track freeboard data from airborne laser scanning (ALS) and the Ice, Cloud, and land Elevation Satellite-2 (ICESat-2), at both the Distributed Network (DN) and L-site scales with a radius of 50 and 25 km from the research vessel Polarstern, respectively."*

*Line 24: The statement regarding ice density estimation is not entirely accurate. Ice density was calculated separately from a combination of buoy ice thickness and altimetry freeboard data and from ice coring data. It might be helpful to differentiate these methods more clearly.*

**Reply:** We have clarified this. **Revised to:** *"Assuming hydrostatic equilibrium, regional IBDs were estimated for the buoy-deployed ice cover, including 11 sites in second-year ice (SYI) and 4 sites in first-year ice (FYI). The regional IBDs were then compared with measurements obtained directly from the MOSAiC Main Coring Sites."*

*Line 28: The assertion that MOSAiC ice floes predominantly consisted of second-year ice needs further support. (1) Several studies suggest otherwise, and (2) the term "predominantly" is vague in terms of areal fraction. Additionally, it would be useful to clarify what is meant by "MOSAiC ice floes"—how many floes are being referenced, and which ones specifically?*

**Reply:** As mentioned above, we actually retrieved the IBDs that are representative of the buoy-deployed ice layers, which we have thoroughly revised in the text. **The MOSAiC ice floes are defined as a cluster of ice floes within 50 km of the CO.**

*Line 32: The ratio of ice freeboard to ice thickness cannot serve as a direct measure of ice density since they are mathematically correlated. Density is typically used to convert ice freeboard to ice thickness. Meanwhile, such relationship may be complicated by the presence of deep snow or absence of accurate snow observations. The latter is directly related to the results of this study. Please clarify the interpretation here.*

**Reply:** In the revision, we demonstrated the validity of the ice freeboard-to-thickness ratio for predicting IBD based on physical analysis and independent validation (**GC1: Physical Factors** and **GC1: Independent Validation**). We also mentioned that the fitting performance of the regional IBD parameterizations does not represent absolute robustness because they are not strictly independent of the input parameters. Meanwhile, the IBD parameterization based on ice

freeboard-to-thickness ratio can be used for the so-called radar (laser) freeboard to thickness conversion to simultaneously estimate IBD and sea ice thickness from space (when other auxiliary parameters are known), which can facilitate sea ice remote sensing (**GC1: Pan-Arctic Application**). **Revised to:** *"Considering a variety of sea ice parameters and other physical factors, we proposed several potential IBD parameterizations, with relative biases ranging from ~0 to 0.9 % for SYI and from 0.1 to 2.1 % for FYI when compared with independent coring measurements."* Please refer to "3.2 Seasonal variability of IBD during MOSAiC", "4.3 Evaluation of IBD parameterizations", and "4.5 Implications of IBD parameterizations for the sea ice thickness retrieval" for specific revisions.

*Line 34: The statement that second-year ice (SYI) dominates the central Arctic Ocean contradicts existing studies, such as Tschudi et al. (2019, 10.5194/tc-14-1519-2020), which show that first-year ice (FYI) is the primary ice type in the Arctic. Could you provide more context or evidence to support this claim?*

**Reply:** We have removed this inappropriate statement.

*Line 59: The referenced study from Timco and Weeks (2010) offer an overview of sea-ice properties but do not draw any specific conclusions about the heterogeneity or seasonal evolution of ice density. They only presented ice density at various temperatures (assuming zero gas volume) and discussed a broad range of density measurements without the kind of conclusions mentioned in your study. Could you clarify this?*

**Reply:** We would like to clarify that this review did discuss the internal structure and porosity of sea ice.

*Timco, G. and Weeks, W.: A review of the engineering properties of sea ice, Cold Regions Science and Technology, 60, 107-129, https://doi.org/10.1016/j.coldregions.2009.10.003, 2010.*

*Line 63: The claim that sea ice density influences permeability, meltwater infiltration, and biogeochemistry is quite bold. The cited studies may show correlations, but it seems inaccurate to assert that density directly defines these processes. Permeability, for example, is largely governed by ice age, microstructure, and brine volume. Could this section be reframed to reflect the complexity of these relationships? Also, the term porosity is used in unconventional and inconsistent way. In most cases it is used as equivalent to gas volume fraction (line 440), while typically it should combine both gas and brine volume fractions. Therefore, please explain how gas volume is influencing permeability.*

**Reply:** We rephrased the sentence, and we defined the ice porosity in the text as the sum of the brine and air volume fractions. **Revised to:** *"IBD is regarded as a pivotal variable for the parameterization of sea ice thermodynamic and dynamic modelling (Ono, 1967; Wang et al., 2021) and indirectly involved in key processes such as sea ice permeability, summer meltwater infiltration, and biogeochemical cycling in the Arctic Ocean (Perovich et al., 2021; Angelopoulos et al., 2022)."*

*Line 66: If by winter you mean a period from December to February, this is not accurate statement. Most referenced measurements were collected in winter and spring.*

**Reply:** We rephrased the sentence. **Revised to:** *"However, the spatial representativeness of direct density measurements is limited by sparse sampling, especially during the early ice growth period (Timco and Frederking, 1996; Hutchings et al., 2015)."*

*Line 71: This (small errors of weighing) was known way before the referenced publication.*
**Reply:** We rewrote the sentence and cited more classical literature. **Revised to:** *" In terms of measurement accuracy, the hydrostatic weighing method has been proven to capture the natural variability of IBD most effectively (Butkovich, 1955; Nakawo, 1983; Pustogvar and Kulyakhtin, 2016). "*

*Line 90: The robustness of the parameterizations is limited by the assumption of hydrostatic equilibrium, which has been shown to be less precise (Hutchings et al., 2015). For instance, Jutila et al. (2022) propose density values above 950 kg/m³ for relatively thin ice, but such values are unrealistic, as they have not been validated by in situ measurements. How do you account for these limitations?*
**Reply:** It is important to note that Hutchings et al. (2015) investigated a ~100 m transect covering a large area of deformed ice. They suggested that the observed unrealistic density values may have arisen due to the hydrostatic equilibrium assumption being inappropriate at these length scales over deformed ice. Jutila et al. (2022) also incorporated numerous deformed units, which may partly explain the estimated relatively high density for thin ice. In contrast, our focus is exclusively on level sea ice over tens of kilometers, where the hydrostatic equilibrium assumption remains valid, as discussed in detail in **GC1** and **GC3**. This also explains why the standard deviation of our IBD results is considerably smaller than that of previous studies. Meanwhile, the predictive accuracy of the regional IBD parameterizations based on sea ice parameters even outperformed those derived from the MOSAiC core samples, further supporting the validity of our IBD calculations (**GC1: Independent Validation**). Please refer to "3.4 Potential parameterization for IBD" and "4.3 Evaluation of IBD parameterizations" for specific revisions.

In the revision , we discussed in detail the uncertainties and limitations of the IBD calculation and its parameterization. In the *Conclusion*, we also highlighted several caveats and shortcomings of this study. Please refer to "4.1 Uncertainties and limitations of IBD retrieval" and "5 Conclusion " for specific revisions.

*Line 101: Most of laser scans were significantly smaller than the linear scale of tens of kilometres (Hutter et al., 10.1038/s41597-023-02565-6). Could you clarify the scale?*
**Reply:** Clarified. **Revised to:** *"Moreover, repeated airborne laser scanning (ALS) measurements yielded high-resolution along-track total freeboard data (Hutter et al., 2023a,b), uncovering sea ice elevation characteristics ranging from the local ice floe (< 5 km from Polarstern) to regional transects of tens of kilometers."*

*Line 132: The exclusion of melt ponds from the sampling is not straightforward, as refrozen melt ponds are a substantial part of the ice cover, covering around 20-30% area. And their smooth surface also influence the amount of snow which may accumulate above such ice type. Additionally, surface conditions are not the primary issue when it comes to hydrostatic balance.*

*The representativeness of the ice thickness at the buoy sites is more important, and this is not adequately discussed. How does excluding ridges and deformed ice improve representativeness?*

**Reply:** According to the buoy installation reports and initial surveys, the IMBs were indeed deployed at sites initially devoid of melt ponds. In addition, the temperature profiles recorded by these IMBs did not show any anomalous signals that would indicate potential refrozen melt ponds. In terms of representativeness, the buoy deployment sites effectively recorded thermodynamically driven ice growth, representing the level ice features in their deployment areas (Koo et al., 2021). **Revised to:** *"The SIMBAs and SIMBs were mainly deployed on level ice, free from melt ponds, to ensure the reliability of the buoy data".* **Added** *"In particular, buoy deployment sites have successfully recorded thermodynamically driven ice growth, effectively representing the level ice features in their deployment areas (Koo et al., 2021)."*

We also acknowledge that the buoy deployment area contained many thick SYIs, and given the spatial heterogeneity of MOSAiC ice floes, it is difficult to fully represent ice conditions across the DN scale. In the revision, we have emphasized that the ice thickness and snow depth measured by the buoys are relatively thicker compared to regional observations. This is also why the spatial scale adjustment is needed. Please refer to "3.1 Integrated observations of sea ice and snow during MOSAiC freezing season" for specific revisions.

**Line 140:** *You state that 27% of the buoys were in FYI and 73% in SYI, but Kortum et al. (2024, 10.5194/tc-18-2207-2024) showed equal fractions of FYI and SYI based on freeboard measurements. How was this potential unrepresentativeness of buoy sites handled? Moreover, SYI is not typically associated with ice deformation when thinner FYI is present, which may influence evolution of its areal fraction.*

**Reply:** Our IBD results only represent the average ice condition across the buoy deployment sites, as described in **GC1**. We have thoroughly revised the description of sea ice types in the text.

**Line 141:** *First, does this influence your results? Probably not. Also, exceptionally large ice growth should be quantified of important. Illustration shows that platelet ice presumably contributed to 10-20 cm ice growth which later disappeared. Is it relevant for this study?*

**Reply:** Agree and this sentence was deleted.

**Line 173:** *There's an inconsistency here—earlier, you stated that 27% FYI was representative in an SYI-dominated area, yet here, 43% FYI is also considered representative. Meanwhile, Kortum et al. estimated this fraction as 50%. Could you clarify this discrepancy?*

**Reply:** As mentioned earlier, we have clarified that the 27% FYI is represented by the selected buoy sites rather than the entire DN spatial domain. In the revision, we have emphasized the ice-type composition of the MOSAiC ice floes as revealed by large-scale airborne observations (Kortum et al., 2024). Also, snow bulk density only serves as an auxiliary data for the IBD retrieval, contributing a relatively low error (less than 5%) . **Revised to** *"The snow pit records used contained approximately 12 % refrozen lead, 43 % FYI, and 45 % SYI (Macfarlane et al., 2021; Nicolaus et al., 2022), which well reflects the seasonal variability of snow properties under different ice conditions during MOSAiC. The ice type composition of the snow pits also closely matches the FYI/SYI ratio (~50%) derived from large-scale airborne observations during MOSAiC (Kortum et al., 2024)."*

*Line 179: It would be more robust to demonstrate the representativeness of the selected sites by comparing their ice thickness or freeboard with large-scale observations, rather than just stating they were selected for representativeness.*

**Reply:** Agreed. **Removed** *"and were sufficiently representative of the FYI and SYI properties during MOSAiC".*

*Line 182: Please clarify what is meant by "ice porosity." What does it include?*

**Reply:** In this study, ice porosity was considered as the sum of brine and air volume fractions. **Revised to:** *"These datasets provided detailed insights into seasonal variations of ice temperature, salinity, density, and ice porosity (brine and air volume fractions)."*

*Line 183: According to Oggier et al. (2023), salinity was measured directly from the sample after weighing, not interpolated from another core. Also, could you explain how brine volume and density were estimated for the measured temperatures?*

**Reply:** We would like to clarify that the density values were indeed interpolated to the depth of salinity measurements, as described in Oggier et al. (2023): *"The depth estimates assume that the total length of all core sections is equal to the measured ice thickness. Each core section has the value of its practical salinity (14), isotopic values (16, 17, 18) (Meyer et al., 2000), as well as sea ice temperature (19) and ice density (20) interpolated to the depth of salinity measurements."*

**We have added detailed descriptions of the ice core parameters. Revised to:** *"Of these, ice temperature was measured in the field using thermometers at 5-cm vertical intervals; ice salinity was determined from melted cores using a conductivity meter at a 5-cm resolution; ice density was derived in the freezing laboratory at an ambient temperature of –15 °C using the hydrostatic weighing method. The temperature and density results were interpolated to match the depth of the salinity measurements, providing a continuous measurement profile throughout the core sample. Assuming disconnected air and brine pockets, brine volume and air volume fractions were then calculated from measured ice salinity, temperature, and density using the equations of Cox and Weeks (1983) for cold ice and Leppäranta and Manninen (1988) for ice warmer than –2 °C, as detailed in Salganik et al. (2024). "*

*Line 189: The statement that some cores were longer and might come from rafted ice, leading to substantially different densities, is not supported by the data from Oggier et al. (2023). Please provide evidence for this assertion.*

**Reply:** We would like to clarify that the FYI cores from 2020-03-21 and 2020-04-14 are labeled as rafted ice. Our focus is specifically on obtaining bulk densities from level ice cores and comparing them to regional results (only level ice). In the revision, we have **removed the statement about substantially different densities**. **Revised to** *"To ensure reasonable comparability of IBD estimates between coring measurements and regional integration for the level sea ice (see Section 3.2), core-based IBD records from rafted ice associated with deformation processes were excluded."*

*Line 268: The ice pack does not consist solely of level ice. Please rephrase this statement, as ridges and deformed ice make up a substantial portion of the ice cover, as shown in transect observations. Also, how can you be certain that modal snow freeboard represents the same level*

*ice described at the buoy sites? A simple comparison of ice thickness distribution or mean, modal and median values could be also useful. What if ice consists of both FYI and SYI level ice, with a bimodal thickness distribution, which is not seen in snow freeboard data?*

**Reply:** We have revised and clarified the above questions accordingly.

1) *The ice pack does not consist solely of level ice.* Agreed and rephrased this statement. **Revised to** *"The MOSAiC ice floes have considerable spatial heterogeneity, covering deformed and level sea ice, pressure ridges, as well as refrozen melt ponds (Nicolaus et al., 2022). To estimate the regional integrated IBD, it is first needs to obtain the total freeboard that is representative of the level ice portions to ensure compatibility with the selected IMBs, which were deployed exclusively on unponded level ice."*

2) *Also, how can you be certain that modal snow freeboard represents the same level ice described at the buoy sites?* This is why we made the so-called spatial scale adjustment, as mentioned in **GC2 Reply**. Please refer to "2.2.2 Spatial scale adjustment " for specific revisions.

3) *A simple comparison of ice thickness distribution or mean, modal and median values could be also useful. What if ice consists of both FYI and SYI level ice, with a bimodal thickness distribution, which is not seen in snow freeboard data?* We would like to clarify that, based on high-resolution IceBird airborne measurements, which encompass both level and deformed FYI, SYI, and MYI, the modal total freeboards of all ice types closely resemble the mean total freeboards of the level ice fractions alone (**Fig. A14**). Furthermore, the total freeboard distributions obtained from ICESat-2 and ALS during MOSAiC both exhibited distinct unimodal characteristics, as did the ice thickness distributions derived from the transects. Several studies have shown that the typical regional characteristics of sea ice freeboard/thickness follow log-normal or negative exponential distributions (Haas, 2010; Farrell et al., 2011; Ricker et al., 2015; Petty et al., 2016; Landy et al., 2019; Koo et al., 2021), despite the inclusion of both deformed and level ice, as well as first-year and multi-year ice. Please refer to "Text S1. Feasibility of the modal approach" for specific details.

[Figure]

**Figure A14.** Comparison of the mode of the total freeboard distribution (encompassing all surface types) with the mean total freeboard of level ice, derived from AWI IceBird measurements conducted in April 2017 and April 2019. The example from 2 April 2017 illustrates: **(a)** the total freeboard distribution, **(b)** the measurement area, and **(c)** the total freeboard profile. **(d)** Results for each measurement date include the modal freeboard for all surface types, mean freeboard for level ice, and the relative percentage difference (RPD) between the two freeboard values, as well as the level ice fraction (at least 20%).

*Line 287: What are "MOSAiC ice floes, type II"? Could you provide more information on this classification? If you refer to the types of uncertainties, then they are described after this part of the manuscript.*

**Reply:** We have provided additional clarification and enhanced coherence. Please refer to "2.2.1 Modal total freeboard derived from ALS and ICESat-2 measurementst " for specific revisions.

**Revised to:** "*ascertaining the modal total freeboard, which is derived from the mean of the five highest frequency peaks in the total freeboard distribution, with the standard deviation quantifying the **uncertainty (Type I)** introduced by these peaks.* →

"*This mean difference was also considered to represent the **uncertainty (Type II)** of the modal total freeboard, stemming from the measurement methods and the spatial heterogeneity of the MOSAiC ice floes.*" →

"*The final modal total freeboard uncertainty was then set as **the sum of the Type I and Type II uncertainties**.*"

*Line 291: The 7 cm adjustment is a significant point, yet it is only briefly mentioned. Could you explain in more detail how this value was estimated and why it was applied as a time-independent adjustment throughout the seven-month period?*

**Reply:** We clarified this in **GC2**, see above. **Added:** *"2.2.2 Spatial scale adjustment"*. Please refer to "2.2.2 Spatial scale adjustment" and "Text S2. Feasibility of the spatial scale adjustment" for specific revisions.

*Line 294: How is this possible if you only use a single source of snow thickness from buoys? Here you attempt to match ice freeboard calculated as difference between ICESat-2 snow freeboard and snow thickness from buoys. You do not have data about large-scale snow thickness (line 314). Therefore, you lift the snow and ice freeboards together which leads to increased estimate of ice thickness which fits better the initial buoy ice thickness. From these estimates of ice freeboard and thickness you estimate the ice density. Therefore, it is not clear what do you mean by the snow depth difference between satellite and buoys sites, as second one is unknown. Similarly, this is an important assumption which cannot even be generalized for future campaigns. It deserved not to be described in supplementary materials.*

**Reply:** Please refer to our **Overall Response** above, as well as our response to **GC2** for further details.

*Line 315: While seawater density might not be the primary source of uncertainty, why not use directly measured seawater salinity and temperature from the expedition? Why was the chosen reference used instead?*

**Reply:** The selected reference is highly representative and widely used in hydrostatic equilibrium calculations for Arctic sea ice. Furthermore, we found that this fixed value aligns well with the mean density estimates from the upper 10-m layer recorded by buoys and CTDs during the MOSAiC expedition (Hoppmann et al., 2022). The density range reported in their study (~1023.5 to 1026.5 kg m$^{-3}$) has a minimal impact on our IBD calculations, with a deviation of no more than 0.5 kg m$^{-3}$ (0.05 %). Therefore, the use of this representative value remains valid and offers a reasonable approximation for our analysis. These data were not used because the combined time series they form, as well as the spatial features represented by the mean values, may not be compatible with the deployment area of the IMBs we selected, potentially introducing additional processing errors.

*Line 317: Again, more evidence is needed to support the claim that the MOSAiC Distributed Network (DN) was dominated by SYI. How this is possible when the measured by ICESat-2 snow freeboard in autumn was close to 10 cm? Was it representative for SYI, especially SYI similar to ice where buoys were installed? Because buoys had ice freeboard around 10 cm and snow thickness of 13 cm, giving the snow freeboard way above 10 cm.*

**Reply:** In the revision, we no longer claim that our IBD results are representative of the ice conditions across the MOSAiC Distributed Network (DN) due to the spatial heterogeneity of MOSAiC ice floes. **Revised to:** *"This method provides an estimate of the regional bulk density of level sea ice mixed with SYI (11 buoy sites) and FYI (4 buoy sites)."*

*Line 318: Krumpen et al. (2020, 10.5194/tc-14-2173-2020) report that the modal ice thickness in L-sites was only 40-50 cm, significantly thinner than at buoy sites. Why is this discrepancy not addressed?*

**Reply:** In the revision, we highlighted the significant differences in thickness and snow depth of level sea ice between the buoy sites and regional transects during MOSAiC. Please refer to "2.2.2 Spatial scale adjustment " and "3.1 Integrated observations of sea ice and snow during MOSAiC freezing season" for specific revisions.

*Line 359: The ice thickness range of 0.6–2.0 m is stated as representative for level ice, yet Krumpen reports much thinner modal thicknesses. Early transects from Itkin et al. (2023,10.1525/elementa.2022.00048) also mainly show modal ice thickness around 0.5 m. Could you provide clarification on this?*

**Reply:** Agreed, we have clarified this. **Revised to:** *"The mean ice thicknesses recorded by the buoys at the DN and L-site scales were 1.41 ± 0.32 and 1.30 ± 0.33 m, respectively (Fig. 5a and Table 2), whereas the average values for individual buoys ranged from ~0.6 to 2 m (Fig. S1). Due to the deployment strategy, the buoy-derived sea ice thicknesses were generally thicker than the regional transects conducted in the early stage of MOSAiC (Krumpen et al., 2020, Table 1)."*

*Line 370: Arbitrary selection of coring sites does not necessarily make them representative in terms of ice thickness. This should be better supported.*

**Reply:** We removed this statement.

*Line 375: You obtain level ice freeboard using modal values of snow freeboard. Why is a different approach applied here to the transect snow thickness? Itkin et al. (2023) showed a snow thickness distribution with initial modal snow thickness around 10 cm. Similarly, the initial snow thickness above level ice in their paper is also close to 10 cm. Therefore, what is the meaning of the initial average 24 cm from transects?*

**Reply:** In the revision, we further calculated the modal snow depths for the local transects and reported the differences relative to the buoy-derived mean snow depths, as shown in **Fig. A11**. Please refer to "3.1 Integrated observations of sea ice and snow during MOSAiC freezing season" for specific revisions.

*Line 377: It appears that specific conditions at the coring sites resulted in shifting the ICESat-2 freeboard upwards. However, ICESat-2 and ALS initially showed freeboards close to 10-15 cm, suggesting the buoys were in areas with unrepresentative snow thickness. Could this be clarified?*

**Reply:** Agreed, we have clarified the differences in snow depths between buoy sites and large-scale observations, which also accounts for the spatial scale adjustment, as discussed in **GC2**. **Revised to** *"Snow depths over level sea ice from the transects (mode), MCS-FYI, and MCS-SYI were generally thinner than those from the buoy array observations."* Please refer to "2.2.2 Spatial scale adjustment" and "3.1 Integrated observations of sea ice and snow during MOSAiC freezing season" for specific revisions.

*Line 384: This is a misinterpretation of the study from Itkin et al. (2023) You claim that buoy snow depth was close to transects. Yet, in the Fig. 9 of Itkin et al. (2023) the shown snow depth above level ice is below 10 cm.*

**Reply:** We would like to clarify that our conclusion is that the buoys exhibit similar seasonal trends in snow depth to the local transects, rather than matching their magnitudes (Fig. A11). **Revised to** *"In particular, comparisons with transect data, which have a more comprehensive measurement coverage, showed that buoy array measurements are sufficiently representative to capture __seasonal snow accumulation on level ice over a wider area__".*

*Line 437: You mention that the DN is dominated by SYI, but later state that MOSAiC ice floes were predominantly FYI. This contradicts previous observations (e.g., Guo et al., 10.5194/tc-17-1279-2023; Kortum et al.). Could you clarify what exactly is being referred to as "MOSAiC ice floes"?*

**Reply:** Thank you for providing these very valuable findings. In this revision, we no longer assert that any of the IBD results represent MOSAiC ice floes (defined as a cluster of ice floes within 50 km of the CO). Instead, we refer to the buoy-deployed ice layers (SYI-dominated), as outlined in **GC1** and **GC2**. **Revised to** *"Figure 6 depicts the seasonal variation of IBD from both **regional integration (buoy sites)** and **sampling point (MCS)** during the MOSAiC freezing season."*

*Line 437: Your estimate of ice density includes 7 cm freeboard adjustment; how can you discuss these minor differences between measured FYI and SYI density considering buoy method uncertainty close to 50 kg/m³? Moreover, there is not any clear difference in winter FYI and SYI density based on weighing. How can potential difference in several kg/m³ can support whether FYI or SYI was dominating the area of investigations? You are saying that 910 kg/m³ is closer to 912 kg/m³, while 908 is closer to 905 kg/m³? A difference in 2 kg/m³ is beyond measuring uncertainties and spatial variability even for the best performing methods to measure ice density. No such claims can be made. Finally, it is not correct to average FYI density for the whole period of observation, as it includes a period of lower densities in autumn. If you exclude those lower values from the average value of 905 kg/m³, would you be able to see any measurable difference between two ice types?*

**Reply:** Agree and the sentence has been revised. **Revised to** *"Throughout the study period, the mean IBD values estimated at the DN scale (buoy sites), L-site scale (buoy sites), MCS-FYI, and MCS-SYI were $910 \pm 7$, $908 \pm 11$, $905 \pm 8$, and $912 \pm 2$ kg m$^{-3}$, respectively. The bulk densities recorded from the FYI and SYI cores were not obviously different except in the early stages. "*

*Line 443: in line 437 you claim that DN is dominated by SYI, while here you claim that MOSAiC ice floes were predominantly FYI. What are MOSAiC ice floes? This contradicts many other observations (Guo et al., Kortum et al.). MOSAiC Central Observatory was one of the thickest floes in the area with the largest SYI fraction, unlike surroundings (Krumpen et al., 2020), which is opposite to the presented claims.*

**Reply:** Agree and this statement has been removed. As mentioned above, our regional IBD results only represent the physical properties of buoy-deployed ice layers (dominated by SYI).

*Line 446: What is the basis of the claim of substantial sub-weekly ice density variability. This is a clear artefact of spatial variability of ICESat-2 tracks and corresponding freeboards which has nearly nothing to do with ice density changes.*

**Reply:** We acknowledge that trajectory changes in ALS/IS2 may introduce non-physical variations in the total modal freeboard, which could affect the derived IBD. Nevertheless, it is important to note that the ALS/IS2-derived modal freeboard generally agrees well with the buoy-derived mean snow depth, which may also be due to variations in snow depth (**Fig. A15**). In Section 4.1, we have **added:** *"Meanwhile, due to the irregular variations in airborne and satellite trajectories and the limitations of the buoys in capturing mean variations of snow depth over their deployment area, it must be emphasized that the retrieval method may not be able to fully resolve synoptic scale IBD variations and thus introduce some non-physical signals, which is a major challenge for estimating IBD using spatially non-overlapping data. Improving the buoy deployment strategy through comparative evaluation with large-scale observations will allow for more robust multi-source data matching to more accurately retrieve IBDs."*

[Figure]

**Figure A15.** Buoy-derived mean snow depth versus airborne/satellite-derived modal total freeboard during the MOSAiC freezing season, including results at both the DN and L-site scales. The statistical metrics include the Pearson correlation coefficient (*R*), the statistical test *P*-value, and the total number of samples (*Num*).

*Line 448: You are saying that FYI had newly formed SYI fractions at the ice bottom? What does it mean?*

**Reply:** This statement has been removed and sea ice bulk salinity and brine volume fraction data have been added. **Revised to** *"Meanwhile, MCS-FYI exhibited a slow brine discharge with a decreasing trend of −0.2 ppt per month for its bulk salinity (Fig. 7a) and 0.5 % per month for its brine volume fraction (Fig. 7c). In contrast, the bulk salinity of MCS-SYI showed an increasing*

*trend of 0.3 ppt per month, which is **closely related to the formation of salty new ice layers (similar to FYI) at the base.***"

*Line 451: First, which layers are you referring to? These layers may contain more brine than which other layers? For FYI, how is it possible, if ice is gradually getting colder and less saline upon ice growth? The brine volume is easy to estimate, and it was estimated in several publications, there is no need for such speculations. And, following the modelling study from Griewank and Notz (2008), brine volume is decreasing in Autumn, contributing to decreasing density if gas is not considered. And if you discuss SYI, there were no new layer under SYI in autumn, only in winter, which can be clearly seen in Fig. 4a. Only since cold winter month, SYI with larger snow and ice thickness may have a substantial thickness gain.*

**Reply:** As answered in **GC3: Physical Explanation,** we re-analyzed the seasonal variations in IBD that were attributed to air reduction, decreasing ice bulk temperatures, and the high growth rate of ice thickness during this period. Please refer to "3.2 Seasonal variability of IBD during MOSAiC " and "4.2 Possible factors contributing to the IBD seasonality" for specific revisions.

*Line 461: What does it mean? There is a clear relationship between temperature, salinity, and brine volume which explain why saltier FYI has a stronger vertical gradient of brine volume. Also, most of SYI formed under the remaining 0.5 m at the surface is a new ice, with its physical properties close to FYI. Therefore, there is no surprise that FYI and SYI are eventually becoming more similar. Homogenization is a too broad description of these processes.*

**Reply:** We added analyses of the seasonal evolution of sea ice bulk temperature, bulk salinity, brine volume fraction, and air volume fraction (**Fig. A12**). Please refer to "3.2 Seasonal variability of IBD during MOSAiC" for specific revisions.

*Line 462: What is the evidence for these claims? You do not present ice salinities in this study. But if you check FYI bulk salinity from Oggier et al. (2023), it has decreasing trend very similar to the broadly used parametrization from Kovacs (1996, 10.21236/ada312027). Also, please explain in more detail the suggested process behind the initial density decrease. If you consider only brine, desalination leads to decrease of brine volume and corresponding density decrease, that is correct. But it cannot be significant only due to brine, the absolute values are less than 5 kg/m³. One can estimate brine volumes from the measured salinity and temperature and draw the same conclusion. Yet, the observed density change is much larger, and it was increasing, not decreasing (Fig. 5a).*

**Reply:** Agreed and the analyses as mentioned above has been inserted. Please refer to "3.2 Seasonal variability of IBD during MOSAiC " and "4.2 Possible factors contributing to the IBD seasonality" for more details of this specific revisions.

*Line 466: This is an example of overfitting. Density estimates from ICESat-2 has a large spread of around 30 kg/m³. The increasing trend in autumn is barely based on a single data point in Fig. 5a.*

**Reply:** We would like to clarify that the early stages did indeed show an increase of the regional IBD estimates (even after excluding the lowest value), as shown in **Fig. A2**. We combined the regional IBD estimates at both DN and L-site scales from late October to early December with a statistical test P-value of less than 0.05 and an R-value of 0.62, indicating no statistically meaningful overfitting. We also acknowledge the relatively high uncertainty and small sample size in the early stages; therefore, we additionally introduced the modeled IBD values and found a similar increasing trend (**Fig. A13**). In

particular, the seasonal and monthly averages of the regional IBD estimates effectively reduced some of the IBD noise and showed more pronounced seasonal variability. Please refer to "3.2 Seasonal variability of IBD during MOSAiC " and "Text S4. Hydrostatic equilibrium-based modeled IBD estimates during MOSAiC" for specific revisions.

*Line 498: Can you explain more specifically what is the mentioned scale-related density variability? You cover different areas in DN and L-site scales, but there is no clear difference between them following Fig. 5a. Also, the mean values are nearly identical, but this would not be possible without in situ measurements, providing less uncertain estimates. And what supports the well-known importance of ice density for ice thickness retrieval but based purely on your measurements? How do you validate thickness estimates?*

**Reply:** In the revised version, we highlighted the scale-related density variability between the MCS-SYI and regional buoy sites (SYI-dominated). We have demonstrated the validity of the proposed IBD parameterizations through independent validation, thus providing the potential to improve satellite retrievals of sea ice thickness. Through a preliminary application to the pan-Arctic SYI, we found that our regional IBD parameterizations result in systematically higher SYI thickness, lower seasonal growth rate, and reduced interannual trend for the original AWI CS2 product. The obtained IBD results were generally consistent with the expected range and evolution of SYI bulk densities. Given the limitations in the IBD parameterizations, we also proposed a new age-related IBD climatology, and demonstrated the high confidence of the new age-related IBD climatology, with relative biases of 0.3 % for SYI and 0.7 % for FYI. Taken together, we suggest that our multifaceted efforts with the MOSAiC observations can advance the future of sea ice remote sensing. Please refer to "3.2 Seasonal variability of IBD during MOSAiC", "3.3 Comparison of IBD during MOSAiC with historical measurements", "4.3 Evaluation of IBD parameterizations", and "4.5 Implications of IBD parameterizations for the sea ice thickness retrieval" for specific revisions.

*Line 520: Repeating this statement doesn't make it more evident. Without scale adjustment, the density estimate from ICESat-2, and buoys not only show an opposite decreasing seasonal trend to in situ measurements, but they also give unrealistic density values above 950 kg/m³. Therefore, it is not clear why this difference in correlation should be compared between different methods.*

**Reply:** This statement has been removed and the details of the spatial scale adjustment have been inserted. Please refer to "2.2.2 Spatial scale adjustment " for specific revisions.

*Line 548: Previously, it was mentioned that seasonality of ice density is significant. Yet, none of the suggested parameters are related to seasons. **Ice thickness has a large range from thin ice in Atlantic sector to thick ice in Pacific sector during the same sampling time.** Therefore, ice freeboard cannot be an accurate factor related to the seasonality of ice density.*

**Reply:** As mentioned above, the samples used for IBD parameterization themselves exhibited time-varying properties, such as ice porosity, temperature, salinity, thickness, and age. Therefore, we suggest that IBDs estimated from the parameterizations are still capable of exhibiting potential seasonal variations, as evidenced by both independent validation (Fig. A8) and preliminary applications (Fig. A9). We also incorporated the physical factors to improve the physical interpretability of IBD parameterizations. Following the potential applicability ranges, spatial

scales and sea ice types of the parameterizations, we suggest that it is still possible to reasonably predict IBD for different thicker and thinner ice layers in the same season.

*Line 580: What is the basis of this claim? A typical modal ice thickness is around 1.7 m (Sumata et al., 2023) which includes both FYI and SYI. Why FYI cannot have 0.8-1.8 m thickness, only SYI? Also, there is a substantial difference in ice thicknesses wit thinner FYI in the start of Transpolar Drift and older ice closer to Canadian coast, where ice thickness is also typically larger than 0.8-1.8 m.*

**Reply:** This sentence has been removed in the revision.

*Line 627: If you claim that L-site ice was FYI and SYI, while DN was mainly SYI, why then snow freeboard from ICESat-2 and ALS showed such a great agreement despite ALS covering only a relatively small area around MOSAiC ice floe? Also, if most of ice was SYI, why ice density seasonality is close to the FYI in situ measurements, not SYI with no seasonality?*

**Reply:** This sentence has been removed in the revision. In fact, the modal total freeboard derived from ICESat-2 at the DN scale, ICESat-2 at the L-site scale, and ALS at the L-site scale are comparable. In the revised version, we have emphasized that the IBD results only represent the buoy-deployed ice layers (SYI-dominated), as described above. The spatial adjustment term systematically adapted the modal total freeboard to be compatible with the buoy-deployed ice cover. Moreover, we attribute the proximity of SYI-dominated regional IBDs to FYI field measurements to the high spatial heterogeneity of SYI, as evidenced by Hornnes et al. (2024). Please refer to "2.2.2 Spatial scale adjustment " and "4.2 Possible factors contributing to the IBD seasonality" for specific revisions.

*Line 675: The study from Kovacs does not properly describe methodology of density measurements. You describe it as "theoretical equation", what does it mean? There is a few better documented ice density reports, why this specific study was used?*

**Reply:** This statement has been removed. The K97 parameterization has been employed in several sea ice thickness retrievals (e.g., Kwok and Cunningham, 2015), making it valuable to compare our parameterizations with K97 to identify uncertainties, particularly since K97 exhibits characteristics **opposite** to those observed in the MOSAiC coring results.

*Line 680: What is the basis for such claims? There are in situ measurements of ice density for relatively thin ice (20-30 cm), and those values are mostly around 900-910 kg/m³ (10.1016/0165-232X(95)00007-X), which is close to MOSAiC values. Also, MOSAiC ice was quite thin, around 40 cm. Sampling ice thinner than 30 cm is typically considered unsafe, and there is a very short time when ice is that thin in Central Arctic, as thin ice growth exponentially faster.*

**Reply:** Thanks to your addition, we have clarified this.

*Line 682: Similarly, why a method based on substantial adjustments to be able to get realistic estimates of ice density can support limitations of more accurate method, based on measurements of ice density? What exactly was done to support this statement?*

**Reply:** We clarified this statement. Considering the differences in measurement methods and spatial scales, we prefer to compare the results from local and regional-scale IBD parameterization in the revised manuscript. Additionally, we have included a physical analysis and independent

validation to comprehensively recommend potential IBD parameterization schemes for future research. **Revised to** *"Overall, we emphasize that the IBD parameterization based on sampling from individual points has the spatial representativeness-related limitations for the application to satellite sea ice thickness retrievals. Nevertheless, ice core sampling remains an indispensable method for the direct acquisition of sea ice properties."* Please refer to "4.4. Intercomparison of IBD parameterizations" for specific revisions.

*Line 697: The was not any seasonality in SYI density estimates. A strong seasonality and corresponding dependence of ice density from its thickness was only observed for FYI. Therefore, why the parametrization based on FYI measurements can optimize retrieval of SYI? This should be better explained.*

**Reply:** This sentence has been modified in the revision. As discussed above, the IBD differences between SYI-dominated buoy sites and local SYI cores suggest a possible high degree of spatial heterogeneity in SYI porosity, which partly explains the greater similarity to the FYI results. We suggest that the IBD parameterizations derived from the **SYI-dominated buoy sites** are still capable of exhibiting seasonal variations for SYI bulk densities. More importantly, the predictive accuracy of the regional IBD parameterizations based on sea ice parameters even outperformed those derived from the MOSAiC core samples. **Added:** *"In particular, Hornnes et al. (2024) found that the spatial heterogeneity of SYI porosity was much greater than that of FYI during the GoNorth expedition in October 2022, and observed similar air volume fractions in FYI (2.5 ± 0.5 %) and SYI cores (2.8 ± 0.5 %)."*

*Line 707: This is generally surprising that SYI is treated as a separate ice type from FYI. Maybe it could be reasonable many years ago, but for MOSAiC observations with SYI having low-salinity remnant layer below 40-80 cm (Oggier et al., 2023), most of SYI thickness by the end of winter is FYI grown under SYI, which may have density values like pure FYI. Since all such data is available, it is worth discussing. In brief, MOSAiC SYI was much closer to FYI than SYI sampled by Jutila et al., with at least two times higher thicknesses.*

**Reply:** This sentence has been modified in the revision. We agree with the reviewer's mention of the low salinity residual layer and have added the analyses of bulk salinity, bulk temperature, brine fraction volume, and air volume fraction accordingly. We have also highlighted the difference in SYI thickness between MOSAiC and J22.

**Revised to** "*Meanwhile, MCS-FYI exhibited a slow brine discharge with a decreasing trend of 0.5 % per month for its bulk brine volume (Fig. 7d) and –0.2 ppt per month for its bulk salinity (Fig. 7b). In contrast, the bulk salinity of MCS-SYI showed an increasing trend of 0.3 ppt per month, which is closely related to the formation of salty ice layers (similar to FYI) at the base.* " Please refer to "3.2 Seasonal variability of IBD during MOSAiC" for specific revisions.

**Revised to** "*Moreover, MCS-SYI had a higher IBD than the IceBird-derived SYI density in April with a difference of 11 kg m$^{-3}$, mainly due to the significantly greater SYI thickness for the IceBird measurements than for the MOSAiC measurements.* "Please refer to "3.3 Comparison of IBD during MOSAiC with historical measurements" for specific revisions.

*Line 734: You previously mentioned that without scale adjustments your estimates would be 50 kg/m³, therefore you should explain how this bias should be removed for future analysis prior to providing parametrization based on adjusted estimates.*

**Reply:** We added suggestions on how to overcome methodological challenges in the future. We can also foresee that the establishment of a regional coring network spanning different seasons and ice types will greatly advance the parameterization of IBD.

**Added:** *"Since both snow depth and total freeboard can be obtained from aerial observations, even at larger scales of several hundred kilometers, strengthening the aerial observations of these two parameters will further optimize the IBD parameterizations."*

*"To further develop the IBD parameterization for the pan-Arctic Ocean, we recommend extending the observational data collection to the MYI regions and deformed ice fields. "*

*"Improving the buoy deployment strategy through comparative evaluation with large-scale observations will allow for more robust multi-source data matching to more accurately retrieve IBDs."*

Please refer to "4.1 Uncertainties and limitations of IBD retrieval" for specific revisions.

*Line 738: You haven't provided a strong proof that the provided parametrization with a strong seasonality in autumn is related to SYI, not FYI. Observations suggest that this is different.*

**Reply:** We would like to clarify that the regional IBDs derived from buoy sites with predominantly SYI did indeed show variations similar to those observed in MCS-FYI. By setting representative regional values and applying a trend assumption, an similar early growth trend in regional IBD was also observed. Consequently, we believe this is related to the significant spatial heterogeneity of SYI porosity well documented by Hornnes et al. (2024). The independent validation in the autumn also well supported the validity of our regional IBD parameterizations.

*Line 742: It is generally looks impossible that estimates from hydrostatic balance have smaller standard deviations than measurements from weighing. First, even in situ measurements of freeboard and thickness give much higher deviations of density as described by Hutchings et al. (10.3189/2015AoG69A814) Second, density of separate layers of ice should not be compared with bulk density.*

**Reply:** Please see our specific answers in **GC3: Standard deviation of IBD results** for more details.

*Line 743: This is one of the most inconsistent claims throughout the whole paper. Can you say specifically what exactly was dominated by FYI and SYI and provide some evidence for this.*

**Reply:** We comprehensively revised the manuscript to clarify that the IBD results pertain solely to the ice layers where the buoys were deployed. The spatial scale adjustment was made to align the modal total freeboard derived from ALS/IS2 with the buoy sites. Therefore, interpretations for IBD that primarily involve SYI are justified and reasonable.

*Line 748: The assertion that warm air intrusions substantially influenced future changes in ice density evolution is significant but not discussed earlier in the text. Could this be elaborated?*

**Reply:** Discussion about the warm air intrusions and sea ice permeability has been inserted. Please refer to "3.2 Seasonal variability of IBD during MOSAiC" for specific revisions.

*Line 756: The criticism of in situ density measurements requires more analysis. How can 15 buoys at the fixed sites capture a broader picture better than around 40 coring events at slightly different sites? Additionally, how does the smaller uncertainty from weighing lead to challenges, as this seems counterintuitive?*

**Reply:** In the revision, we no longer claim the superiority of any method or parameterization derived from regional or local observations. Instead, we focus on comparing the differences between regional (buoy sites, hydrostatic equilibrium method) and local (coring sites, weighing) results, as detailed in GC1 and CG3. We highlight the strengths and weaknesses of the different observational methods and recommend all potential IBD parameterizations derived from these observations using both sea ice parameters or other physical factors, together with their preliminary validation, as a reference for future research. Please refer to "4.1 Uncertainties and limitations of IBD retrieval" and "5 Conclusion" for specific revisions.

**Reference**

Ackley, S.F., Hibler, W.B., Kugrzuk, F., Kovacs, A., Weeks, W.F., 1974. Thickness and roughness variations of arctic multi-year ice. Oceans '74, IEEE International Conference on Engineering in the Ocean Environment, vol. 1, pp. 109–117. Halifax, NS, Canada.

Alexandrov, V., Sandven, S., Wahlin, J., and Johannessen, O.: The relation between sea ice thickness and freeboard in the Arctic, The Cryosphere, 4, 373-380, 2010.

Angelopoulos, M., Damm, E., Simões Pereira, P., Abrahamsson, K., Bauch, D., Bowman, J., Castellani, G., Creamean, J., Divine, D. V., and Dumitrascu, A.: Deciphering the properties of different arctic ice types during the growth phase of MOSAiC: Implications for future studies on gas pathways, Frontiers in Earth Science, 10, 864523, 2022.

Butkovich, T. R.: Density of Single Crystals of Ice from a Temperate Glacier, Journal of Glaciology, 2, 553-559, 1955.

Cox, G. F. N. and Weeks, W. F.: Equations for Determining the Gas and Brine Volumes in Sea-Ice Samples, Journal of Glaciology, 29, 306-316, 1983.

Crabeck, O., Galley, R., Delille, B., Else, B., Geilfus, N. X., Lemes, M., Des Roches, M., Francus, P., Tison, J. L., and Rysgaard, S.: Imaging air volume fraction in sea ice using non-destructive X-ray tomography, The Cryosphere, 10, 1125-1145, 2016.

Crabeck, O., Galley, R. J., Mercury, L., Delille, B., Tison, J.-L., and Rysgaard, S.: Evidence of Freezing Pressure in Sea Ice Discrete Brine Inclusions and Its Impact on Aqueous-Gaseous Equilibrium, Journal of Geophysical Research: Oceans, 124, 1660-1678, 2019.

Farrell, S. L., Kurtz, N., Connor, L. N., Elder, B. C., Leuschen, C., Markus, T., McAdoo, D. C., Panzer, B., Richter-Menge, J., and Sonntag, J. G.: A first assessment of IceBridge snow and ice thickness data over Arctic sea ice, IEEE Transactions on Geoscience and Remote Sensing, 50, 2098-2111, 2011.

Haas, C.: Dynamics versus thermodynamics: The sea ice thickness distribution, Sea ice, 82, 113-152, 2010.

Hoppmann, M., Kuznetsov, I., Fang, Y. C., and Rabe, B.: Mesoscale observations of temperature and salinity in the Arctic Transpolar Drift: a high-resolution dataset from the MOSAiC Distributed Network, Earth Syst. Sci. Data, 14, 4901-4921, 2022.

Hornnes, V., Salganik, E., and Høyland, K. V.: Relationship of physical and mechanical properties of sea ice during the freeze-up season in Nansen Basin, Cold Regions Science and Technology, 2024. 104353, 2024.

Hornnes, V., Salganik, E., and Høyland, K. V.: Relationship of physical and mechanical properties of sea ice during the freeze-up season in Nansen Basin, Cold Regions Science and Technology, 229, 104353, 2024.

Hutchings, J. K., Heil, P., Lecomte, O., Stevens, R., Steer, A., and Lieser, J. L.: Comparing methods of measuring sea-ice density in the East Antarctic, Annals of Glaciology, 56, 77-82, 2015.

Jutila, A., Hendricks, S., Ricker, R., von Albedyll, L., Krumpen, T., and Haas, C.: Retrieval and parameterisation of sea-ice bulk density from airborne multi-sensor measurements, The Cryosphere, 16, 259-275, 2022.

Koo, Y., Lei, R., Cheng, Y., Cheng, B., Xie, H., Hoppmann, M., Kurtz, N. T., Ackley, S. F., and Mestas-Nuñez, A. M.: Estimation of thermodynamic and dynamic contributions to sea ice growth in the Central Arctic using ICESat-2 and MOSAiC SIMBA buoy data, Remote Sensing of Environment, 267, 112730, 2021.

Kovacs, A.: Estimating the full‐scale flexural and compressive strength of first‐year sea ice, Journal of Geophysical Research: Oceans, 102, 8681-8689, 1997.

Krumpen, T., Birrien, F., Kauker, F., Rackow, T., von Albedyll, L., Angelopoulos, M., Belter, H. J., Bessonov, V., Damm, E., and Dethloff, K.: The MOSAiC ice floe: sediment-laden survivor from the Siberian shelf, The Cryosphere, 14, 2173-2187, 2020.

Kwok, R. and Cunningham, G.: Variability of Arctic sea ice thickness and volume from CryoSat-2, Philosophical Transactions of the Royal Society A: Mathematical, Physical and Engineering Sciences, 373, 20140157, 2015.

Landy, J. C., Tsamados, M., and Scharien, R. K.: A Facet-Based Numerical Model for Simulating SAR Altimeter Echoes From Heterogeneous Sea Ice Surfaces, Ieee Transactions on Geoscience and Remote Sensing, 57, 4164-4180, 2019.

Leppäranta, M. and Manninen, T.: The brine and gas content of sea ice with attention to low salinities and high temperatures, 1988. 1988.

Nakawo, M.: Measurements on air porosity of sea ice, Annals of Glaciology, 4, 204-208, 1983.

Ono, N.: Specific heat and heat of fusion of sea ice, Physics of Snow and Ice: proceedings, 1, 599-610, 1967.

Perovich, D., Smith, M., Light, B., and Webster, M.: Meltwater sources and sinks for multiyear Arctic sea ice in summer, The Cryosphere, 15, 4517-4525, 2021.

Petrich, C. and Eicken, H.: Overview of sea ice growth and properties, Sea ice, 2017. 1-41, 2017.

Petty, A. A., Tsamados, M. C., Kurtz, N. T., Farrell, S. L., Newman, T., Harbeck, J. P., Feltham, D. L., and Richter-Menge, J. A.: Characterizing Arctic sea ice topography using high-resolution IceBridge data, The Cryosphere, 10, 1161-1179, 2016.

Pustogvar, A. and Kulyakhtin, A.: Sea ice density measurements. Methods and uncertainties, Cold Regions Science and Technology, 131, 46-52, 2016.

Ricker, R., Hendricks, S., Perovich, D. K., Helm, V., and Gerdes, R.: Impact of snow accumulation on CryoSat‐2 range retrievals over Arctic sea ice: An observational approach with buoy data, Geophysical Research Letters, 42, 4447-4455, 2015.

Salganik, E., Crabeck, O., Fuchs, N., Hutter, N., Anhaus, P., and Landy, J. C.: Impacts of air fraction increase on Arctic sea-ice thickness retrieval during melt season, EGUsphere, 2024, 1-30, 2024.

Salganik, E., Hornnes, V., Rübsamen-Von Döhren, P. J., Panchi, N., and Høyland, K. V.: First- and second-year sea-ice salinity, temperature, and density during GoNorth 2022 leg 1. PANGAEA, 2023.

Shi, H., Lee, S.-M., Sohn, B.-J., Gasiewski, A. J., Meier, W. N., Dybkjær, G., and Kim, S.-W.: Estimation of snow depth, sea ice thickness and bulk density, and ice freeboard in the Arctic winter by combining CryoSat-2, AVHRR, and AMSR measurements, IEEE Transactions on Geoscience and Remote Sensing, 2023. 2023.

Timco, G. and Frederking, R.: A review of sea ice density, Cold regions science and technology, 24, 1-6, 1996.

Wang, Q., Zong, Z., Lu, P., Zhang, G., and Li, Z.: Probabilistic estimation of level ice resistance on ships based on sea ice properties measured along summer Arctic cruise paths, Cold Regions Science and Technology, 189, 103336, 2021.

**Responses to the Referee 2**

We sincerely appreciate the reviewer's positive and constructive comments on our manuscript. We have made careful and thorough modifications according to your suggestions in the revised manuscript. The original RC2 comments are shown in *red italics*, followed by our responses in standard black. Fig. B and Table B represent the additional figure and table information, respectively. Moreover, specific locations of the modifications in the revised manuscript are highlighted in purple.

*This manuscript, titled "**Estimating seasonal bulk density of level sea ice using the data derived from in situ and ICESat-2 synergistic observations during MOSAiC**", deals with the sea ice density, one of the key parameter of the sea ice. Sea ice density is indicative of the thermodynamic history of the sea ice, and it plays an essential role in estimating the sea ice thickness during satellite altimetry. However, it is also one of the most poorly observed parameter. So I consider the topic, as well as the major results, are highly relevant to this journal's scope. A large body of a variety of dataset are used for deriving the IBD, including in-situ, airborne and satellite data. Relevant issues and extensions to the work, such as the impact on satellite altimetry are also covered. I consider the majority of the work is reasonable in its current form. However, I have the following comments for major revisions before it be considered for acceptance.*

**Overall Reply** In the revision, we have added analyses of the physical interpretation and validity of the IBD parameterizations, as well as the associated uncertainties and limitations for their applications. Please see below for more details.

*GC1: First, one key issue is the independency of the estimated IBD and the contributing factors. It is important to differentiate between direct measurements of IBD and the derived IBD. Ideally, only the direct measurements (in this case, coring) should be used for the parameterization of IBD, which however, is not available at large scale. If the derived IBD is used/evaluated, one should be very careful to make sure that it is independent from other parameter used. For example in Eq. 2, IBD is estimated, but not directly observed. One notable example is that the good correlation in Fig. 8.f totally disappears in Fig. 9.f. I wonder whether this contrast can be really explained by the scale of the observation, but not by something else. Is the significant correlation in Fig. 8.f largely due to the way IBD is derived in Eq. 2? Since the ice freeboard is computed as (hf-hs) and the ratio to the ice thickness is directly written in the equation of: ((hf-hs)/hi), then is the correlation physical, or did it just arise because how you **ESTIMATE** the ice density? One should be very careful here, since uncertainty in the input values could be wrongly interpreted as correlation, since the uncertainty is transferred to the derived value. Then the correlation could be purely artificial. At least some more explanation is needed here. Especially please pay attention to representation issues for DN and L-site studies, which should be at least discussed more thoroughly in Sec. 4.*

**GC1 Reply** We fully agree with the reviewer's concerns about the representativeness and independency issues associated with the regional IBD parameterizations. In the revision, we have provided additional results to address and clarify these issues. In general, we have shown that the regional IBD parameterizations remain valid despite the influence of error propagation, as demonstrated by both physical factor analysis and independent validation. Meanwhile, we

recommended all potential IBD parameterizations derived from regional or local measurements, incorporating both sea ice parameters and physical factors. Considering the limitations of the IBD parameterization, we also proposed a new age-related IBD climatology for satellite retrievals of sea ice thickness.

**GC1: Representativeness**

**We have clarified the representativeness of the developed IBD parameterizations.** Throughout the freezing season, the IBD parameterization derived from the regional buoy sites included a total of 78 samples, with 47 at the DN scale (FYI/SYI = 4/11) and 31 at the L-site scale (FYI/SYI = 4/7), as shown in Fig. B1. In contrast, the local scale comprised 23 samples, including 10 from FYI cores and 13 from SYI cores. The samples used for parameterization themselves exhibited time-varying properties, such as ice porosity, temperature, salinity, thickness, and age. In the revision, we focus on comparing *potential* IBD parameterizations derived from regional (Fig. B2; hydrostatic equilibrium method) and local estimates (Fig. B3; weighing) considering the inherent differences in their spatial scales, measurement methods, sample sizes, and ice types. Please refer to "2.2 Methods" and "3.4 Potential parameterization for IBD" for specific revisions.

[Figure]

**Figure B1.** Flowchart of the IBD retrieval process.

[Figure]

**Figure B2.** Potential parameterizations for IBD at the regional scale and comparison with results at the local scale, including regression models using **(a, b)** sea ice freeboard, **(c, d)** ice freeboard-to-total freeboard ratio, and **(e, f)** ice freeboard-to-thickness ratio. Each panel shows model fit metrics, including the coefficient of determination ($R^2$) and root mean square error (*RMSE*, kg m$^{-3}$). The significance level for the statistical test is set at 95 %. The total number of samples *(Num)* is 78 for the regional scale and 23 for the local scale.

[Figure]

**Figure B3.** Potential parameterizations for IBD at the local scale and comparison with results at the regional scale, including regression models using **(a, b)** sea ice thickness, **(c, d)** sea ice draft, and **(e, f)** total thickness. Each panel shows model fit metrics, including the coefficient of determination ($R^2$) and root mean square error (*RMSE*, kg m$^{-3}$). The significance level for the statistical test is set at 95 %. The total number of samples (*Num*) is 23 for the local scale and 78 for the regional scale.

The parameterized equations for IBD at the regional scale using sea ice parameters are defined as follows:

$$\overline{\rho_i(x,y,t)} = a1 \times \overline{X(x,y,t)} + a2, \tag{B1}$$

where the $\overline{\rho_i(x,y,t)}$ and $\overline{X(x,y,t)}$ represent the regional IBD and other sea ice parameters at any given grid cell over tens of kilometers $(x,y)$ and time $(t)$, respectively. Table B1 lists the regression coefficients and their 95 % confidence intervals for Eq. (B1).

The parameterized equations for IBD at the local scale using sea ice parameters are defined as follows:

$$\rho_i(x,y,t) = a1 \times \left(\frac{1}{X(x,y,t)}\right)^{a2} + a3, \tag{B2}$$

where the $\rho_i(x,y,t)$ and $X(x,y,t)$ represent the local IBD and other sea ice parameters at any given site $(x,y)$ and time $(t)$, respectively. Table B2 lists the regression coefficients and their 95 % confidence intervals for Eq. (B2).

Table B1. Regression coefficients of the parameterized equations based on sea ice parameters at the regional scale (kg m$^{-3}$), and parentheses represent 95 % confidence intervals for the regression coefficients.

| $\overline{X(x,y,t)}$ | Variable type | a1 | a2 | Input range |
|---|---|---|---|---|
| Sea ice freeboard (m) | *Univariate* | −127 (−204, −51) | 923 (915, 932) | 0.05−0.15 m |
| Ice freeboard-to-total freeboard ratio | *Bivariate* | −115 (−158, −71) | 950 (935, 966) | 0.25−0.45 |
| Ice freeboard-to-thickness ratio | *Bivariate* | −953 (−1036, −870) | 980 (974, 986) | 0.05−0.09 |

Table B2. Regression coefficients of the parameterized equations based on sea ice parameters at the local scale (kg m$^{-3}$), and parentheses represent 95 % confidence intervals for the regression coefficients.

| $X(x,y,t)$ | Variable type | a1 | a2 | a3 | Input range |
|---|---|---|---|---|---|
| Sea ice thickness (m) | *Univariate* | −0.6 (−1.7, 0.6) | 4 (2, 7) | 912 (910, 914) | 0.4−2.2 m |
| Sea ice draft (m) | *Univariate* | −0.4 (−1.2, 0.4) | 4 (2, 7) | 912 (910, 913) | 0.4−2.0 m |
| Total thickness (m) | *Bivariate* | −0.7 (−2.1, 0.7) | 5 (2, 8) | 912 (910, 914) | 0.5−2.3 m |

**GC1: Independent Physical Factor**

In order to enhance the physical interpretation of the IBD parameterization, we also introduced several independent physical factors. Physical factors at the regional scale included air and ice bulk temperatures derived from buoy sites, while at the sampling point, sea ice bulk temperature, bulk salinity, brine volume fraction, and air volume fraction derived from ice core samples were included. Please refer to "2.2.5 Principles for IBD parameterization" and "3.4 Potential parameterization for IBD" for specific revisions.

The analysis revealed that sea ice bulk temperature effectively improved the performance of the regional IBD parameterizations (**Fig. B4**), particularly the parameterization using sea ice freeboard ($R^2$ improved by ~85 %), while air temperature was not effective, suggesting its insufficiency in explaining the seasonal variability of IBD. However, when the relationship between the two temperature variables and IBD was considered individually, neither showed statistical significance (P > 0.05).

[Figure]

**Figure B4.** Potential parameterizations for IBD at the regional scale using using both sea ice parameters and the physical factor of ice bulk temperature, including **(a)** sea ice freeboard and sea ice bulk temperature, **(b)** ice freeboard-to-total freeboard ratio and sea ice bulk temperature, **(c)** ice freeboard-to-thickness ratio and sea ice bulk temperature. Each panel shows model fit metrics, including the coefficient of determination ($R^2$) and root mean square error *(RMSE*, kg m$^{-3}$). **Note:** The significance level for the statistical test is set at 95% and the total number of samples is 78.

- **The parameterized equations for IBD at the regional scale using both sea ice parameters and the physical factor of ice bulk temperature are defined as follows:**

$$\overline{\rho_i(x,y,t)} = a1 \times |\overline{\delta(x,y,t)}| + a2 \times \overline{X(x,y,t)} + a3, \tag{B3}$$

where $|\overline{\delta}|$ represent the absolute value of sea ice bulk temperature at any given grid cell over tens of kilometers ($x, y$) and time ($t$). **Table B3** lists the regression coefficients and their 95 % confidence intervals for Eq. (B3). The input ranges for the sea ice parameters are consistent with those listed in **Table B1**, while the range for the sea ice bulk temperature is from −12 to −4 °C.

**Table B3.** Regression coefficients of the parameterized equations using both sea ice parameters and sea ice bulk temperature at the regional scale (kg m⁻³). The parentheses represent 95 % confidence intervals for the regression coefficients.

| $\overline{X(x,y,t)}$ | Variable type | a1 | a2 | a3 |
|---|---|---|---|---|
| Sea ice freeboard (m) | *Univariate* | 2.1 (0.7, 3.5) | −198 (−290, −106) | 913 (902, 924) |
| Ice freeboard-to-total freeboard ratio | *Bivariate* | 2.7 (1.5, 3.8) | −155 (−197, −112) | 942 (928, 956) |
| Ice freeboard-to-thickness ratio | *Bivariate* | 1.2 (0.8, 1.6) | −967 (−1036, −899) | 970 (964, 976) |

We further found significant correlations between physical factors and IBD data from local coring samples (Fig. B5). A negative linear relationship was observed between IBD and both sea ice bulk temperature and air volume fraction, while a power function best described the relationship with sea ice bulk salinity and brine volume fraction. Except for air volume fraction, incorporating any other physical factor into the local IBD parameterizations based on sea ice parameters did not improve the original fitting performance (Table B4), possibly related to a more complex nonlinear relationship between these factors and IBD.

[Figure]

**Figure B5.** Potential parameterizations for IBD at the local scale using physical factors, including regression models using **(a)** sea ice bulk temperature, **(b)** air volume fraction, **(c)** sea ice bulk salinity, and **(d)** brine volume fraction. Each panel shows model fit metrics, including the coefficient of determination ($R^2$) and root mean square error *(RMSE*, kg m⁻³). **Note:** The significance level for the statistical test is set at 95 % and the total number of samples (*Num*) is 23.

**Table B4.** Fitting performance of different parameter combinations with IBD based on a multivariate linear regression model. The variance inflation factor (VIF) of these equations is less than 2. **Note** that **SID**, **SIT**, and **TOT** represent ice draft, ice thickness, and total thickness, respectively. **Tem**, **AV**, **Sal**, and **BV** are ice bulk temperature, air volume fraction, ice bulk salinity, and brine volume fraction, respectively.

| Combination | $R^2$ | RMSE ( kg m$^{-3}$ ) |
|---|---|---|
| SID + Tem | 0.30 | 5.3 |
| SID + AF | 0.95 | 1.4 |
| SID + Sal | 0.49 | 4.6 |
| SID + BV | 0.56 | 4.2 |
| TOT + Tem | 0.35 | 5.3 |
| TOT + AF | 0.95 | 1.4 |
| TOT + Sal | 0.52 | 4.5 |
| TOT + BV | 0.58 | 4.2 |
| SIT + Tem | 0.30 | 5.3 |
| SIT + AF | 0.95 | 1.4 |
| SIT + Sal | 0.56 | 4.2 |
| SIT + BV | 0.56 | 4.2 |

- **The parameterized equations for IBD at the local scale using physical factors are defined as follows:**

$$\rho_i(x, y, t) = a1 \times X(x, y, t) + a2, \tag{B4}$$

$$\rho_i(x, y, t) = a1 \times Y(x, y, t)^{a2} + a3. \tag{B5}$$

where $X$ represents the sea ice bulk temperature or air volume fraction, and $Y$ is the sea ice bulk salinity or brine volume fraction, at any given site $(x, y)$ and time $(t)$. Table B5 lists the regression coefficients and their 95 % confidence intervals for Eqs. (B4−5).

**Table B5.** Regression coefficients (kg m$^{-3}$) of the parameterized equations using physical factors at the local scale. The parentheses represent 95 % confidence intervals for the regression coefficients.

| Parameters | Variable type | a1 | a2 | a3 | Input range |
|---|---|---|---|---|---|
| Sea ice bulk temperature (°C) | $X(x, y, t)$ | −1 (−2, 0.1) | 900 (892, 909) | n/a | 4−12 (below zero) |
| Air volume fraction (%) | $X(x, y, t)$ | −7.1 (−7.8, −6.3) | 919 (918, 920) | n/a | 0.5−4 |
| Sea ice bulk salinity (ppt) | $Y(x, y, t)$ | −2 (−10, 8)×10$^{(-4)}$ | 6.5 (3.2, 9.8) | 912 (910, 915) | 0.5−6 |
| Brine volume fraction (%) | $Y(x, y, t)$ | −0.07 (−0.4, 0.3) | 2.6 (0.3, 5.0) | 913 (909, 917) | 1.5−8 |

**n/a:** not applicable

**GC1: independency issues**

We presented several pieces of evidence that the error propagation from different input parameters in Eq. (2) **has not obscured the potential physical phenomena** revealed by the regional IBD parameterizations (not dominated by error propagation). We suggest that these regional IBD parameterizations, which are representative of the hydrostatic equilibrium-based IBD, and the regression coefficients derived from observations, still provide valuable guidance for satellite retrievals of sea ice thickness. **Four main findings are presented below.** Nevertheless, we also highlighted the inherent limitations of regional IBD parameterizations in the revised version: *"It should also be noted that the fitting performance of the regional IBD parameterizations does not represent absolute robustness because they are not strictly independent of the input parameters."* Please refer to "3.4 Potential parameterization for IBD", "4.1 Uncertainties and limitations of IBD retrieval" and "4.3 Evaluation of IBD parameterizations" for specific revisions.

**Independence of the input parameters and validity of IBD results.** The input parameters in Eq. (2) were independent of each other and exhibited relatively high measurement precision (Section 2.1). Meanwhile, the distinct seasonality observed in these parameters, together with the generally well agreement in both magnitude and seasonality between the derived IBDs and direct coring measurements (**Fig. B6**), effectively supports the validity of the developed parameterizations based on these input parameters. Please refer to "3.2 Seasonal variability of IBD during MOSAiC" for more details.

[Figure]

**Figure B6. (a)** Seasonal variation of IBD estimates at the DN scale (blue bars), L-site scale (red bars), and MCS (green symbols) during the MOSAiC freezing season. The black dots represent the mean values of the regional IBD estimates at the DN and L-site scales. The orange dashed line shows the linear fit for regional IBD estimates from late October to early December ($R = 0.62$, $P < 0.05$), while the purple dashed line indicates the mean ± one standard deviation from late December to April. **(b)** Comparison of regional modeled values, monthly mean of regional buoy estimates, and linearly interpolated MCS results. The error bars for regional buoy estimates represent one standard deviation. **(c)** Box plots of all IBD estimates over the study period, showing the interquartile range (IQR, Q3−Q1, boxes), median (black lines) and outliers (values greater than 1.5 × IQR, red crosses). **(d)** Probability density distribution (PDF) of all IBD estimates over the study period, with labelled values indicating the mean ± one standard deviation.

**High-precision airborne observation.** We used high-precision, spatially coincident airborne measurements from AWI IceBird to further investigate the relationship between ice freeboard-to-thickness ratio and IBD at a spatial resolution of 25 km × 25 km. We identified a similar negative linear function (**Fig. B7**) in the same range as Eq. (B1), which is expected to be less affected by error propagation, thus strengthening our conclusions.

[Figure]

**Figure B7.** Relationship between ice freeboard-to-thickness ratio and IBD at a spatial resolution of 25 km × 25 km based on AWI IceBird measurements. The model fit metrics include the coefficient of determination ($R^2$) and root mean square error ($RMSE$, kg m$^{-3}$). **Note:** The significance level for the statistical test is set at 95 % and the total number of samples ($Num$) is 448.

**Physical interpretation.** By incorporating the physical factor of sea ice bulk temperature into the regional IBD parameterizations based on sea ice parameters, we found significant improvements in fitting performance compared to the original models (**Fig. B4**). This suggests that the initial good fit between different sea ice parameters and IBD is **unlikely to be dominated by computational redundancy or spurious correlations**; otherwise the inclusion of independent physical information would tend to weaken model performa1nce (e.g., Beven, 2001).

**Independent validation with ice cores (the most direct evidence).** An independent validation with coring samples collected from the 2022 GoNorth expedition showed the relative biases of the developed parameterizations ranged from approximately 0 to 0.9 % for SYI and from 0.1 to 2.1 % for FYI, within their respective potential applicability ranges (**Fig. B8**). **The predictive accuracy of the regional IBD parameterizations based on sea ice parameters even outperformed those derived from the MOSAiC core samples, further supporting the validity of our IBD**

**calculations.** The regional IBD parameterizations incorporating both sea ice parameters and physical factors demonstrated well accuracy for both FYI and SYI, particularly the combination considering the ice freeboard-to-thickness ratio and ice bulk temperature. Furthermore, the A10 configuration (882 kg m⁻³) significantly underestimated the observed SYI bulk density, while our newly proposed age-related IBD climatology (907 kg m⁻³) was significantly improved (see details in **GC3 Reply**).

[Figure]

**Figure B8.** Evaluation of the regional and local IBD parameterizations within their respective potential applicability ranges during the GoNorth expedition in late October 2022, including the results for **(a)** SYI and **(b)** FYI, respectively. The updated age-related IBD climatology is also shown. **Note** that **Weighing** represents the direct measurement of IBD using the hydrostatic weighing method. **Reg.** and **Loc.** are the regional and local scales, respectively. **IFB**, **SIT**, **SID**, **TOT**, and **IFB/SIT** represent ice freeboard, ice thickness, ice draft, total thickness, and ice freeboard-to-thickness ratio, respectively. **Tem**, **AV**, **Sal**, and **BV** are ice bulk temperature, air volume fraction, ice bulk salinity, and brine volume fraction, respectively.

*GC2: Besides, it really surpised me that in Fig. 9.a (also in 5.a) that the coring based FYI bulk density increases since October to December. **Is this physical or due to sampling representations?***

**GC2 Reply** We discussed in detail the possible causes of the increasing trend in IBD from late October to early December, which is mainly influenced by the air volume fraction (physical). Please refer to "4.2 Possible factors contributing to the IBD seasonality" for specific revisions.

According to the theoretical relationship between the density of gas-free sea ice and its temperature and salinity (Timco and Frederking, 1996), the observed variations in bulk temperature (from about −4 to −8 °C) and bulk salinity (from about 6 to 5 ppt) of the FYI cores would hardly contribute to an increase in their bulk density from late October to early December (IBD variation < 5 kg m⁻³). **Thus, the significant increase in IBD during the early stages could be closely related to the significant reduction in the air volume fraction** (Fig. B9), as the conversion of liquid water to solid ice in closed brine pockets causes volume expansion, compressing trapped air bubbles and pushing gas into the brine solution as temperatures fall (Crabeck et al., 2016; Crabeck et al., 2019; Salganik et al., 2024).

**The rapid thickening of the sea ice may also have significantly regulated the brine volume and air volume fractions within the ice layer.** For example, the newly formed ice layer will gradually transition from a loose granular structure to a dense columnar structure, resulting in a more compact layer (Oggier and Eicken, 2022). Indeed, the air volume fraction has been shown to be the most important factor influencing IBD, as it is much less dense than sea ice and brine (Timco and Frederking, 1996). Furthermore, no clear evidence of main desalination processes (Petrich and Eicken, 2017) was observed during the early phase of the study, which partly explains the relatively low IBD observed in late October. The warm air intrusion events were not shown to significantly affect sea ice bulk salinity and brine volume fraction, although they were expected to enhance sea ice permeability, which may be related to their shorter duration (Fig. B9).

**Our results also suggest significant spatial heterogeneity of SYI porosity between local (MCS-SYI) and regional (SYI-dominated buoy sites) scales.** During the early period, buoy-derived mean ice thicknesses at the DN and L-site scales showed monthly growth rates of ~15 and 21 %, respectively, with the majority of buoy-monitored ice layers beginning to grow. In contrast, the MCS-SYI showed a relatively low thickness growth rate (~9 % per month) and low porosity with a brine volume fraction of ~1 to 4 % and an air volume fraction of ~0.8 to 1 % (Fig. B9). In particular, Hornnes et al. (2024) found that the spatial heterogeneity of SYI porosity was much greater than that of FYI during the GoNorth expedition in October 2022, and observed **similar air fractions in FYI (2.5 ± 0.5 %) and SYI cores (2.8 ± 0.5 %)**.

[Figure]

**Figure B9.** Seasonal variation of **(a)** sea ice bulk salinity, **(b)** air temperature, **(c)** brine volume fraction, **(d)** sea ice bulk temperature, and **(e)** air volume fraction during the MOSAiC freezing season. The red boxes in panel **(b)** indicate the typical warm air intrusion events with air temperature increasing to near or above −5 °C. The dashed line in panels **(a, c)** represents the seasonal linear trend ($P < 0.01$), while that in panel **(e)** denotes the power function fit ($R^2 = 0.78$, $P < 0.01$).

*Second, I suggest the authors add some more discussion on the satellite altimetry. During the derivation of ice thickness with buoyancy relationship, ice bulk density is one of the model parameters. However, the IBD proposed in this work is parameterized with ice freeboard, freeboard-thickness ratio, or ice thickness. **Neither of these 3 parameters are directly available for altimetry retrieval. Rather, they should be estimated with the retrieved ice thickness.** Even for ice freeboard, a conversion from radar freeboard and snow data is needed, which in itself is a large unresolved issue. So I suggest that at least the detailed treatment be introduced. For example, how radar freeboard is converted into ice freeboard. The bulk density estimations (Fig. 12.c) seem to be more reasonable than AWI's default setting. **But any other proof that the new IBD scheme is better than the default scheme? For example, any proof of better draft or ice thickness retrievals?** Surrounding altimetry I want to mention that according to the authors the IBD scheme is developed for level ice. Although CryoSat-2 currently is not used for ice topography retrieval, ridged ice certainly plays important roles in modulating the CS2 waveform and retrieved freeboard. **Is there a potential problem for applying it to the whole basin, especially given that a lot of deformed ice is present and the density estimation has potential limitations?** Some discussion is suggested to be added in Sec. 4 on this issue.*

**GC3 Reply** **We agree with the reviewers' suggestion and included more details on the satellite altimetry. Moreover, the limitations and uncertainties of the developed IBD parameterizations were also discussed in detail.** Please refer to "4.1 Uncertainties and limitations of IBD retrieval", "4.5 Implications of IBD parameterizations for the sea ice thickness retrieval", and "Text S5. Application of regional IBD parameterizations to the AWI CS2 sea ice thickness product" for specific revisions.

**Extended Application.** We further investigated the impact of regional IBD parameterizations (including three equations) on satellite retrievals of sea ice thickness. For this purpose, we combined satellite altimetry data, ice age field, and regional IBD parameterizations (Eq. B1 and Table B1) to provide preliminary insights into the pan-Arctic SYI bulk densities and their updated thicknesses during the 2010 to 2024 freezing seasons (Fig. B10).

**Processing Chain.** We added details describing how to apply the regional IBD parameterizations to satellite retrievals of sea ice thickness, using the AWI CryoSat-2 sea ice thickness product (AWI CS2) as an example. Specifically, we considered the processing chain of the AWI CS2 as a benchmark, replacing its original IBD settings based on A10 climatology with the three regional IBD parameterizations (Eq. B1). ***Given the length of the main text, we provided specific details in the supplementary material, Please see details below.***

CryoSat-2 initially provided radar backscatter data from the surface target, which can be processed to derive radar freeboard ( $h_{\text{fr}}$ ) by differentiating between ice floes and leads, as well as by retracking radar waveforms to identify the main radar scattering interface. Following the processing chain outlined by Hendricks and Paul (2023), the AWI CS2 algorithm includes a velocity correction term to convert radar freeboard to sea ice freeboard, taking into account the delayed radar velocity in the snow layer and assuming that the main radar scattering interface aligns with the snow-ice interface.

The CryoSat-2 sea ice freeboard can be estimated as follows:

$$h_{\text{fi}}^{\text{cor}} = h_{\text{fr}} + h_{\text{c}}, \tag{B6}$$

$$h_{\text{c}} = \left(\frac{c_{\text{v}}}{c_{\text{s}}} - 1\right) h_{\text{s}}, \tag{B7}$$

where $h_{\text{fi}}^{\text{cor}}$ is the radar velocity-corrected sea ice freeboard, $h_{\text{fr}}$ is the radar freeboard derived using the Threshold First Maximum Retracker Algorithm (TFMRA), $h_{\text{c}}$ represents the velocity correction term; $c_{\text{v}}$ is the velocity of light in a vacuum ($3 \times 10^8$ m s$^{-1}$) and $c_{\text{s}}$ is the radar propagation velocity in the snow layer. Here, $c_{\text{s}}$ was obtained from a snow bulk density-dependent parameterization: $c_{\text{s}} = c_{\text{v}} \times [1 + (5.1 \times 10^{-4})\rho_{\text{s}}]^{-1.5}$ m s$^{-1}$ (Ulaby et al., 1982); $\rho_{\text{s}}$ was derived from a time-dependent equation: $\rho_{\text{s}} = 6.50t + 274.51$ kg m$^{-3}$ ($t$ ranges from 0 to 6 representing October to April) (Mallett et al., 2020).

Assuming hydrostatic equilibrium, the CryoSat-2 sea ice thickness can then be determined:

$$h_{\text{i}} = \left(\frac{\rho_{\text{w}}}{\rho_{\text{w}} - \rho_{\text{i}}}\right) h_{\text{fi}}^{\text{cor}} + \left(\frac{\rho_{\text{s}}}{\rho_{\text{w}} - \rho_{\text{i}}}\right) h_{\text{s}}. \tag{B8}$$

where $h_{\text{i}}$, $h_{\text{fi}}^{\text{cor}}$, and $h_{\text{s}}$ represent the sea ice thickness, sea ice freeboard, and snow depth, respectively. $\rho_{\text{w}}$, $\rho_{\text{i}}$, and $\rho_{\text{s}}$ are the seawater density, sea ice bulk density, and snow bulk density, respectively. Here, $\rho_{\text{w}}$ was set to 1024 kg m$^{-3}$, $\rho_{\text{s}}$ derived from the time-dependent equation mentioned above, and $h_{\text{s}}$ was obtained from a merged snow depth product by combing W99 climatology and passive microwave measurements (MW99/AMSR2). $\rho_{\text{i}}$ was configured according to the regional IBD parameterizations and the sea ice type-weighted A10 climatology (AWI original settings), respectively.

Taking the above equations together, the unknown variables are only $\rho_{\text{i}}$ and $h_{\text{i}}$ when applying the regional IBD parameterizations. For the parameterizations that depend on the sea ice freeboard or the ratio of ice freeboard to total freeboard, the IBD can be calculated directly using the individual parameters of $h_{\text{fi}}^{\text{cor}}$ and $h_{\text{s}}$ from AWI CS2. In contrast, the parameterization depending on the ratio of ice freeboard to thickness requires a combination with Eqs. (B6–8) to estimate IBD and sea ice thickness simultaneously. The final results were calculated as the mean IBD and thickness of the pan-Arctic SYI derived using all three regional IBD parameterizations (referred to as updated; Fig. B10). Our processing details are also applicable to other CryoSat-2/ICESat-2 sea ice thickness products.

[Figure]

**Figure B10.** Retrieved IBDs and thicknesses of the pan-Arctic SYI during the 2010 to 2024 freezing seasons based on the AWI CryoSat-2 sea ice thickness product (AWI CS2), including results using the regional IBD parameterizations (updated) and the original settings (sea ice type-weighted A10 climatology). The shaded areas denote mean ± one standard deviation and dashed lines indicate linear trends. **(a)** Multi-year monthly average estimates for IBD from October to April. **(b)** Multi-year monthly average estimates for ice thickness from October to April. **(c)** Freezing season average estimates for IBD from 2010/2011 (10/11) to 2023/2024 (23/24). **(d)** Freezing season average estimates for ice thickness from 2010/2011 (10/11) to 2023/2024 (23/24).

**Potential Improvement. We agree with the reviewer's comment that the IBD parameterization should improve the accuracy of satellite-based sea ice thickness retrievals.** However, at this stage, we aim to demonstrate the validity and improvement of the regional IBD parameterizations themselves compared to the A10 climatology used in most sea ice thickness products, as detailed in the **GC1**. In the context of the CryoSat-2 sea ice thickness retrieval, Landy et al. (2020) demonstrated the uncertainty and contributions of various parameters in the processing chain, with IBD being only one of the key parameters. **This suggests that improvements in sea ice thickness retrieval by applying the regional IBD parameterization may be significantly influenced by other factors,** such as surface roughness, radar penetration, snow depth, and snow bulk density, leading to a bias compensation effect. In particular, both the W99-based snow depth and the assumption of full radar penetration in the snow layer used in the AWI CS2 processing chain may have contributed to an overestimation of sea ice thickness. **The accuracy of the parameterizations themselves have been reported in the independent validation. Therefore, we focus on reporting the impact of the regional IBD parameterization on the retrieval of sea ice thickness, highlighting the "directional changes" in its relative magnitude, seasonality and interannual variations. More importantly, the retrieved SYI bulk densities were in good agreement with their expected range and seasonal evolution.**

In the revision, we have highlighted that future incorporation of pan-Arctic SYI bulk temperatures, combined with more accurate sea ice ancillary parameters, will be more helpful in retrieving SYI bulk densities. Overall, the incorporation of regional IBD parameterizations into the current AWI CS2 product introduced more realistic variations in IBD, resulting in systematically higher SYI thickness, lower seasonal growth rate, and reduced interannual trend.

**Caveats and limitations.** We carefully discussed the impact of ice deformation on the application of our regional IBD parameterizations, which were developed solely from level ice observations.

**In Section 4.1, we have added:** *"In terms of interpretation, the IBD results at the DN and L-site scales primarily reflect level ice mixed with SYI (11 buoy sites) and FYI (4 buoy sites), lacking information on other ice ages and types, **particularly deformed ice**, which introduces the potential limitations and uncertainties of the parameterization scheme. The bulk density of deformed sea ice varies depending on the degree of compaction and seawater inclusion. In general, deformed and unconsolidated sea ice has a lower bulk density due to high macroporosity (Petrich and Eicken, 2017), which can lead to an overestimation of satellite-derived sea ice thickness when using IBD parameterizations based on level ice only. Conversely, if deformed sea ice contains relatively high amounts of seawater due to compression or ice interactions, it tends to have a higher bulk density, which may lead to an underestimation (Jutila et al., 2022a). To further develop the IBD parameterization for the pan-Arctic Ocean, we recommend extending the observational data collection to the MYI regions and deformed ice fields."*

**We also summarized the limitations and caveats of this study at the end of the revised manuscript:** *"1) the hydrostatic equilibrium calculations of the IBD have relatively high uncertainties and sub-weekly variability compared to direct measurements; 2) the different MOSAiC observations used do not overlap spatially; 3) the regional IBD parameterizations are*

*not strictly independent of the input parameters; 4) the IBD results represent only level sea ice and lack information on **deformed ice**. We can also foresee that the establishment of a regional coring network spanning different seasons and ice types will greatly advance the parameterization of IBD."*

**Updated IBD Climatology.** Both the MOSAiC and other historical records of IBD all reported higher values than the A10-MYI (up to ~30 kg m$^{-3}$). Therefore, we proposed a new age-related IBD climatology during the freezing season by combining the mean IBD values from the Arctic Sever expedition(Alexandrov et al., 2010; Shi et al., 2023), the IceBird campaigns (Jutila et al., 2022), and the MOSAiC expedition (this study), as shown in Table B6.

The updated IBD climatology showed a significant improvement over the traditional A10 climatology for SYI, **reducing the relative bias from 2.4 to 0.3 %**, well demonstrating its potential for application to satellite retrievals of sea ice thickness (Fig. B8).

**Table B6** Age-related bulk densities of level sea ice during the freezing season derived from a combination of the Arctic Sever expedition (from 1980 to 1989), the IceBird campaigns (April 2017 and 2019), and the MOSAiC expedition (from 2019 to 2020). Uncertainty is given as the maximum difference in age-related mean IBD values across observations.

| Ice age | Mean (kg m$^{-3}$) | Standard deviation (kg m$^{-3}$) | Uncertainty (kg m$^{-3}$) |
|---|---|---|---|
| FYI (0–1 years) | 917 | 7.5 | 20 |
| SYI (1–2 years) | 907 | 6.8 | 13 |
| MYI (2+ years) | 900 | 14.0 | 34 |

**Reference**

Alexandrov, V., Sandven, S., Wahlin, J., and Johannessen, O.: The relation between sea ice thickness and freeboard in the Arctic, The Cryosphere, 4, 373-380, 2010.

Crabeck, O., Galley, R., Delille, B., Else, B., Geilfus, N. X., Lemes, M., Des Roches, M., Francus, P., Tison, J. L., and Rysgaard, S.: Imaging air volume fraction in sea ice using non-destructive X-ray tomography, The Cryosphere, 10, 1125-1145, 2016.

Crabeck, O., Galley, R. J., Mercury, L., Delille, B., Tison, J.-L., and Rysgaard, S.: Evidence of Freezing Pressure in Sea Ice Discrete Brine Inclusions and Its Impact on Aqueous-Gaseous Equilibrium, Journal of Geophysical Research: Oceans, 124, 1660-1678, 2019.

Hendricks, S. and Paul, S.: Product User Guide & Algorithm Specification - AWI CryoSat-2 Sea Ice Thickness (version 2.6) Issued by (v2.6), Zenodo, https://doi.org/10.5281/zenodo.10044554, 2023.

Hornnes, V., Salganik, E., and Høyland, K. V.: Relationship of physical and mechanical properties of sea ice during the freeze-up season in Nansen Basin, Cold Regions Science and Technology, 2024. 104353, 2024.

Jutila, A., Hendricks, S., Ricker, R., von Albedyll, L., Krumpen, T., and Haas, C.: Retrieval and parameterisation of sea-ice bulk density from airborne multi-sensor measurements, The Cryosphere, 16, 259-275, 2022.

Landy, J. C., Petty, A. A., Tsamados, M., and Stroeve, J. C.: Sea Ice Roughness Overlooked as a Key Source of Uncertainty in CryoSat-2 Ice Freeboard Retrievals, Journal of Geophysical Research-Oceans, 125, 2020.

Mallett, R. D. C., Lawrence, I. R., Stroeve, J. C., Landy, J. C., and Tsamados, M.: Brief communication: Conventional assumptions involving the speed of radar waves in snow introduce systematic underestimates to sea ice thickness and seasonal growth rate estimates, Cryosphere, 14, 251-260, 2020.

Oggier, M. and Eicken, H.: Seasonal evolution of granular and columnar sea ice pore microstructure and pore network connectivity, Journal of Glaciology, 68, 833-848, 2022.

Petrich, C. and Eicken, H.: Overview of sea ice growth and properties, Sea ice, 2017. 1-41, 2017.

Salganik, E., Crabeck, O., Fuchs, N., Hutter, N., Anhaus, P., and Landy, J. C.: Impacts of air fraction increase on Arctic sea-ice thickness retrieval during melt season, EGUsphere, 2024, 1-30, 2024.

Shi, H., Lee, S.-M., Sohn, B.-J., Gasiewski, A. J., Meier, W. N., Dybkjær, G., and Kim, S.-W.: Estimation of snow depth, sea ice thickness and bulk density, and ice freeboard in the Arctic winter by combining CryoSat-2, AVHRR, and AMSR measurements, IEEE Transactions on Geoscience and Remote Sensing, 2023. 2023.

Timco, G. and Frederking, R.: A review of sea ice density, Cold regions science and technology, 24, 1-6, 1996.

Ulaby, F., Moore, R., and Fung, A.: Microwave remote sensing: Active and passive. Volume 2-Radar remote sensing and surface scattering and emission theory. 1982.